# Sparse Probes, Murky Physics: Interpretability Challenges in a Foundation Model for Continuum Dynamics

**Katherine Rosenfeld**
Gates Foundation

**Maike Sonnewald**
UC Davis

## Abstract

Generative AI emulators are increasingly used in scientific domains where we already have strong theory, benchmarks, and physical intuition. This raises a central evaluation and interpretability question: when a foundation-style model can reproduce known continuum dynamics, what internal mechanism supports that behavior, is the internal behaviour consistent with known physics, and how does it relate to where the emulator succeeds or fails? We investigate a cross-domain foundation model for continuum dynamics, Walrus by Polymathic, using mechanistic interpretability guided by physical principles. We apply a sparse autoencoder (SAE) to probe a selected layer, and address the practical challenge of triaging a large feature set (over 20,000) using enstrophy as a physically grounded metric. As a deliberately simple testbed, we focus on shear flow and compare feature recruitment across multiple shear-flow setups, i.e. parameter values in the numerical simulation. Across setups we find evidence of piecewise consistency, with subsets of features recurring in similar roles, but this structure is intermittent and does not map cleanly onto standard physical decompositions. In parallel, direct comparisons between numerical simulation and the emulator reveal systematic output-level discrepancies, including regimes where energy/structures become too diffuse or too localized. We connect parts of these discrepancies to changes in specific SAE feature usage. Our work highlights open questions for scientific foundation models: how to robustly prioritize mechanistically meaningful features, how to separate stable structure from analysis artifacts (including single-layer and SAE limitations), and how to use established benchmarks to decide when "different" internal representations are genuinely informative rather than merely effective.

## 1 Introduction

Modern machine-learning (ML) models routinely learn high-dimensional mappings between inputs and outputs that are difficult to reason about directly. These mappings are implemented through a sequence of internal representations—latent spaces, whose geometry and structure are only weakly constrained by the task loss. As a result, even when a model performs well, it is often unclear how information is represented internally, or whether those representations correspond to anything beyond task-specific correlations, i.e., it is unclear if the underlying dynamics have been learned.

This challenge is particularly evident in large models trained on complex data, where internal activations live in spaces with thousands of dimensions and no obvious semantic axes. From this perspective, it is already nontrivial that internal representations exhibit any consistent structure at all. The fact that meaningful patterns can sometimes be extracted from a single layer, or even a single subspace of that layer, is therefore somewhat surprising, and raises the question of what, if anything, these patterns reflect about the system being modeled.

Physics-based systems offer a useful setting in which to probe the question of what is actually learned. While physical systems are not "simple" in any absolute sense, they provide controlled environments where governing equations, symmetries, and conserved quantities offer partial reference points. This makes it possible to probe the ML guided by known principles to learn more about its learned representations, without assuming that they mirror human reasoning, such as in language

models (Carleo et al., 2019). In principle, having a grounding in physics enables external validation through known principles.

Recently, foundation models have gained traction in the physical sciences. These models are pre-trained on broad, diverse data at scale, designed to be fine-tuned for downstream tasks rather than trained from scratch for each one. The model we study, Walrus, is a 1.3B-parameter transformer pre-trained on nineteen distinct physical scenarios spanning astrophysics, geoscience, plasma physics, acoustics, and classical fluids (McCabe et al., 2025). The scale and heterogeneity of such models make interpretability challenging. To make progress, we focus on a simple shear flow, where the governing physics is well understood and known principles provide concrete reference points against which learned features can be compared.

Foundation models like Walrus show astounding results, and one can both be skeptical about whether the model has actually learned the physics it purports to model, or excited to ask if this success could be the key to solving problems that have stymied science for centuries such as turbulence. Addressing both these requires us to determine what the model is *doing* to make its predictions, and effectively if it has the governing relationships, or convincingly, and perhaps also usefully, memorized the data. Following MacMillan & Ouellette (2025), we focus explicitly on the internal mapping learned by a neural model by examining a single intermediate layer through the lens of a sparse autoencoder (SAE) (Cunningham et al., 2023; Gao et al., 2024). The SAE provides a way to decompose dense activations into a sparse set of features, effectively selecting a low-dimensional slice of the model's latent space. Importantly, this analysis does not modify or retrain the original model as it is purely diagnostic.

We apply the SAE method to Walrus, but limit ourselves to a simple shear-flow simulation suite, allowing us to compare learned features across multiple simulation realizations and parameter settings. The simulations vary based on their parameter values, specifically the Reynolds and Schmidt numbers, and we use the enstrophy of the flow as our grounding metric (see section 2.3). This setup lets us ask whether 1) the same feature is related to similar dynamics across Walrus simulations, and 2) if the feature is related to similar dynamics throughout the time-steps of the Walrus simulation. Despite being post-hoc and somewhat qualitative, this allows us to start the process of seeing if Walrus has actually learned physical relationships, assessing how sensitive the features are to changes in the underlying dynamics, and how they do, or do not, relate to familiar physical quantities.

Our goal is not to validate the model, but to use interpretability tools grounded in an SAE to examine the gap between learned internal representations and physical intuition. More broadly, this work is intended as a starting point for a discussion both around what model fidelity means and what the limits of the tools we have are. We believe that addressing such questions is keenly important before such foundation models come into wide circulation for scientific investigations. Our work raises several questions that we believe are particularly relevant for the interpretability and representation-learning communities: Is analyzing isolated slices of latent space a meaningful way to study what models learn? Are we at risk of over-interpreting internal features simply because they are consistent or sparse? And if models do learn reusable structure, to what extent can we extract insights from these representations, rather than merely projecting our own expectations onto them?

## 1.1 RELATED WORK

**Representation Learning and Latent Spaces.** A large body of work has examined how neural networks organize information internally, often through probing methods, feature attribution, or concept-based analyses (Shu et al., 2025). These approaches aim to identify directions or subspaces in latent space that correlate with interpretable properties. However, many such methods rely on external labels or predefined concepts, making it difficult to distinguish genuinely emergent structure from imposed semantics.

Sparse autoencoders provide an alternative by decomposing activations into a sparse set of features without requiring supervision. Recent work has used SAEs to study internal representations in large models, suggesting that even highly entangled activations can sometimes be expressed in terms of a relatively small number of active features (Cunningham et al., 2023; Bricken et al., 2023; ?). Our work builds on this idea, but focuses on cross-simulation consistency as a diagnostic, rather than semantic labeling of individual features.

**AI in Physics and Scientific Modeling.** Machine learning has been widely applied in physics for tasks ranging from surrogate modeling and closure schemes to discovery of governing equations (Chen et al., 2021; Sanderse et al., 2024; Wetzel et al., 2025). While much of this literature emphasizes predictive accuracy or computational efficiency, a smaller subset has explored interpretability and internal structure, often using simplified or idealized systems as testbeds. Such systems make it possible to study learned representations under controlled variations, but they also highlight the tension between physically meaningful quantities and the abstractions favored by data-driven models.

**AI does not think like a mathematician.** Recently, work looking at how AI reasons in mathematics has been illuminating, and offers promise for determining if a model like Walrus has learned useful physics. Neural networks perform mathematical operations very differently from humans (Nikankin et al., 2025). Where a person might apply symbolic rules, recognize problem types, or reason step-by-step through a derivation, a neural network computes through high-dimensional linear transformations and point-wise nonlinearities, operations that bear no obvious resemblance to formal reasoning. This means that even when a model produces correct outputs, the internal process that generated them may not correspond to anything like human mathematical intuition. Features that appear interpretable to us may be artifacts of how we choose to probe the model, rather than reflections of how the model actually represents the problem.

## 2 METHODS

### 2.1 WALRUS

Walrus is a 1.3B parameter transformer-based foundation model developed by Polymathic AI for modeling fluid-like continuum dynamics (McCabe et al., 2025). It was trained on nineteen scenarios spanning astrophysics, geoscience, rheology, plasma physics, acoustics, and classical fluids, and has demonstrated competitive performance against prior foundation models across both short- and long-term prediction horizons and a range of pretraining datasets. The project is fully open source, with published data sources, weights, and supporting code. The model has also been studied using other mechanistic interpretability methods, including steering vectors (Fear et al., 2025). We selected Walrus for our analysis because of its explicit focus on physical scenarios and its position as a leading model in this class (cf. Herde et al. (2024)).

A full description of the Walrus architecture can be found in McCabe et al. (2025); here we highlight several features relevant to our analysis. The model employs convolutional stride modulation to adapt the level of downsampling and upsampling in each encoder and decoder block (Mukhopadhyay et al., 2025). This enables a shared encoder-decoder architecture with a fixed token count of 32 per spatial axis (for 2D scenarios), while allowing the downsampling and upsampling rates to vary according to the task. The core of the model consists of 40 transformer blocks, each containing a temporally causal attention layer and a spatially parallel attention layer (Ho et al., 2019; McCabe et al., 2023). Note that in our analysis we do a single-step forward prediction with a fixed number of input timesteps to $n_t = 6$. The inputs to the model are always taken from the numerical simulation.

The majority of the training data for Walrus is drawn from The Well, a 15TB collection of numerical simulations spanning a wide variety of spatiotemporal physical systems (Ohana et al., 2025). To ensure an in-sample evaluation, we conduct our analysis on the `shear_flow` dataset, one of the datasets included in Walrus' training data.

### 2.2 SHEAR-FLOW

Shear-flow is a classical fluid system characterized by the continuous deformation of adjacent fluid layers sliding past each other with different velocities. Shear flows are non-linear phenomena and are unstable at large Reynolds number, $Re$. Using Walrus, we explore a two-dimensional periodic shear flow governed by the incompressible Navier-Stokes equations:

$$\frac{\partial u}{\partial t} + \nabla p - \nu \Delta u = -u \cdot \nabla u,$$
$$\frac{\partial s}{\partial t} - D \Delta s = -u \cdot \nabla s, \tag{1}$$
$$\nabla \cdot u = 0$$

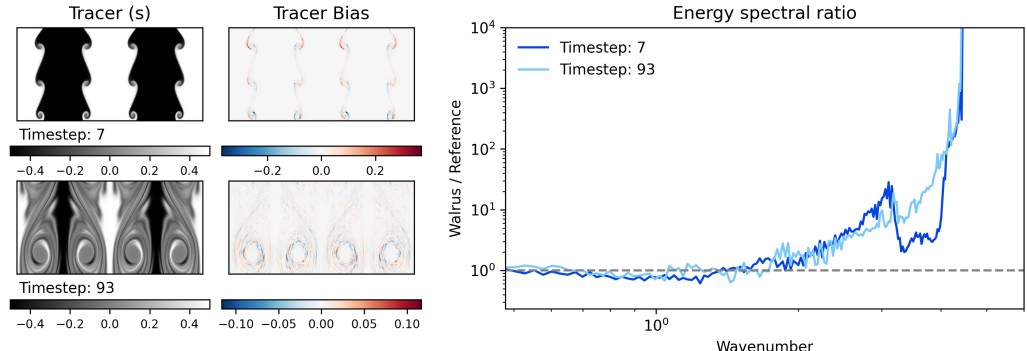

Figure 1: Comparing Walrus to the numerical reference: the tracer field, bias, and resulting energy spectral ratio for our reference shear flow simulation, $\text{Sim}_{50}$, (see section 2.3; $Re = 10^4$, $Sc = 0.2$). We show two time-steps, one from the beginning of the simulation and another after the system has some time to evolve ($t \in \{7, 93\}$).

where $u = (u_x, u_y)$ are horizontal and vertical velocity components, $s$ is the tracer, and $p$ is the pressure. The problem equations are parametrized in `The Well` by the viscosity, $\nu = 1/Re$, and diffusivity, $D = \nu/Sc$ where $Sc$ is the Schmidt number. The initial conditions is set by the number of shears, $n_{\text{shear}}$, along with the position and width. Finally there are $n_{\text{blobs}}$ located at the shear that introduce perturbations to the flows. The data lies on a uniform, cartesian grid ($256 \times 512$) and is composed of the velocity vector field, $u$ and two scalar fields, tracer ($s$) and pressure ($p$). The PDEs were solved using spectral methods for 200 timesteps (Burns et al., 2020; Morel, 2024).

The bias introduced by Walrus for a single timestep projection is indiscernible to the eye (see bias heatmaps in Figure 1). However, the energy spectrum reveals that there are significant and systematic differences introduced by Walrus. In Figure 1, we plot for two timesteps the energy spectral ratio (the ratio between Walrus and the reference numerical simulation). This plot reveals that Walris systematically underpredicts at mid-wavenumber (k ∼ [8-15]) and over-predicts at large wavenumber (e.g., small spatial scale, k 15). The intuitive explanation is that Walrus is producing too much small-scale structure.

Our analysis will focus on three simulations which we call $\text{Sim}_3$ ($Re = 1e4$, $Sc = 0.1$), $\text{Sim}_{50}$ ($Re = 1e4$, $Sc = 0.2$), and $\text{Sim}_{56}$ ($Re = 5e4$, $Sc = 0.1$). Here, as a simple demonstration and due to space constraints, we use figures showing the tracer field, not the velocity fields, as these are effective visual illustrations of physical metrics like enstrophy (see section 2.3). For example, when two or more fluid layers move in opposite directions, these initially appear as columns. At the interface, instabilities can develop into vortices (or "whorls"). In high-viscosity fluids, these dissipate quickly, but in low-viscosity fluids, they persist longer and promote mixing. The vortices represent regions where the kinetic energy from the shear between layers is released. Figure 2 illustrates these principles visually, where the the middle row shows the evolution of the tracer field for simulation 50 across timestep, $t_{\text{step}} \in \{15, 40, 60, 80\}$.

### 2.3 PHYSICS-BASED METRICS

In order to probe whetherWalris has learned something about the physics of energy cascades, we focused on enstrophy, $\mathcal{E}$:

$$\mathcal{E}(u) = \int_{\Omega} |\nabla u|^2 dx \tag{2}$$

which, for an incompressible flow can be described as the integral of the square of the vorticity, $\omega$:

$$\mathcal{E}(\omega) = \int_{\Omega} |\omega|^2 dx \tag{3}$$

Additionally the time derivative of enstrophy, $\dot{\mathcal{E}}$, is a useful metric that reflects the instantaneous balance of injection and dissipation ($\dot{\mathcal{E}} < 0$; dissipation exceeds injection for the two-dimensional shear flow scenario).

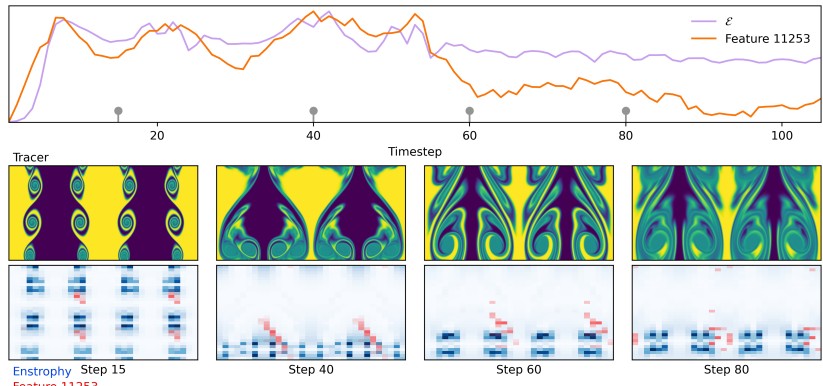

Figure 2: Comparing enstrophy, $\mathcal{E}$, of $\text{Sim}_{50}$ versus total feature activation for the feature with greatest Spearman's rank corelation coefficient ($\rho = 0.85$). We also show the tracer field (middle row) and blue enstrophy overlaid by red feature activation heatmaps (bottom row).

As part of our analysis we use $\mathcal{E}$ to identify a reference simulation, $\text{Sim}_{50}$. To do so we calculate $\mathcal{E}$ and $\dot{\mathcal{E}}$ per time-step for each simulation in our dataset. We then calculate the median $\dot{\mathcal{E}}$ and select the trajectory with the highest value. The parameters for this simulation are $Re = 10^4$ and $Sc = 0.2$. Figure 1 shows two snapshots of the tracer field from the beginning ($t_{\text{step}} = 7$) and middle ($t_{\text{step}} = 93$).

## 2.4 SPARSE AUTO-ENCODER

SAEs have been proven to be a valuable tool for finding highly interpretable features in language models (Bricken et al., 2023; Shu et al., 2025). They work by learning sparse, linear reconstructions of model activations that can help resolve the problem of polysemanticity in the model neurons (Elhage et al., 2022; Cunningham et al., 2023). The application of this method has found an extensions in biology for protein and gene models, radiology, and weather forecasts (Simon & Zou, 2025; Gujral et al., 2025; Guan et al., 2025; Abdulaal et al., 2024; MacMillan & Ouellette, 2025).

We trained a Top-K Sparse Autoencoder (SAE) on activations from a single layer, $l$, ofWalris. We trained the SAE using the Adam optimizer with a learning rate schedule that combines a 5% linear warm up with cosine annealing over the remaining steps (Kingma & Ba, 2017; Murphy, 2022). In each forward pass, the input batch was mean-centered and L2-normalized, then passed through the SAE which returned a main reconstruction, latent codes, and an auxiliary reconstruction from "dead" neurons. We utilized a two part loss function: the first part was the mean squared error (MSE) between the normalized input and the main reconstruction and the second was an auxiliary loss, parameterized by the Top-$k_{\text{aux}}$ dead latents and the size of the dead window, $D$. The combination of Top-K and Aux loss has been shown to help with stabilizing training and avoid feature collapse (Gao et al., 2024). After each backward pass, gradients are clipped to a max norm of one, and two post-optimization constraints are enforced without gradients: the decoder columns are re-normalized to unit norm, and the model's dead neuron mask is updated based on which latent codes were active in the batch. We used Weights & Biases to log the combined loss, MSE, auxiliary loss, dead neuron count, and learning rate throughout training.

We chose to use the activations from the middle block, $l = 20$, focusing on the parallel spatial mixing layer with 2816 activations. Our SAE had an expansion factor of 8 which resulted in 22528 features. Figure 6 shows the two loss components and latent active fraction for a sweep over $k \in \{16, 32, 64\}$. For the main analysis we show $k_{\text{active}} = 32$, $k_{\text{aux}} = 2048$, $D = 5\text{e}6$, $l_r = 3\text{e}{-4}$, $\beta = 0.1$, across 5 epochs of 112 trajectories that comprised our training dataset.

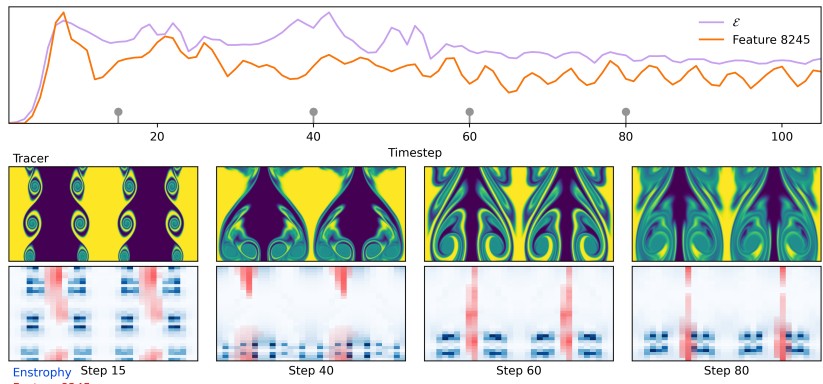

Figure 3: As in Fig. 2 for feature 8245 is $Sim_{50}$. Note the similarity of the feature activations in timestep 15 in Fig. 4

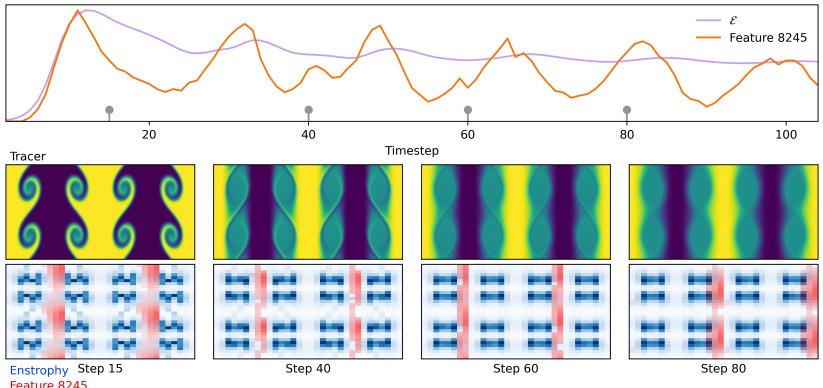

Figure 4: As in Fig. 2 for feature 8245 is $Sim_{56}$. Note the similarity of the feature activations in timestep 15 in Fig. 3

# 3 RESULTS

## 3.1 FEATURE IDENTIFICATION

With the SAE trained, the key challenge is to surface interpretable explanations of the learned features. To identify features that may capture relevant physics, we begin by looking at global metrics (space-averaged, per time step). For our reference trajectory ($Sim_{50}$; see section 2.3), we calculate the Spearman's rank coefficient, $\rho$ and corresponding $p$-value for each individual feature of the SAE between $\mathcal{E}(t)$ and the feature activations summed over all spatial tokens ($32\times32$). The feature that ranked highest had $\rho = 0.85$ ($p$-value $= 3.33e-31$). Given the large number of features (22,528), we conduct a permutation test by calculating $\rho$ across 100 iterations of randomly shuffled $\mathcal{E}$. The 99-th percentile of the permutation null or significant threshold was $\rho=0.30$. This suggest that the correlation found for the feature is significant. However, about 10% of the SAE features have correlation coefficients exceeding this value which suggests that $\mathcal{E}$ is not highly distinguishing (e.g., many values may correlate with $\mathcal{E}$; see figure 8). To compare, we repeat this ranking on $\dot{\mathcal{E}}$. In this case the maximal $\rho = 0.50$ and the significance threshold is $\rho = 0.48$.

Having ranked features by $\rho$ using $\mathcal{E}$, we can now generate maps of the feature activation levels to see where attention is distributed in the data (i.e., $Sim_{50}$). Figure 2 shows the trace of $Sim_{50}$'s

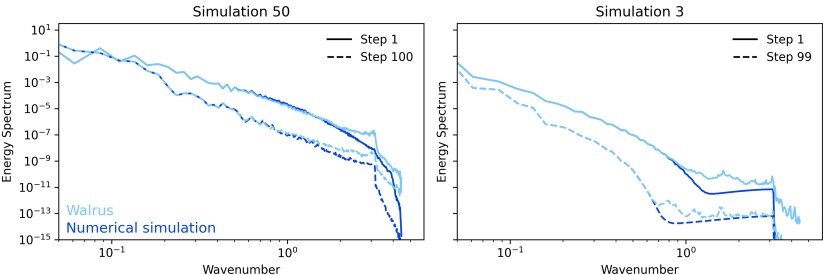

Figure 5: Energy spectra for simulations $\text{Sim}_{50}$ (left) and $\text{Sim}_3$ (right) and two timesteps near the beginning (solid line) and middle (dashed line) of the simulation. We show results from the numerical simulation (dark blue) as well as the Walris single step prediction (light blue).

$\mathcal{E}$ versus the maximal $\rho$ feature as well as the tracer, $\mathcal{E}$ (calculated on a 32×32 grid), and feature activation levels as heatmaps.

We repeated this calculation for $\text{Sim}_3$, $\text{Sim}_{56}$, and all simulations ($\rho = 0.88$) in our dataset. Results are shared in the appendices and discussed in the next section (section 3.2).

## 3.2 Assessing the activation patterns

Clearly, some features are more visually apparent than others, and we structure our discussion of the results around these. We will discuss sorting via individual simulations, using the features from only $\text{Sim}_{50}$, and using the top features across all simulations. Overall, throughout all our analysis of the features, we see that features generally have an alignment with activation on the left or right side of an interface, activating in the center, or is very localized and "noisy" looking patterns. However, this behavior can change completely from one timestep to the next, where we did not find a clear coherence across features we expected to be the same, e.g. similarly shaped whorls.

When looking at the top features individually determined, we saw the clearest patterns, meaning large patches of activation that coincided with recognizable features, in simulation 50. Some features seemed to activate consistently in the channels between the interfaces. Figure 2 illustrates feature 11253, where there is a consistent activation on the right side of the channel, and a coherent diagonal stripe, which moves us and then fades. Feature 1453 in simulation 50 (Appendix A.5) illustrates how a large area in the center is covered by the feature, which stays in the center but moves both up and down. In simulation 56, feature 11253 seems somewhat similar, and feature 1453 appears to have a similar pattern in the first time snapshot but disappears in subsequent snapshots (Appendix A.6).

We note an apparent relationship between how diffuse, from a numerical modeling perspective, a simulation appears and a lack of spatially coherent feature activations for the top features as selected by within the given simulation. For example, in $\text{Sim}_3$ (Appendix A.4), even the top features do not show coherent patterns. In $\text{Sim}_{56}$ (Appendix A.6), which also appears more diffuse from a numerical modeling perspective, there is also less obviously coherent patterns. Notably, simulation 56 does have more coherent patterns if features are sorted according to simulation 50, suggesting that the model concentrates its activations intensely in a manner not apparent using our analysis.

Selecting the top features across the entire batch of simulations, there is little to be seen in simulation 3 (Appendix A.9)) in terms of activations. However, there is a striking similarity between the activations in $\text{Sim}_{50}$ and $\text{Sim}_{56}$ for the first snapshot at time step 15. This similarity, meaning that similar features are picked out persists across all 10 features (Appendix A.10) and A.11), respectively). For example, features 7090 activate the yellow column regions, while feature 8245 activates in the blue columns (see figures 3 and 4. Feature 4206 activates between regions of high enstrophy on the right, while feature 15633 activates in the center of areas surrounded by high enstrophy on the left. However, this coherence does not persist into subsequent timesteps. There seems to be some persistence in if there is an area in a channel in $\text{Sim}_{56}$, but this does not consistently hold for $\text{Sim}_{50}$. Overall, it is clear that a highly complicated relationship between activations and fluid flow properties exists, which becomes increasingly challenging to disentangle further into the simulations.

### 3.3 MODEL AND FEATURE DIFFUSIVITY

The preceding analysis showed that feature activations can appear coherent at early timesteps but lose structure as the simulation evolves. There are two possible sources of this diffusivity, which are physical and representational. In the physical sense, Walrus may over-smooth the flow fields themselves, so the energy spectra show that the model dissipates structure at scales where the simulation retains it, and this bias grows over time and vice versa. This is illustrated in Fig. 5 for $SIM_{50}$ and $Sim_3$. If sharp gradients are smeared out early, later fields offer less coherent dynamics to predict. But diffusivity may also be representational, in that even when physical structure remains, the model's encoding of it may be spread across many weakly active features rather than concentrated in a few interpretable ones. Both effects could compound, physical smoothing leaves less for the representation to capture, while representational spreading makes whatever remains harder to isolate. However, neither pattern is universal. For some parameter combinations, Walrus produces overly localized rather than overly diffuse structure, and feature activations can be sparse yet spatially incoherent. The relationship between output-level errors and internal representations remains unclear.

## 4 DISCUSSION

We trained an SAE on Walris and used a deliberately simple shear-flow testbed to probe whether interpretable physical structure emerges from the model's internal representations. The central finding is that while some features show striking consistency across simulations, particularly at early timesteps, this coherence deteriorates rapidly as the flow evolves. Even restricting analysis to top-ranked features within a single simulation does not recover a stable mapping onto familiar physical quantities. What initially appears to be meaningful structure often dissolves into noise or shifts unpredictably, making it difficult to draw firm conclusions about what Walris has actually learned.

This difficulty reflects the open questions we posed at the outset. First, how should one robustly prioritize mechanistically meaningful features? Our enstrophy-based ranking surfaced correlated features, but roughly 10% of all features exceeded the significance threshold, which suggests that correlation with a single physical metric is insufficient to isolate the features that matter. Second, how can stable structure be separated from analysis artifacts? We cannot rule out that our SAE training, single-layer focus, or feature-ranking methodology introduces spurious patterns, as the lack of persistent structure across timesteps could reflect limitations of our tools rather than properties of Walris itself. Third, how should established benchmarks be used to decide when different" internal representations are genuinely informative? Walrus reproduces shear-flow dynamics reasonably well at the output level, yet its internal encoding does not map cleanly onto the physical decompositions we tested. Whether this means the model has learned something beyond our intuitions or simply found an effective shortcut remains unresolved.

Several practical challenges compound these conceptual issues. The sheer volume of features (over 20,000) makes systematic analysis prone to error and confirmation bias. Many features activate only in small, localized regions, whether this reflects irrelevance or fine-grained specialization is unclear. And as discussed in Section 3.3, both physical and representational diffusivity may obscure structure as simulations progress.

We do not propose a new model or objective here. Rather, we use a controlled physical system to examine how stable a foundation model's internal representations are when viewed through sparse decomposition. Our results suggest that, absent foundational advances in interpretability methods or models designed to be interpretable from the outset (Rudin, 2019), claims about what such models have "learned" should be treated cautiously. A model that has effectively memorized training data can still be useful, but understanding its limitations requires knowing whether it has captured governing relationships or merely statistical regularities. Building trust in foundation models for science may require starting from simpler, more interpretable architectures and scaling up systematically (MacMillan & Ouellette, 2025), rather than reverse-engineering billion-parameter transformers post hoc.

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

# A APPENDIX

## A.1 TRAINING

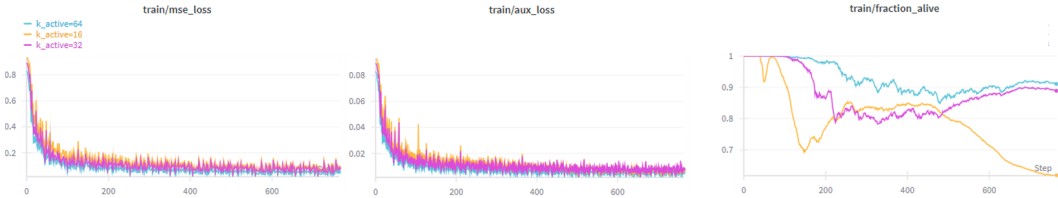

Figure 6: MSE loss (left), Aux loss (center), and alive fraction (right) from our training runs for $k=\{16,32,64\}$.

## A.2 ENSTROPHY DISTRIBUTIONS

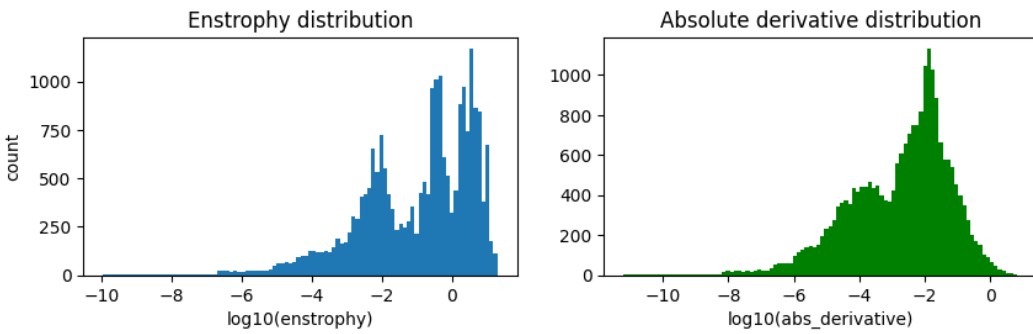

Figure 7: Enstrophy distributions per time-step and trajectory. We order the simulations by the mean absolute enstrophy derivative and choose the simulation with the largest value to be our reference simulation 50.

## A.3 DISTRIBUTIONS OF $\rho$

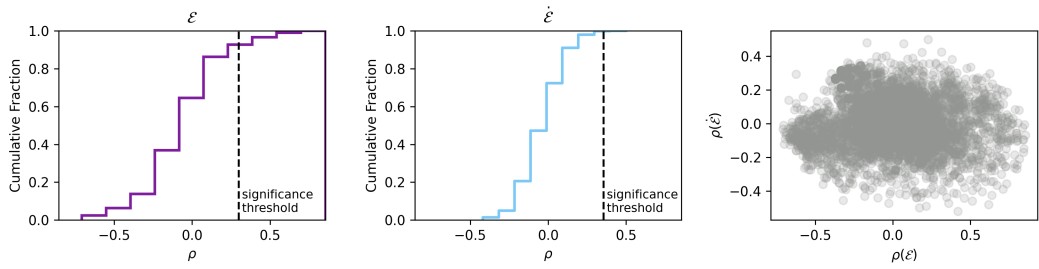

Figure 8: Distribution of Spearman's rank coefficient, $\rho$, between $\mathcal{E}(\text{Traj}_{\text{ref}})$ and all (non-zero) SAE features. Vertical black line is the significance threshold calculated from a permutation test (see section 3.1. 6% of the SAE features have $\rho$ exceeding this value.

## A.4 TOP ENSTROPHY FEATURES: TRAJECTORY 3

## Feature 12454

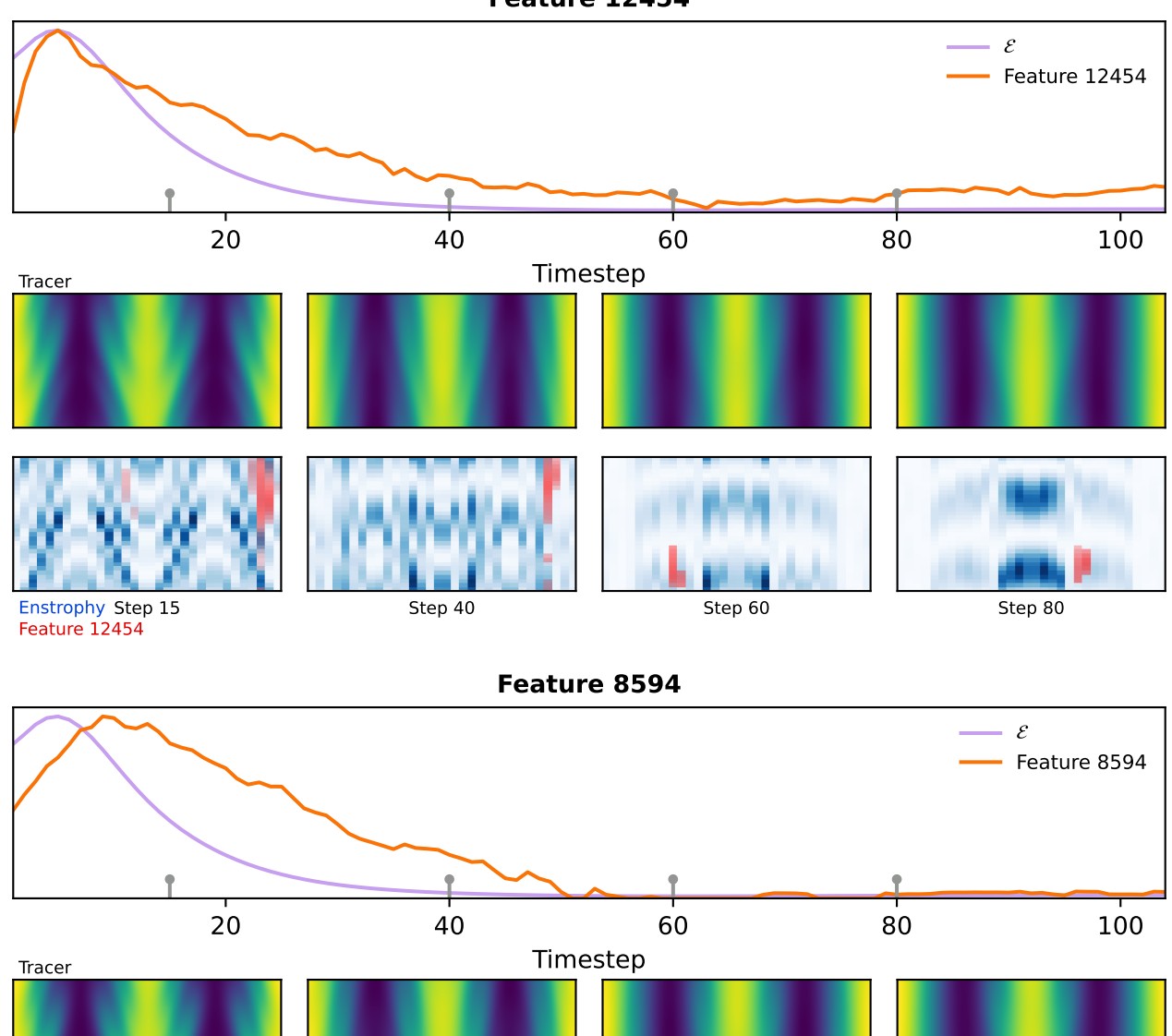

Tracer

Enstrophy Step 15
Feature 12454

Step 40

Step 60

Step 80

## Feature 8594

Tracer

Enstrophy Step 15
Feature 8594

Step 40

Step 60

Step 80

**Feature 12258**

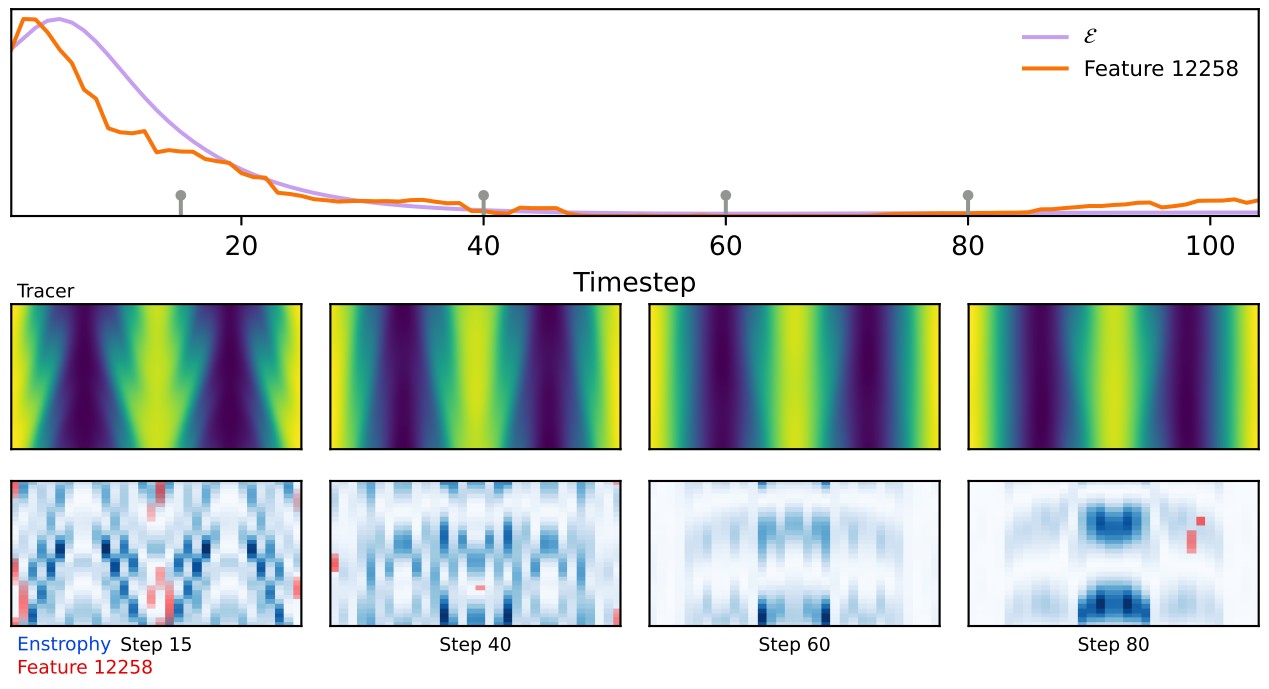

**Feature 11246**

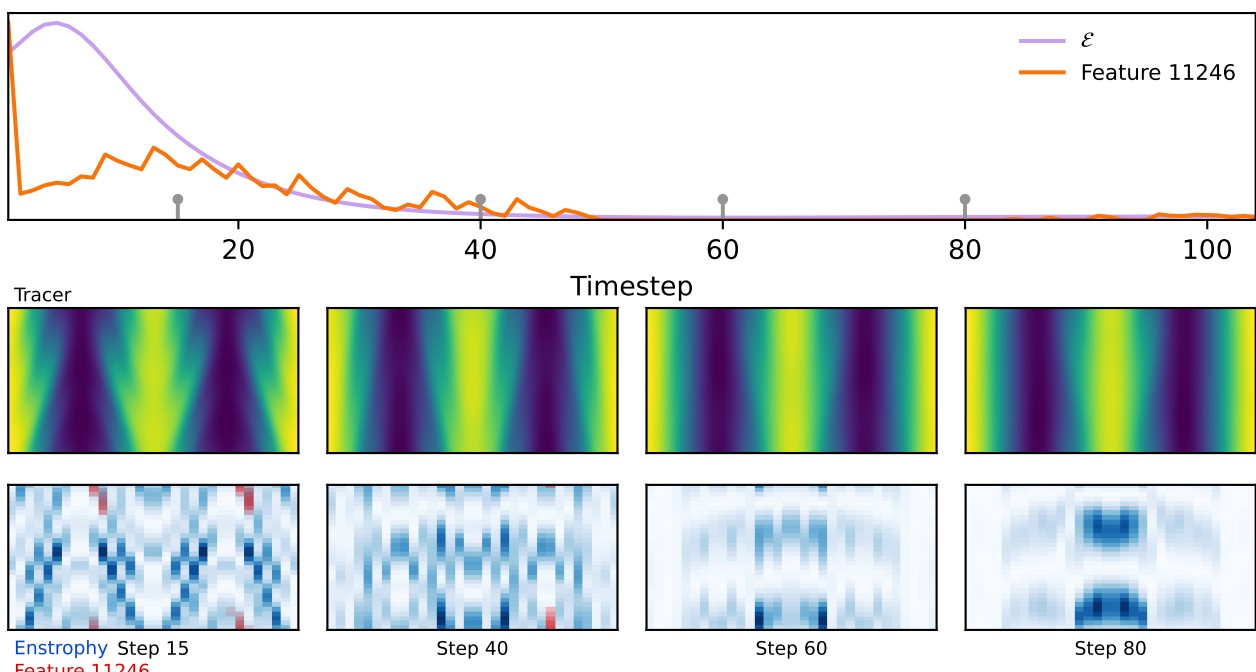

**Feature 19171**

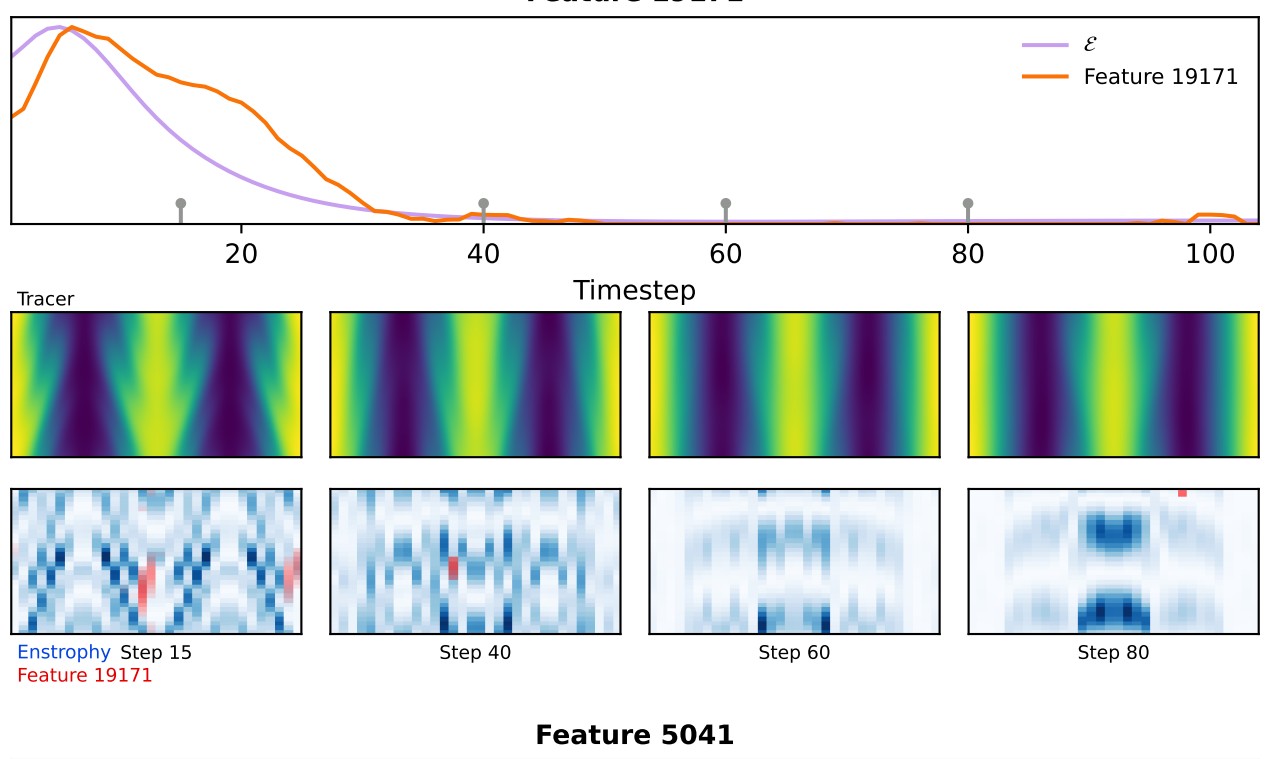

**Feature 5041**

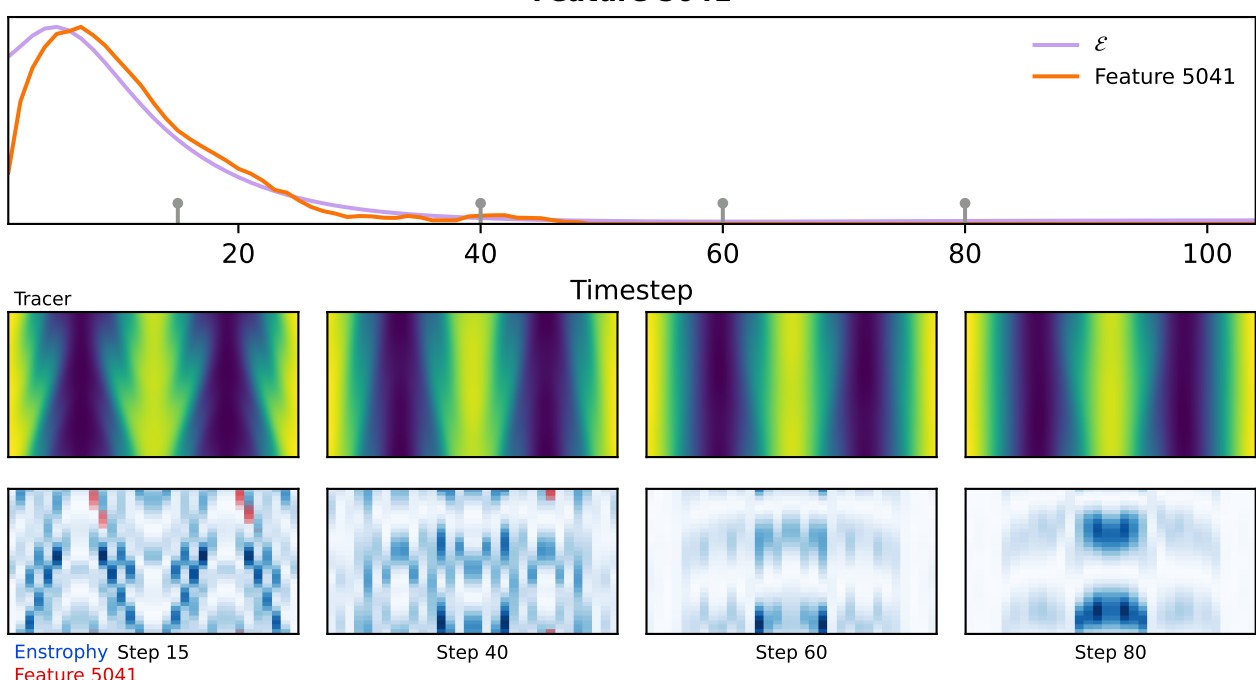

## Feature 4260

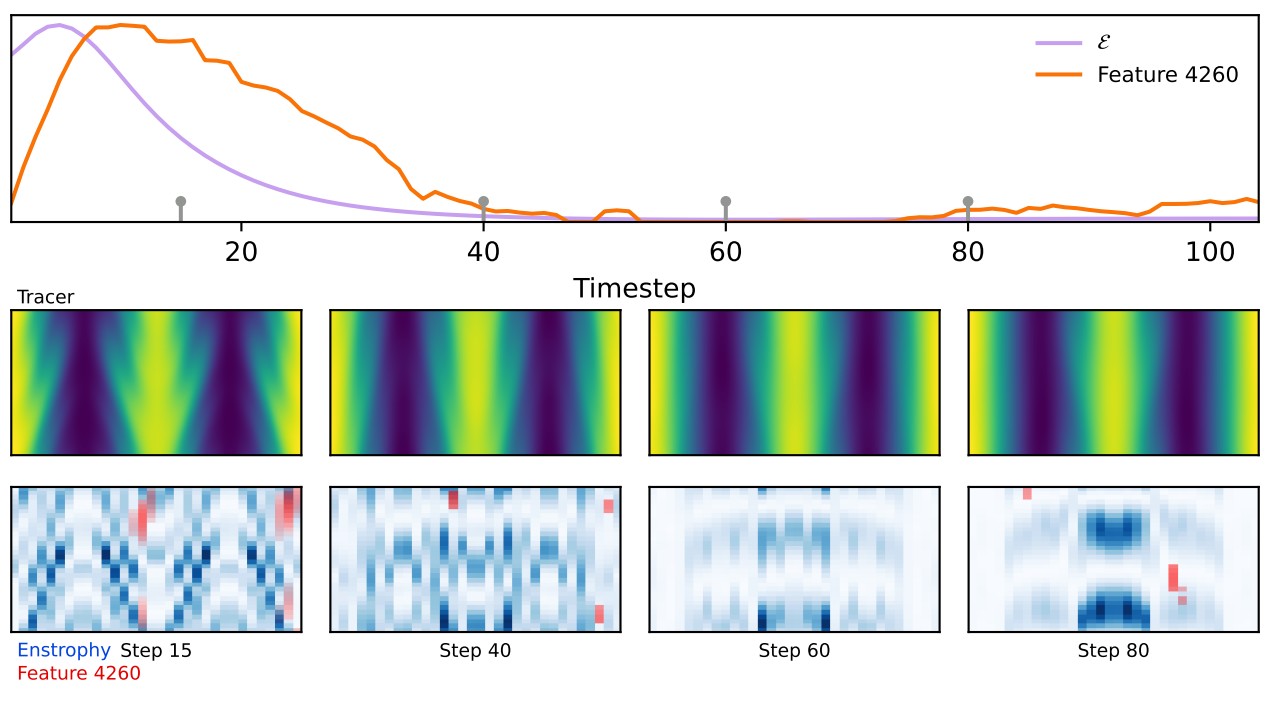

Tracer

Enstrophy Step 15
Feature 4260

Step 40

Step 60

Step 80

## Feature 8952

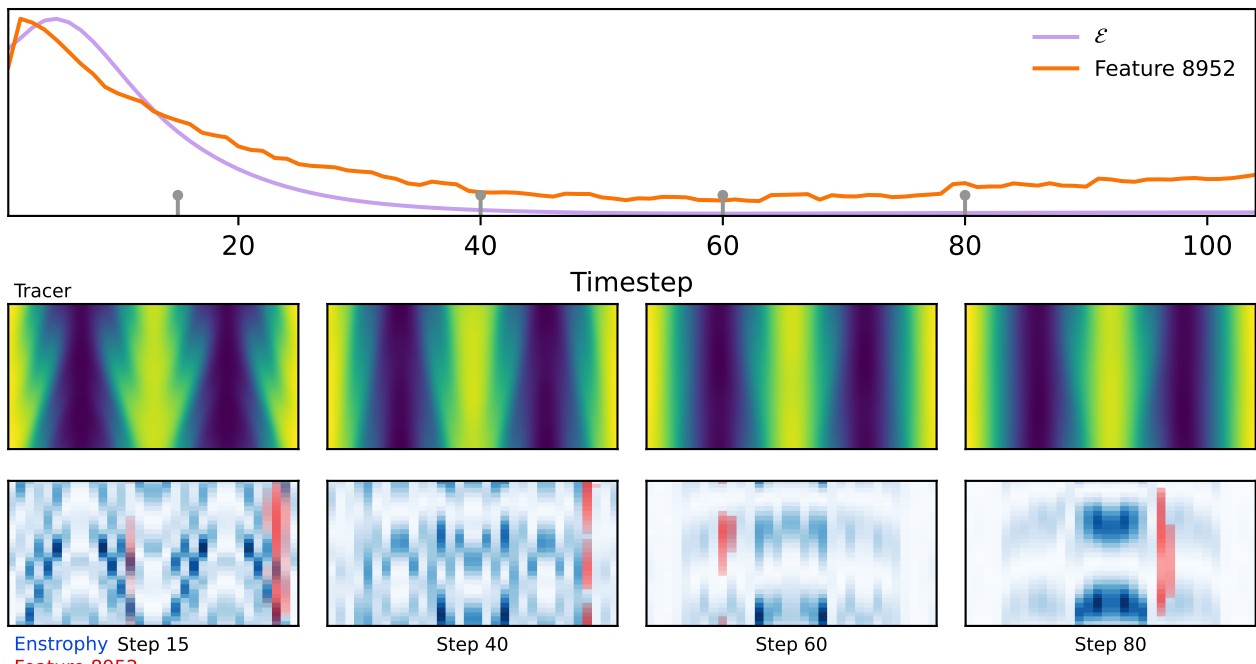

Tracer

Enstrophy Step 15
Feature 8952

Step 40

Step 60

Step 80

## Feature 11624

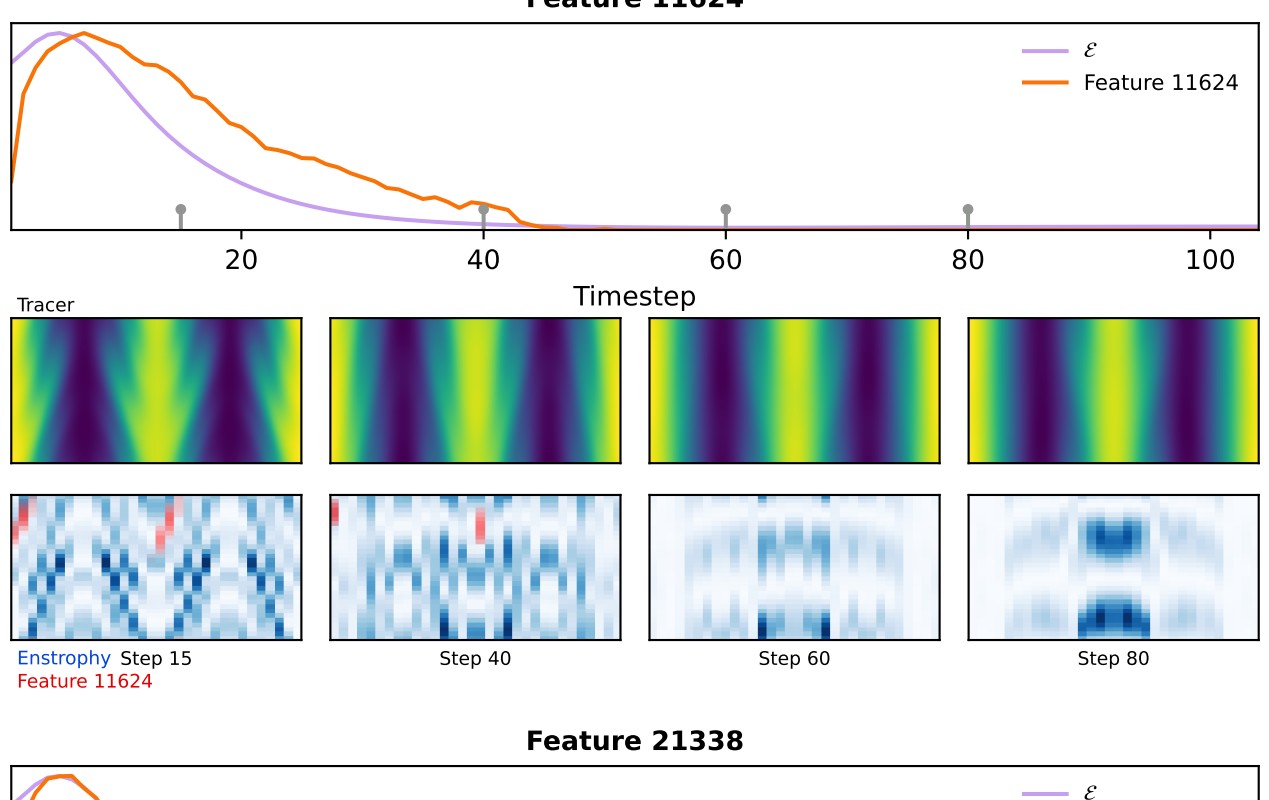

Tracer

Enstrophy Step 15
Feature 11624

Step 40

Step 60

Step 80

## Feature 21338

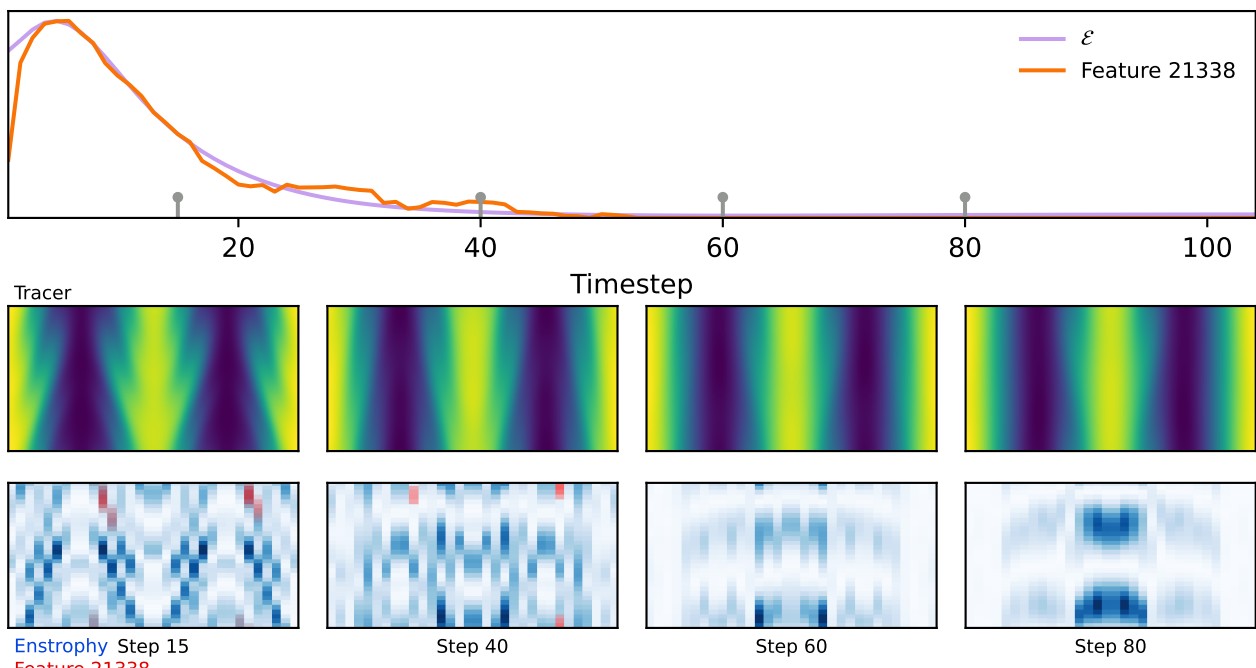

Tracer

Enstrophy Step 15
Feature 21338

Step 40

Step 60

Step 80

A.5    TOP ENSTROPHY FEATURES: TRAJECTORY 50

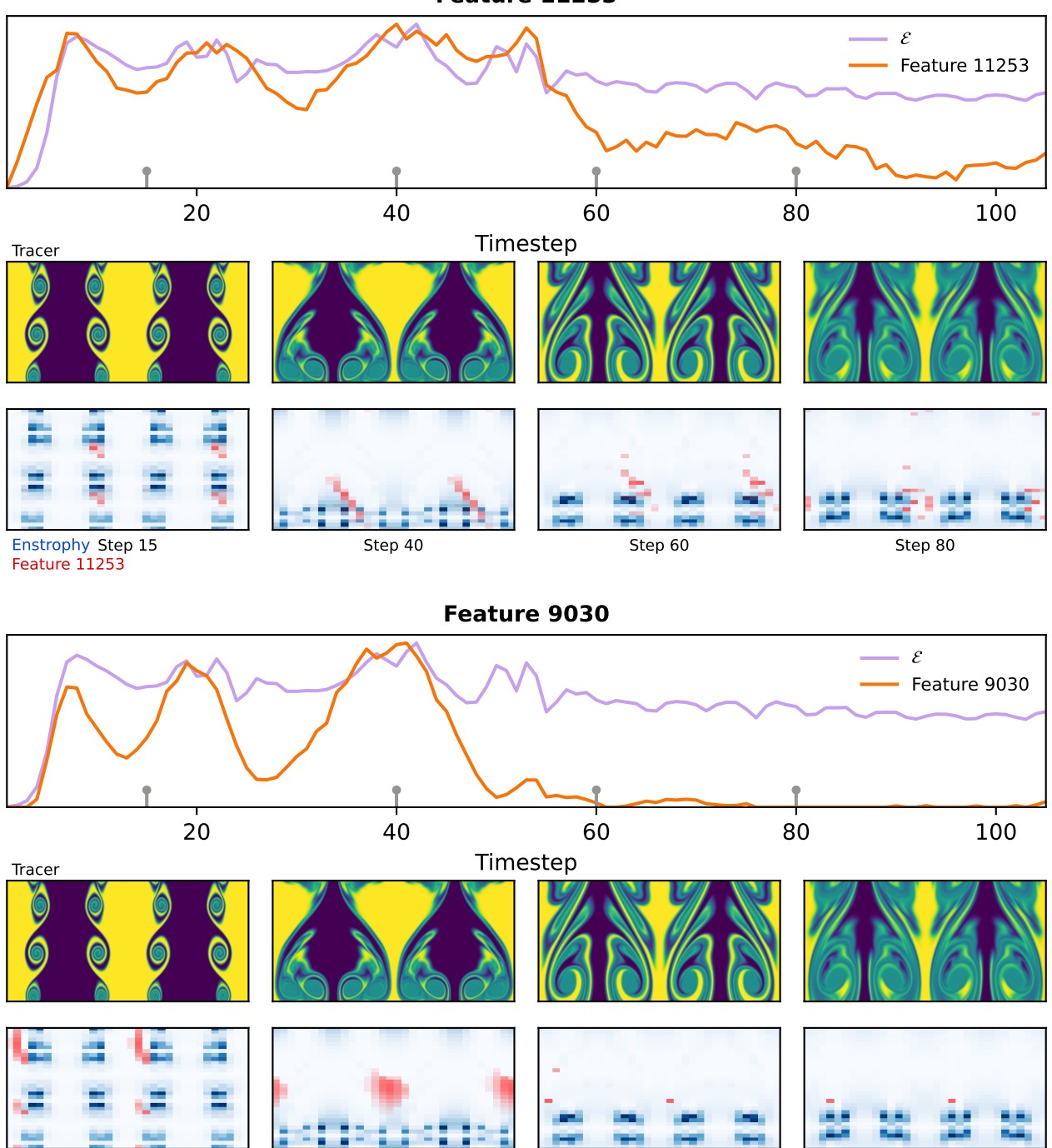

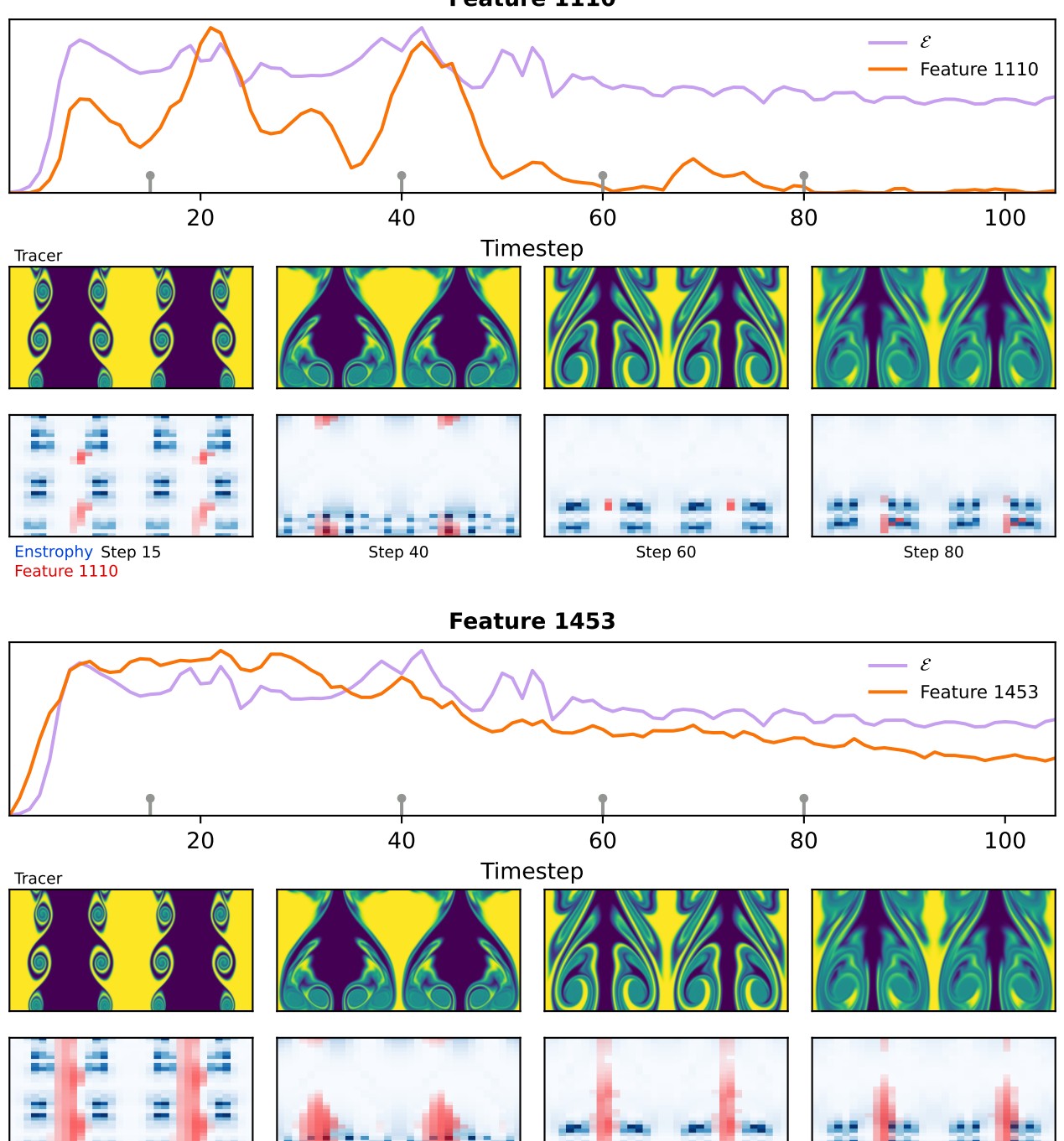

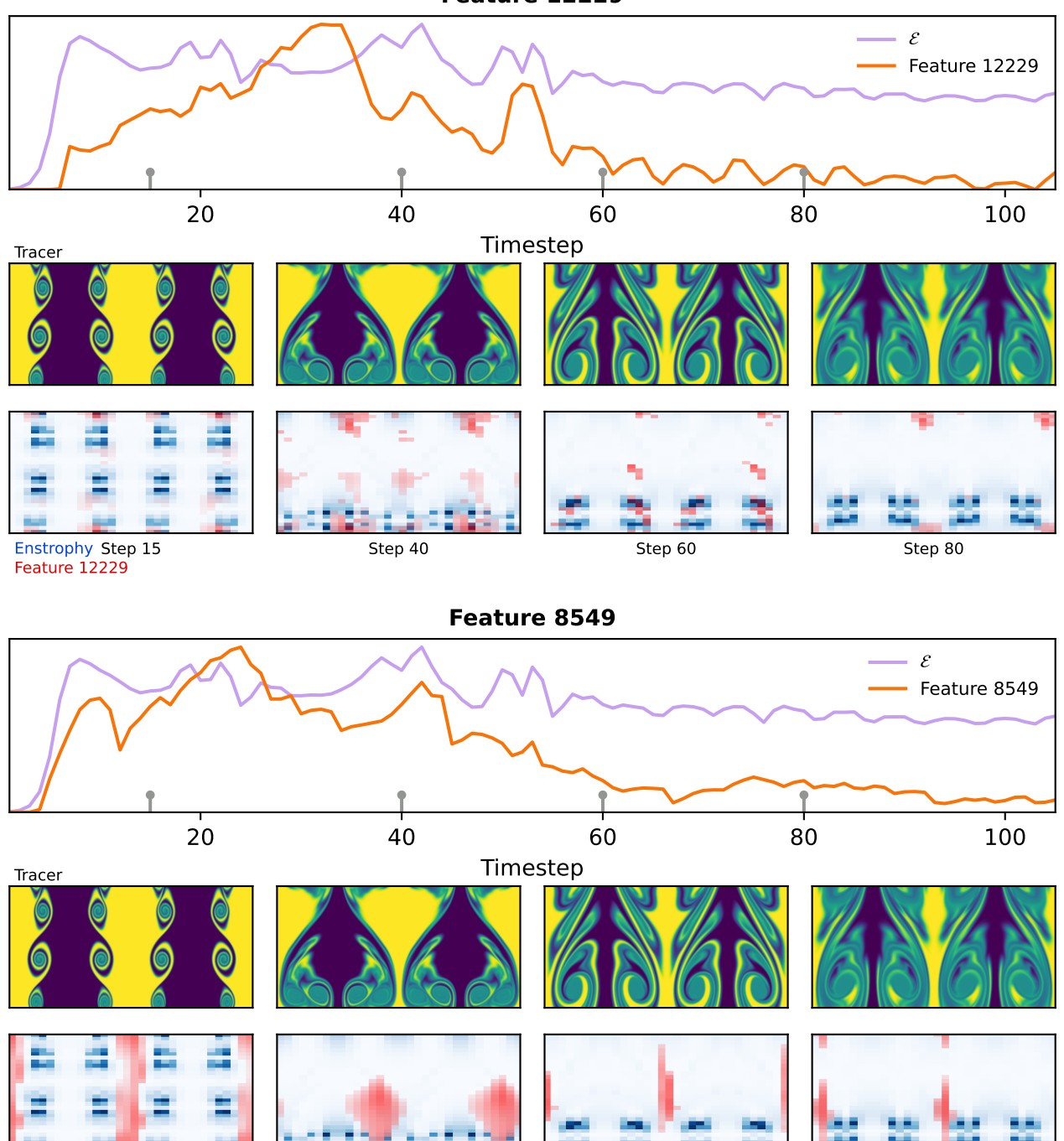

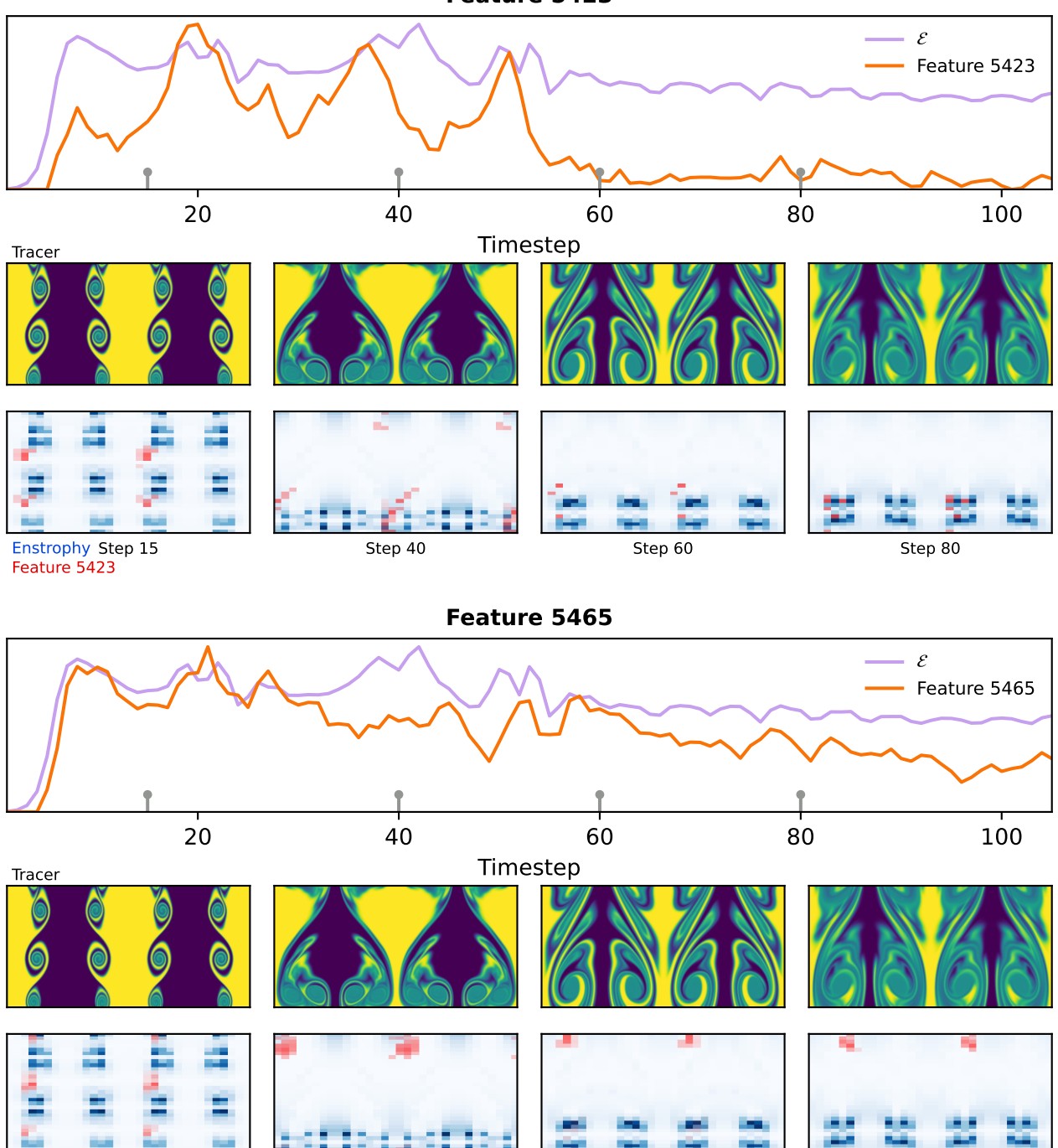

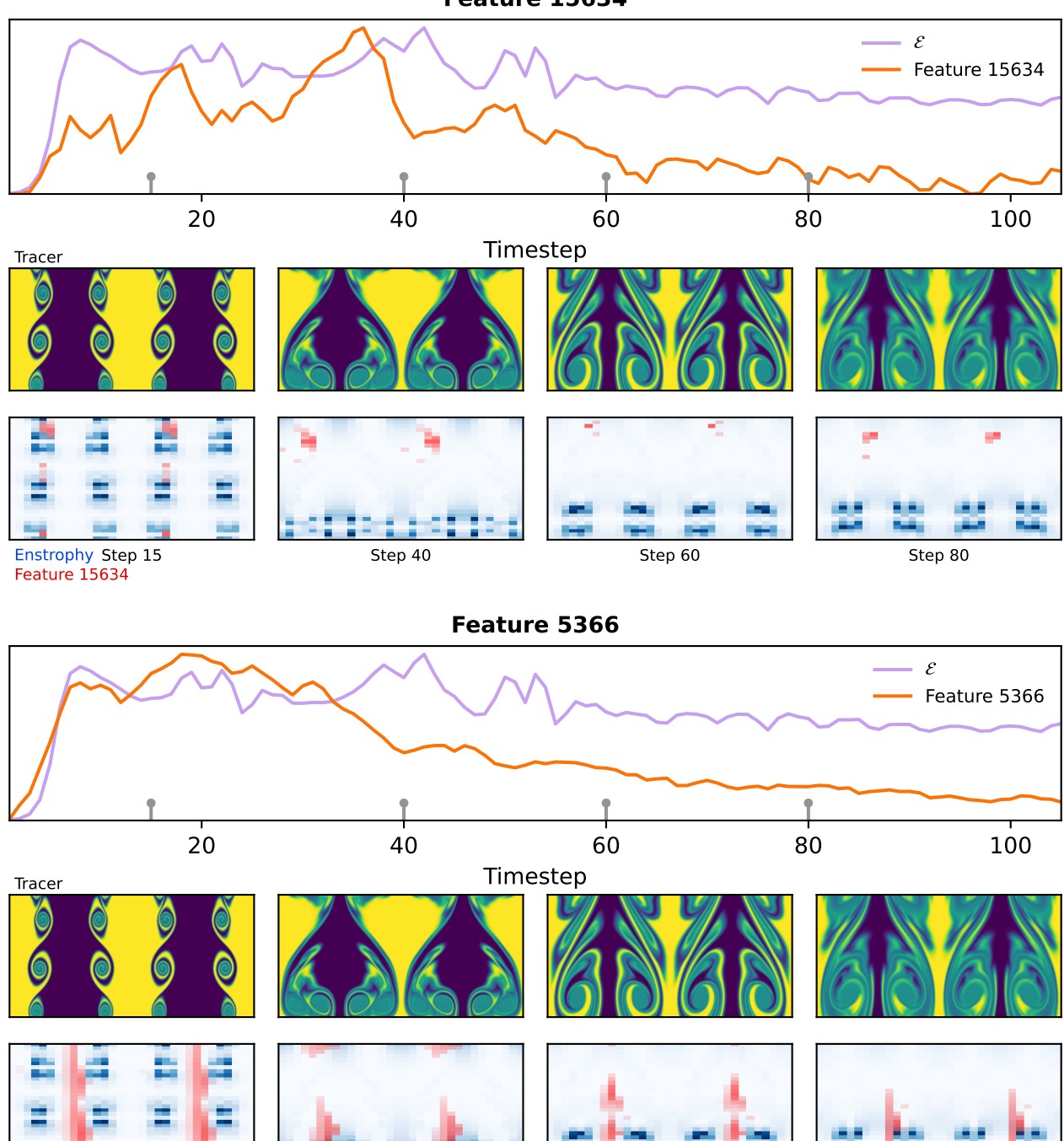

## A.6 TOP ENSTROPHY FEATURES: TRAJECTORY 56

## Feature 14304

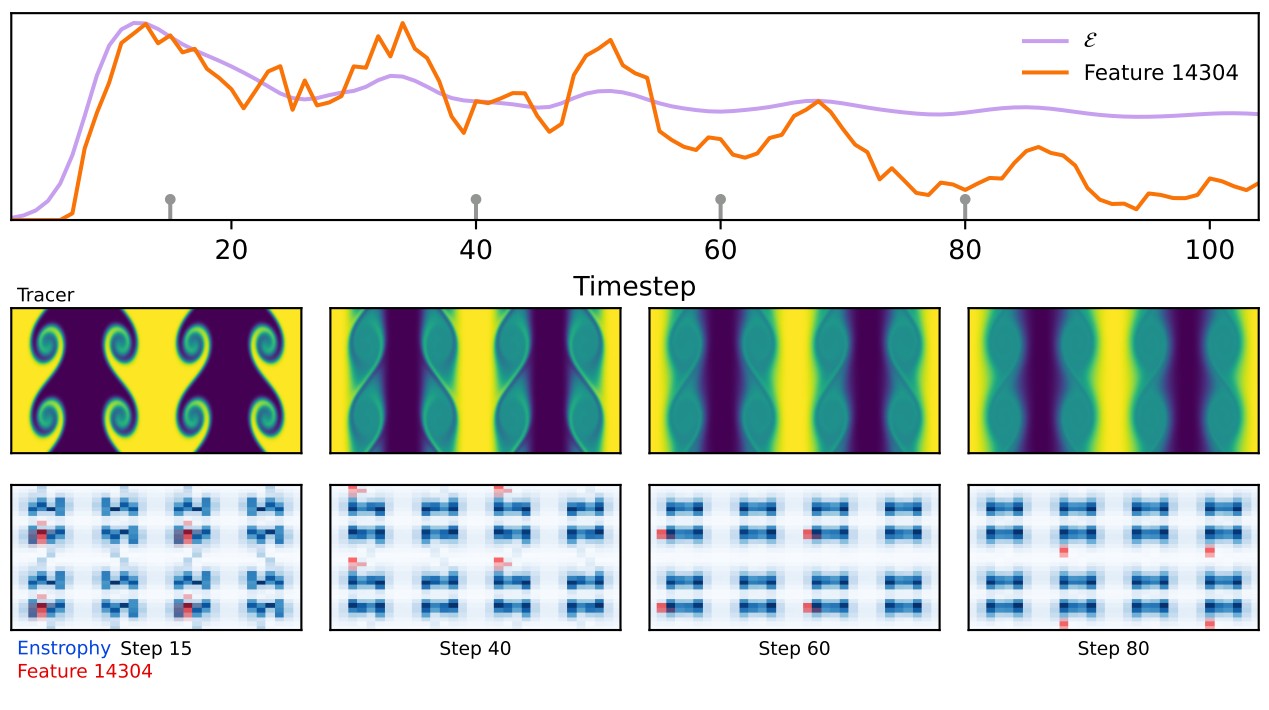

## Feature 9019

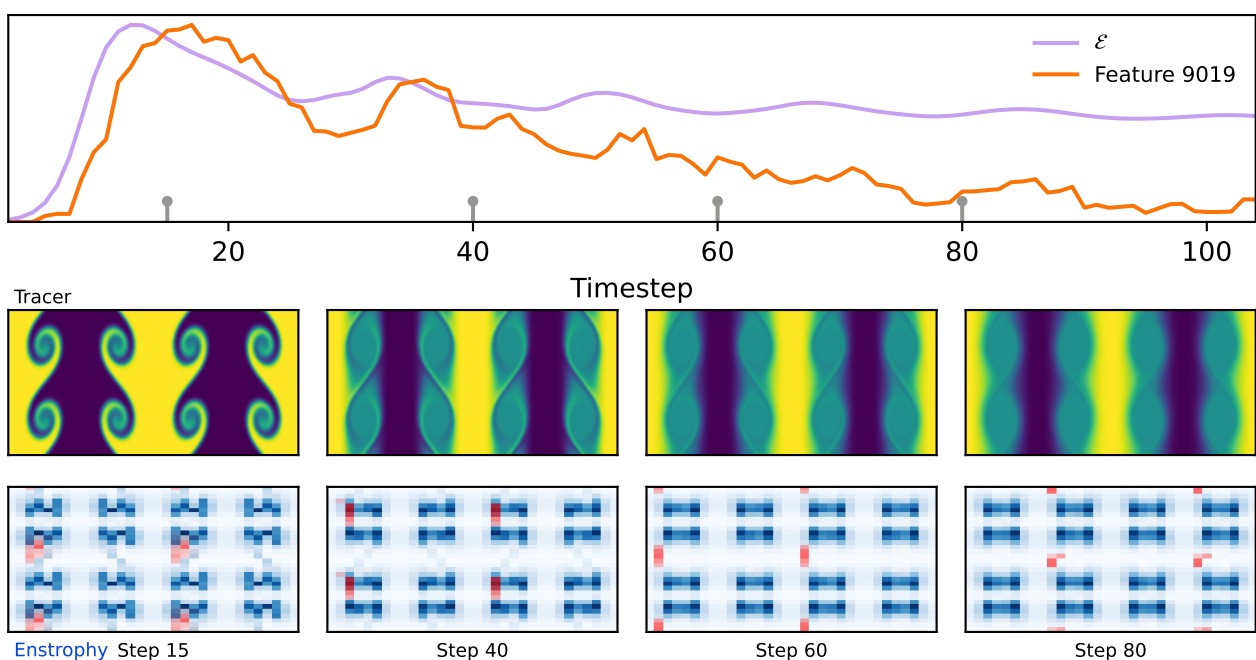

## Feature 10843

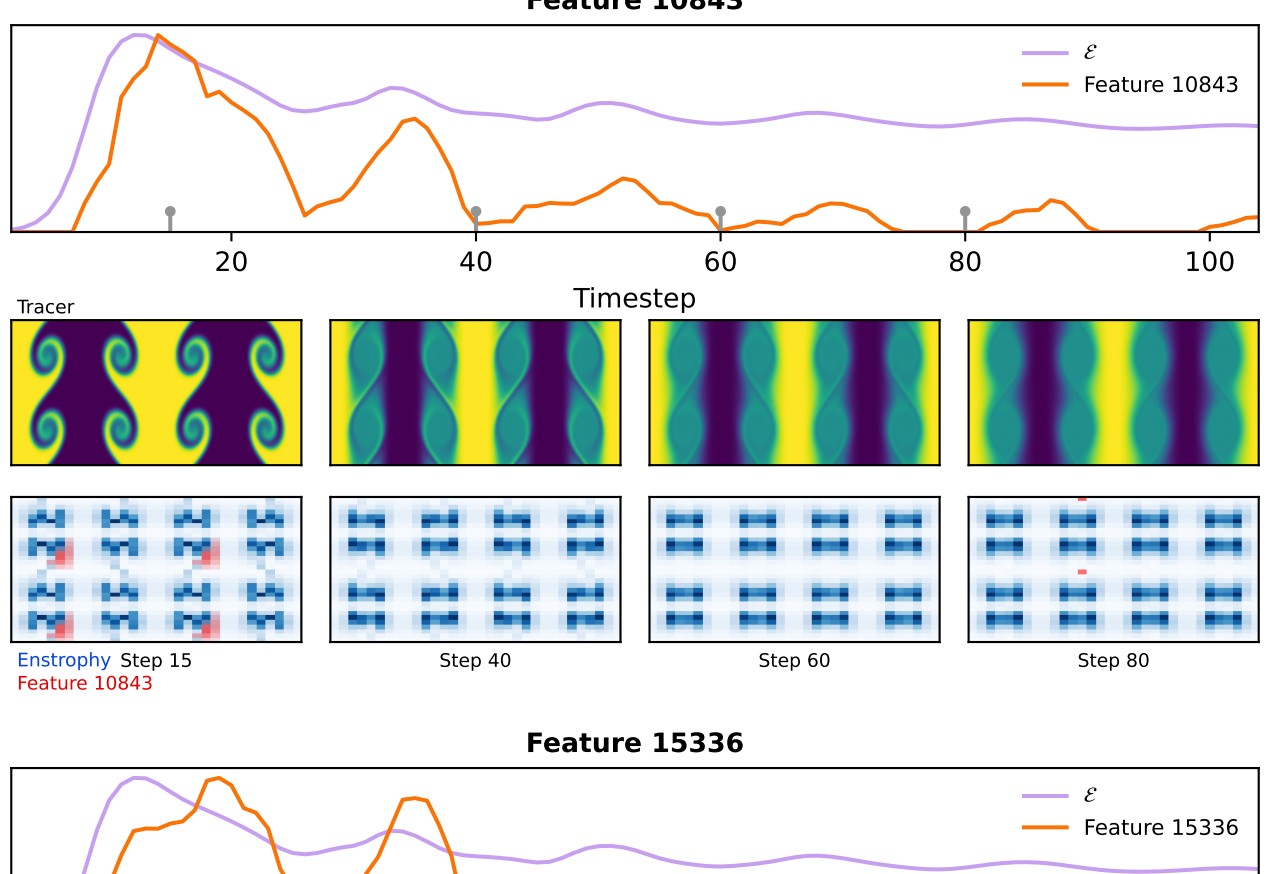

Tracer

Enstrophy Step 15
Feature 10843

Step 40

Step 60

Step 80

## Feature 15336

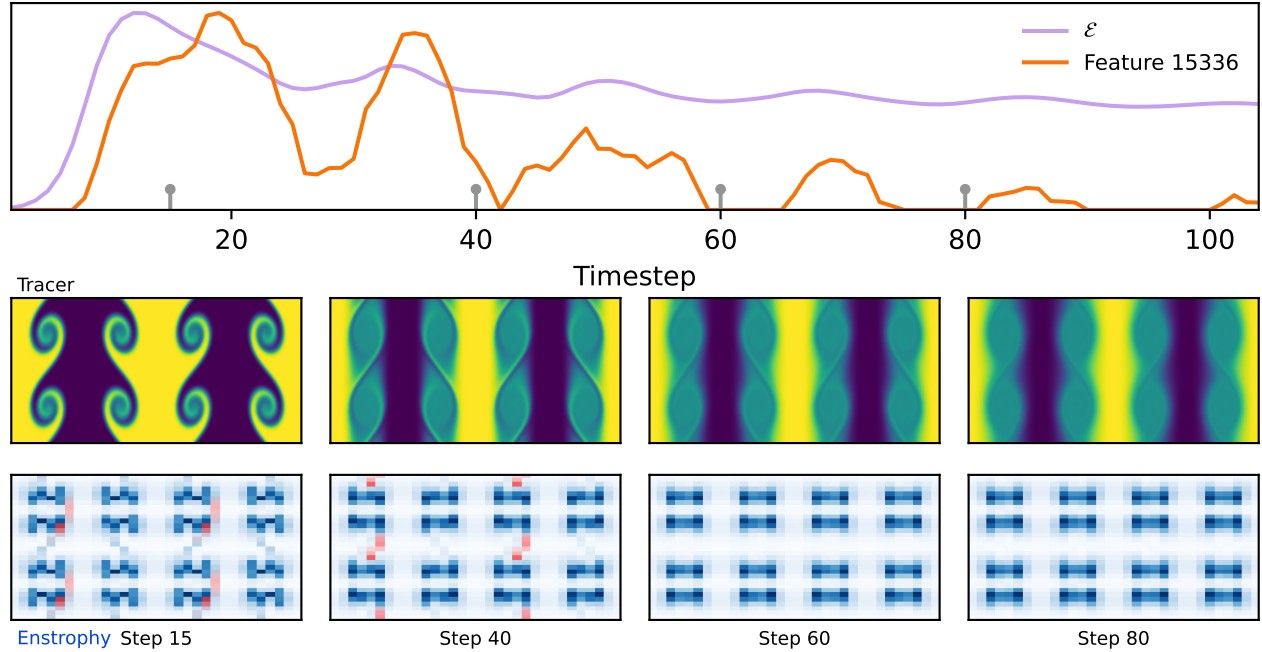

Tracer

Enstrophy Step 15
Feature 15336

Step 40

Step 60

Step 80

## Feature 6226

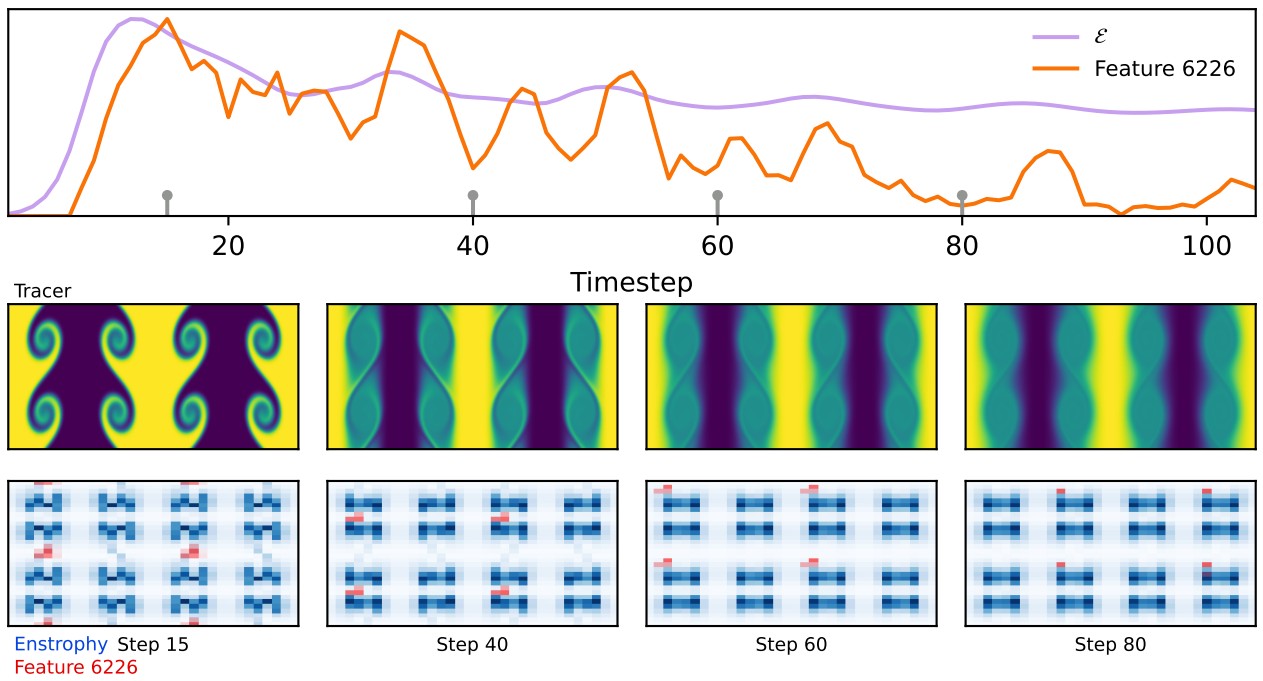

Tracer

Enstrophy Step 15
Feature 6226

Step 40      Step 60      Step 80

## Feature 446

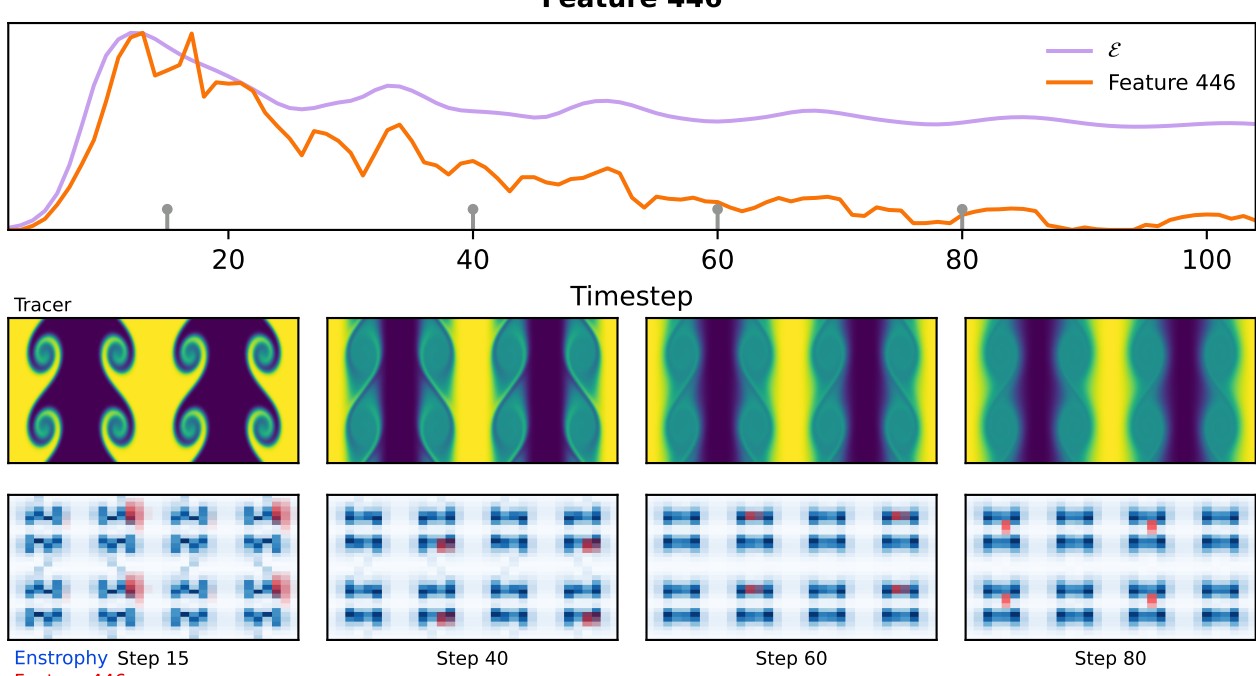

Tracer

Enstrophy Step 15
Feature 446

Step 40      Step 60      Step 80

# Feature 10246

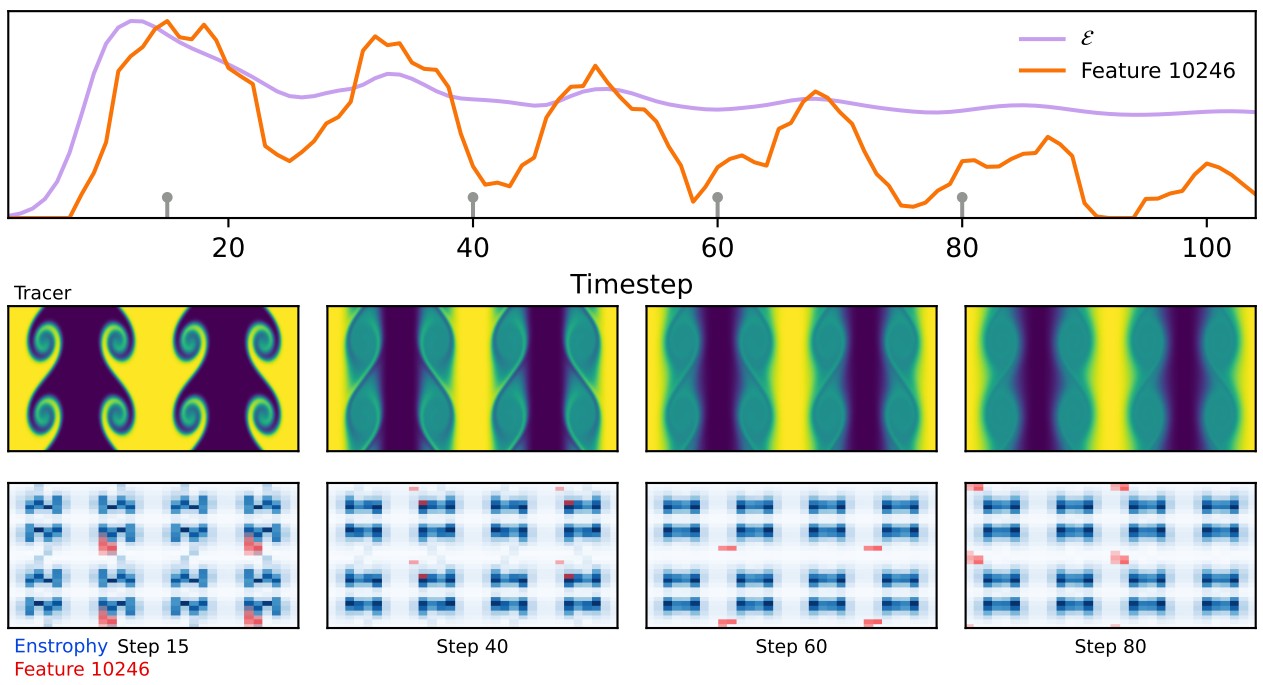

Tracer

Enstrophy Step 15
Feature 10246

Step 40   Step 60   Step 80

# Feature 9113

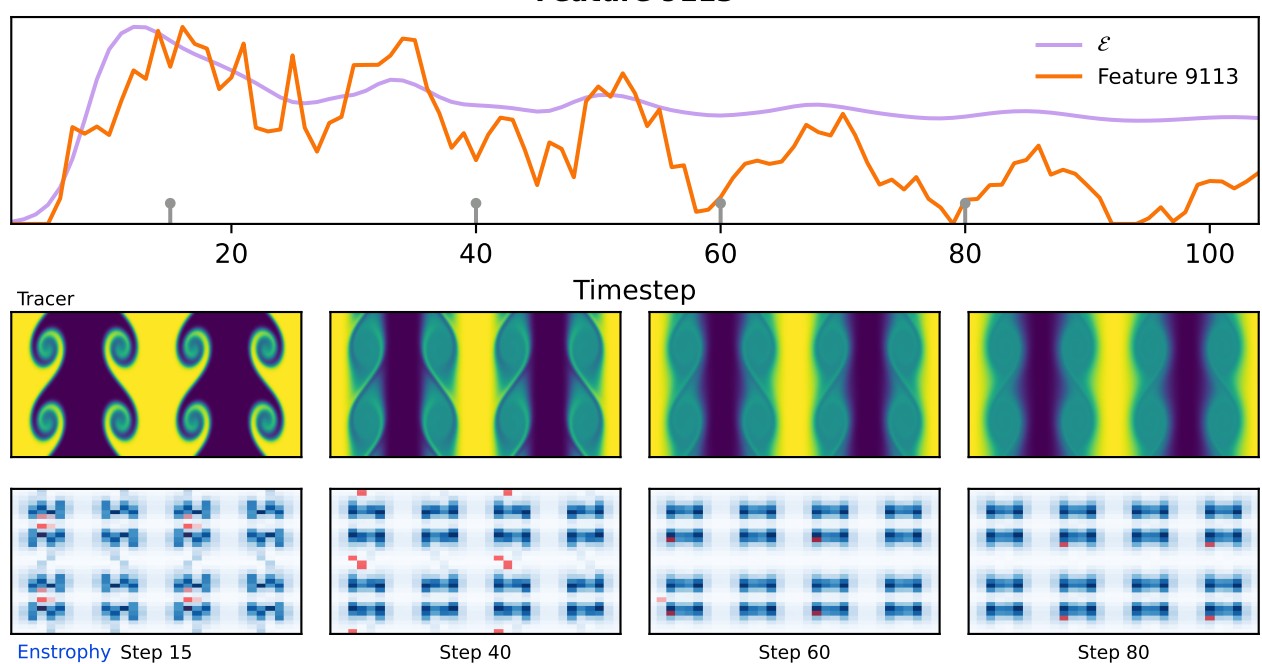

Tracer

Enstrophy Step 15
Feature 9113

Step 40   Step 60   Step 80

## Feature 13866

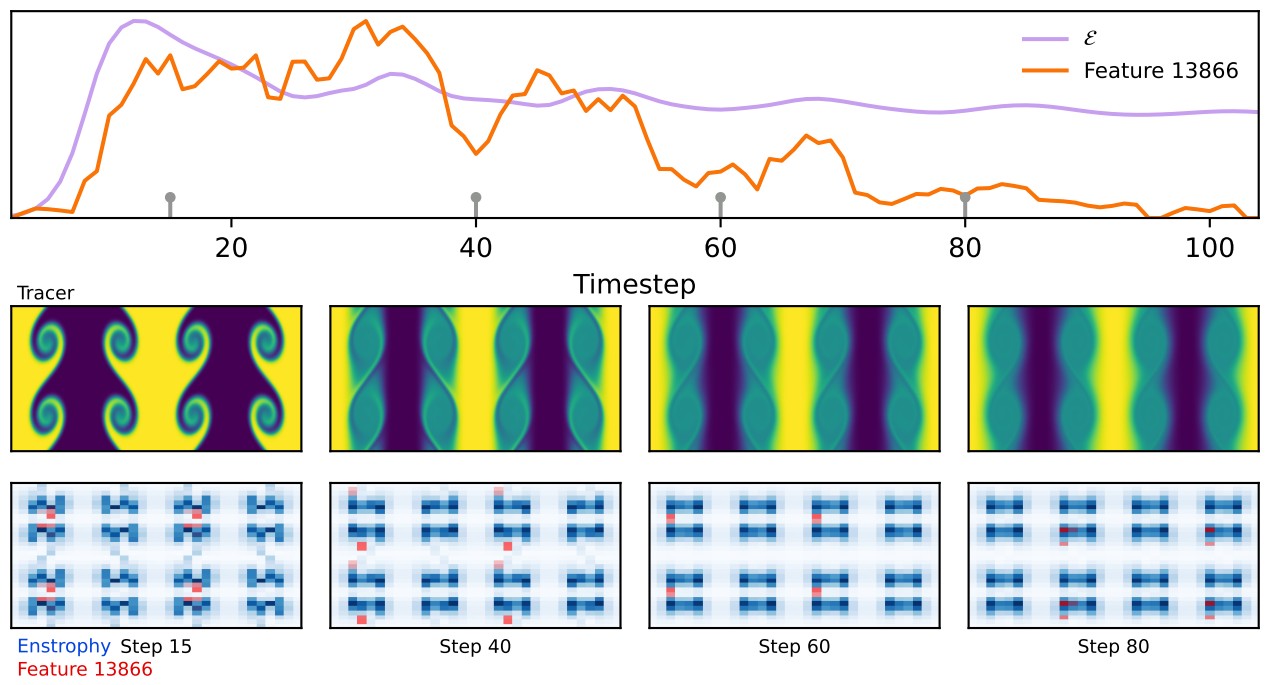

Tracer

Enstrophy Step 15
Feature 13866

Step 40          Step 60          Step 80

## Feature 9591

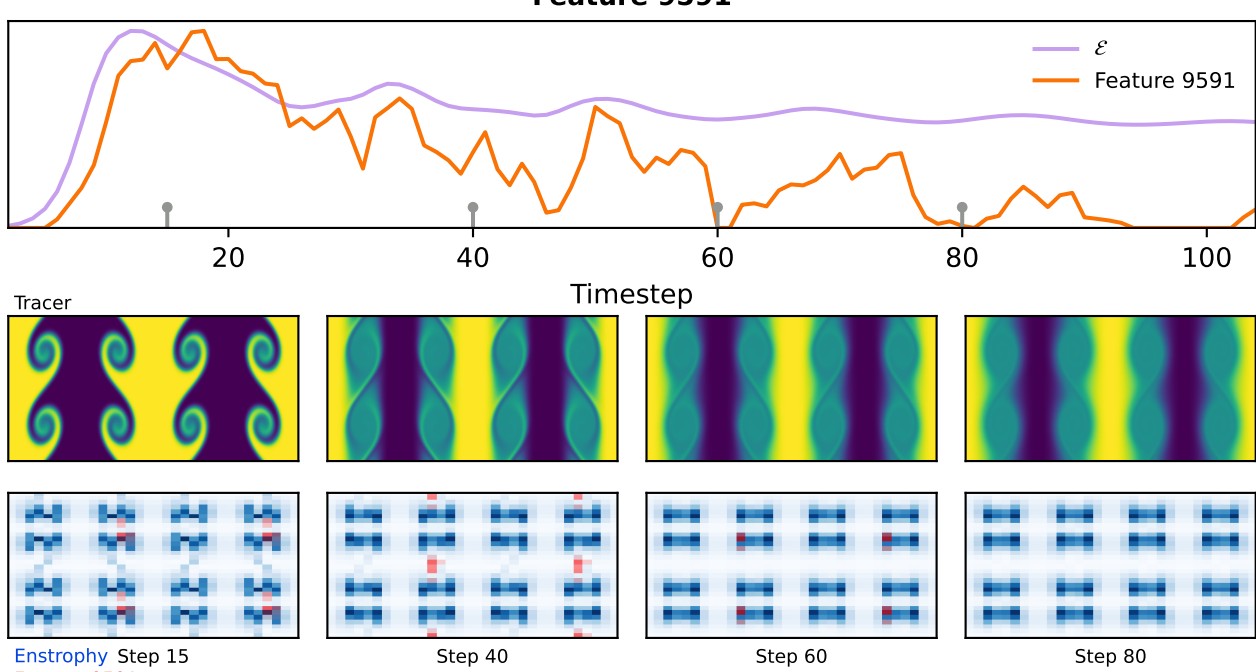

Tracer

Enstrophy Step 15
Feature 9591

Step 40          Step 60          Step 80

A.7 TOP ENSTROPHY FEATURES: TRAJECTORY 3 (TRAJECTORY 50 FEATURES)

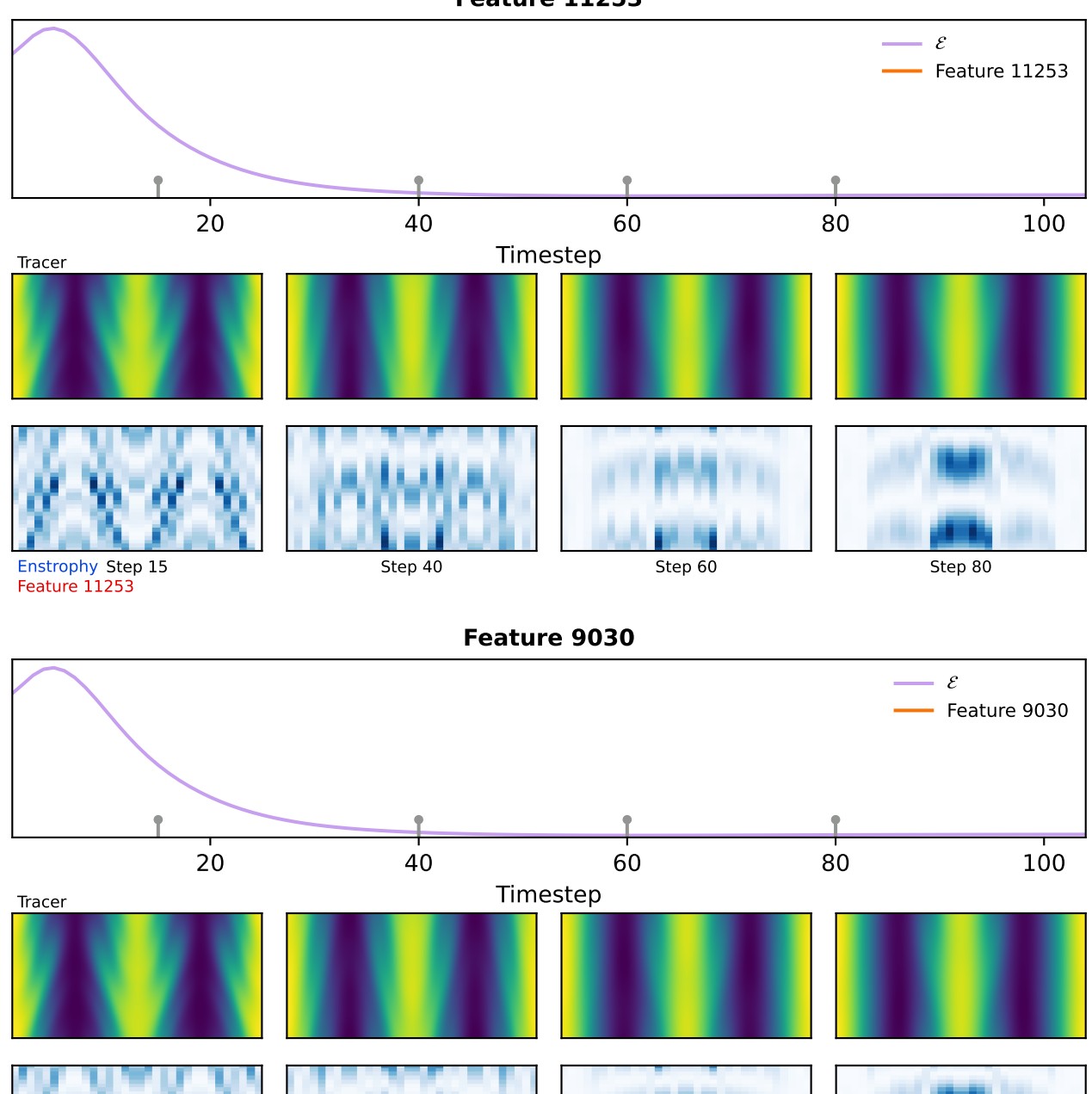

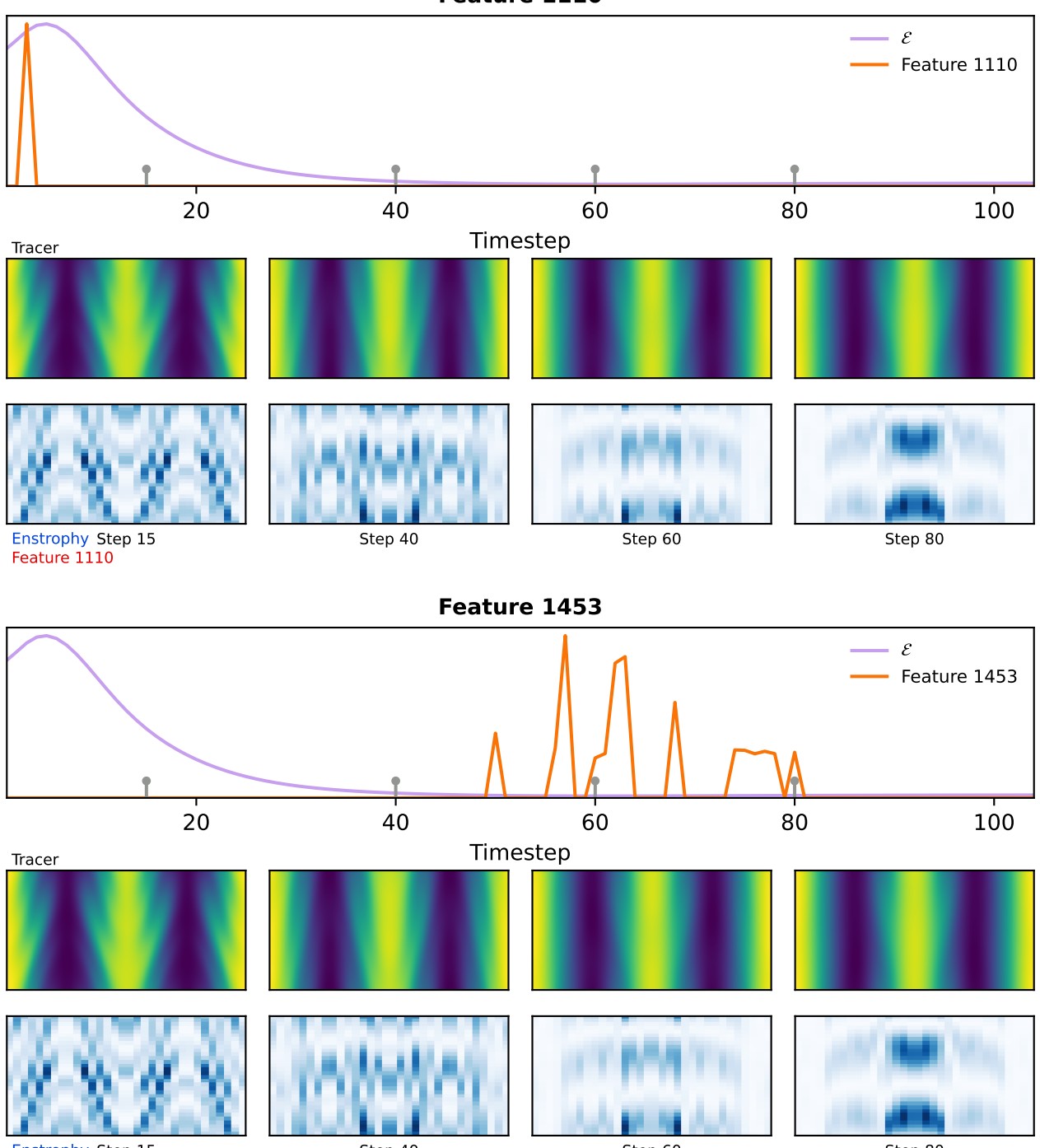

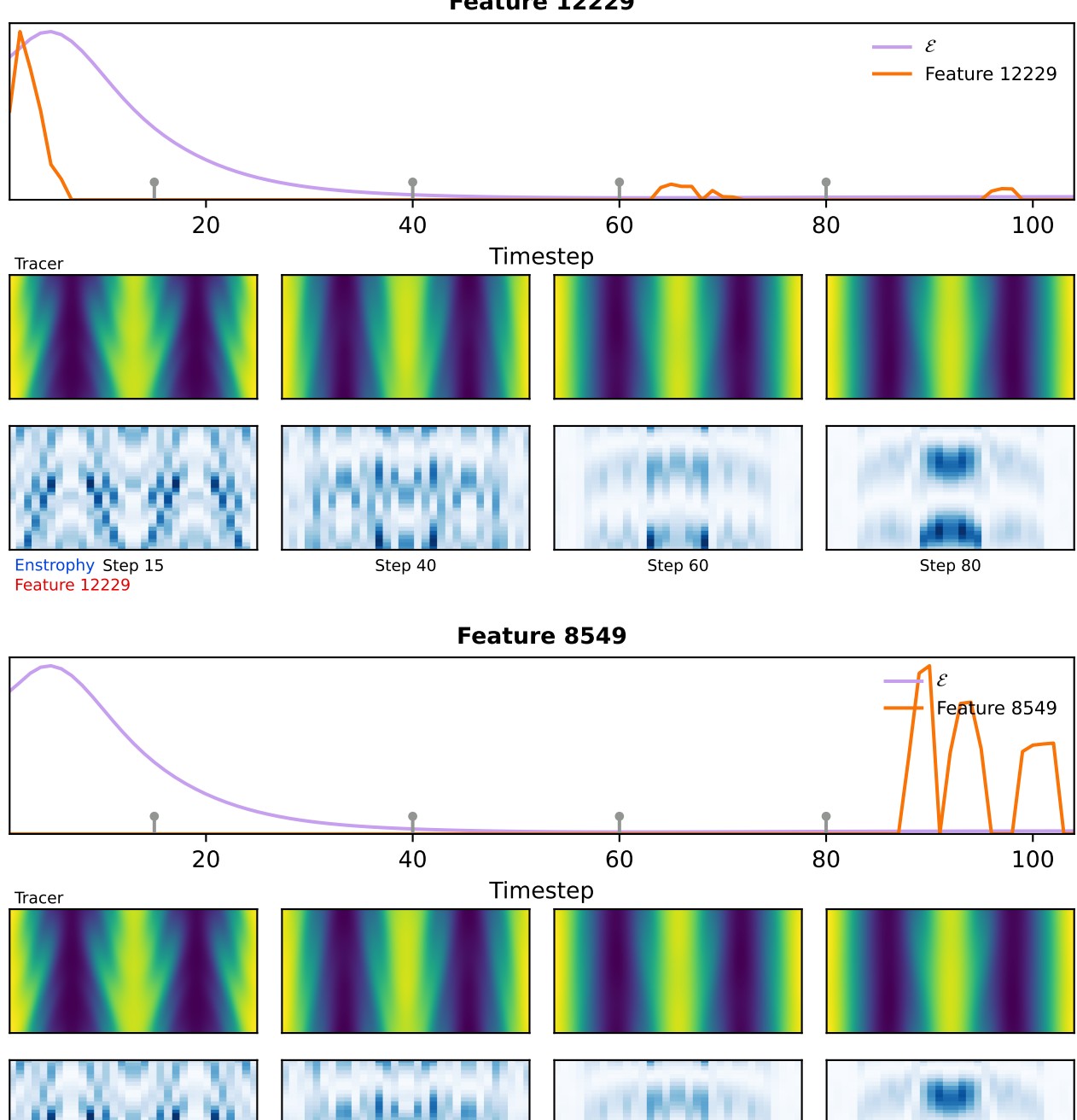

# Feature 5423

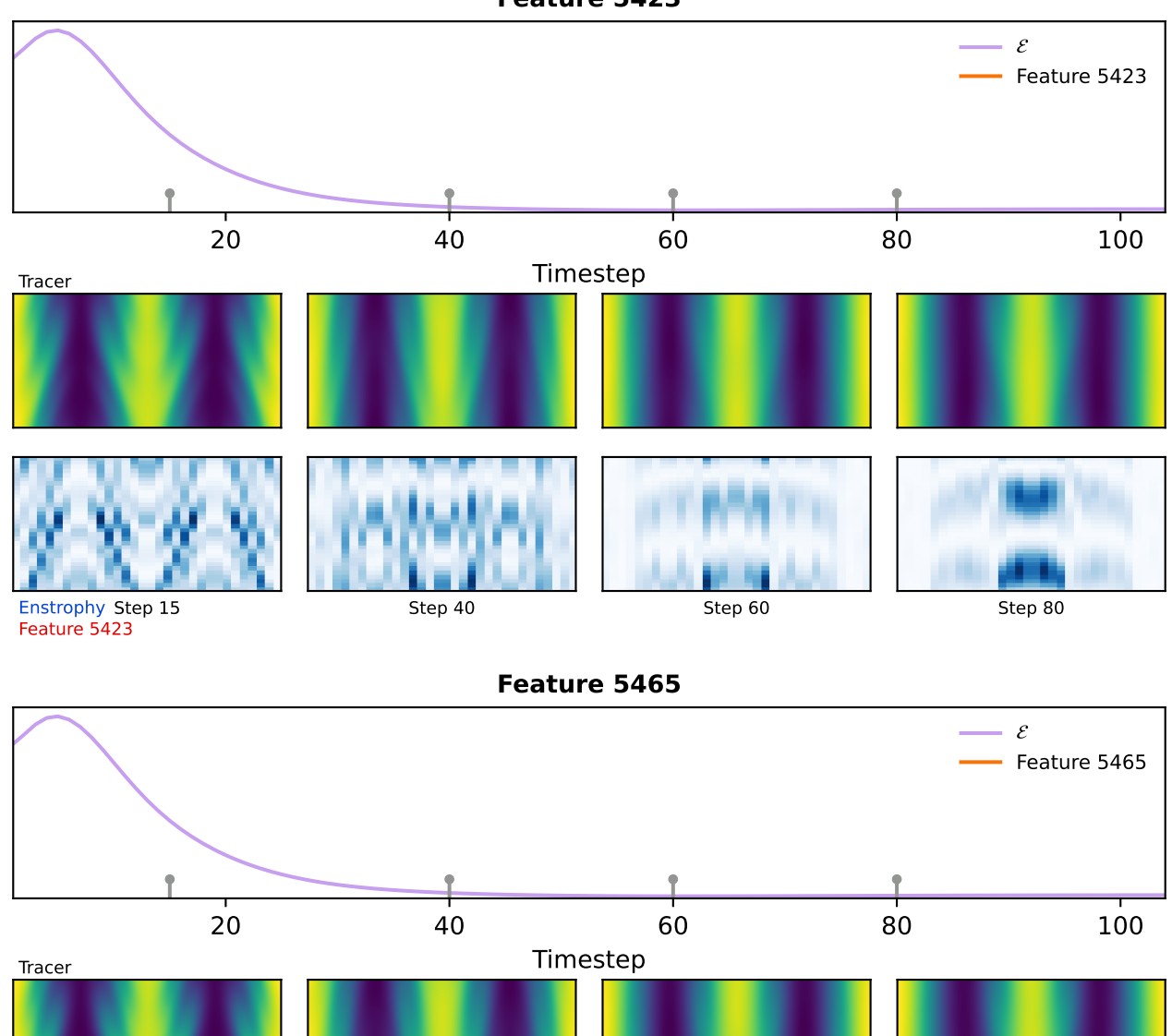

Tracer

Enstrophy Step 15
Feature 5423

Step 40

Step 60

Step 80

# Feature 5465

Tracer

Enstrophy Step 15
Feature 5465

Step 40

Step 60

Step 80

**Feature 15634**

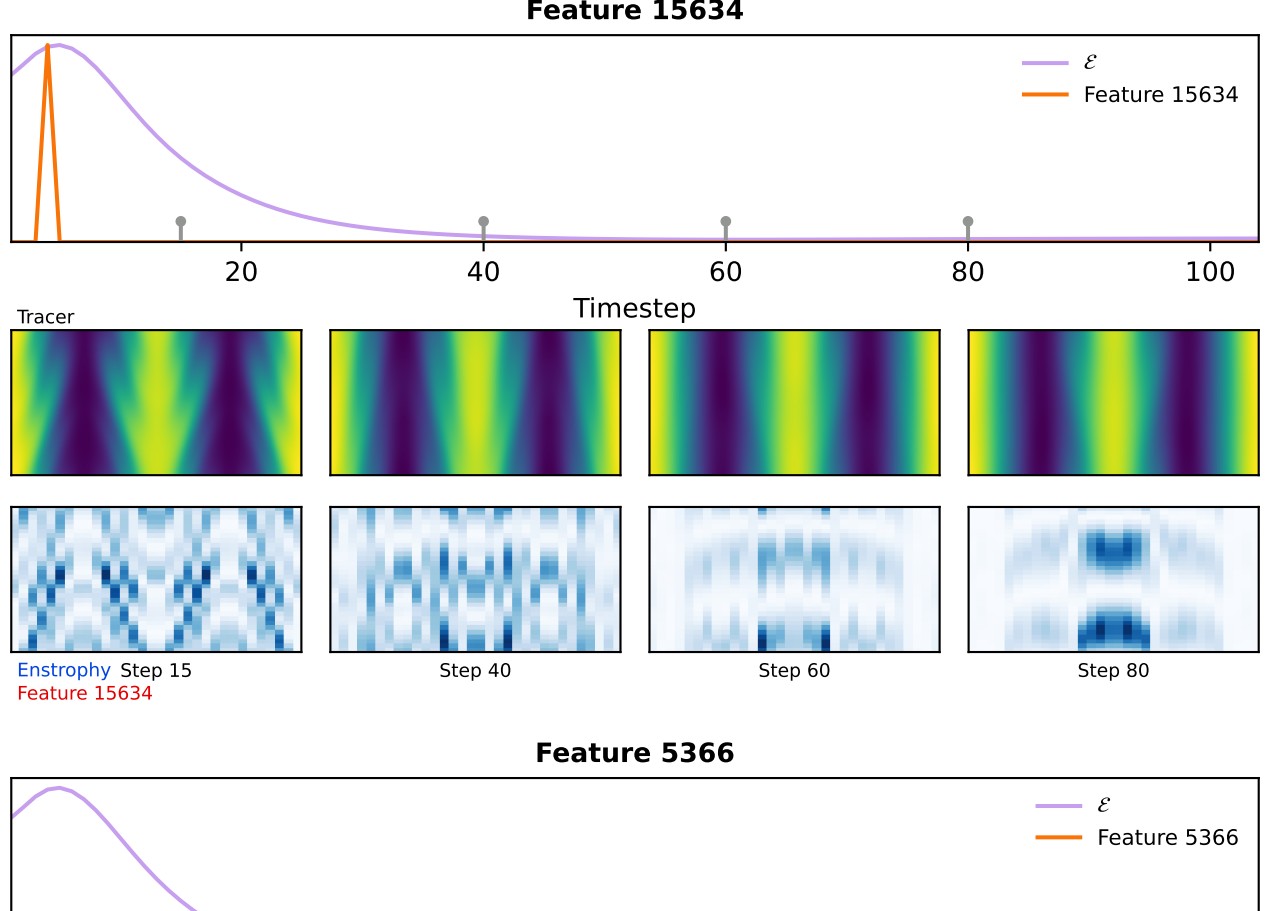

Tracer

Enstrophy Step 15
Feature 15634

Step 40

Step 60

Step 80

**Feature 5366**

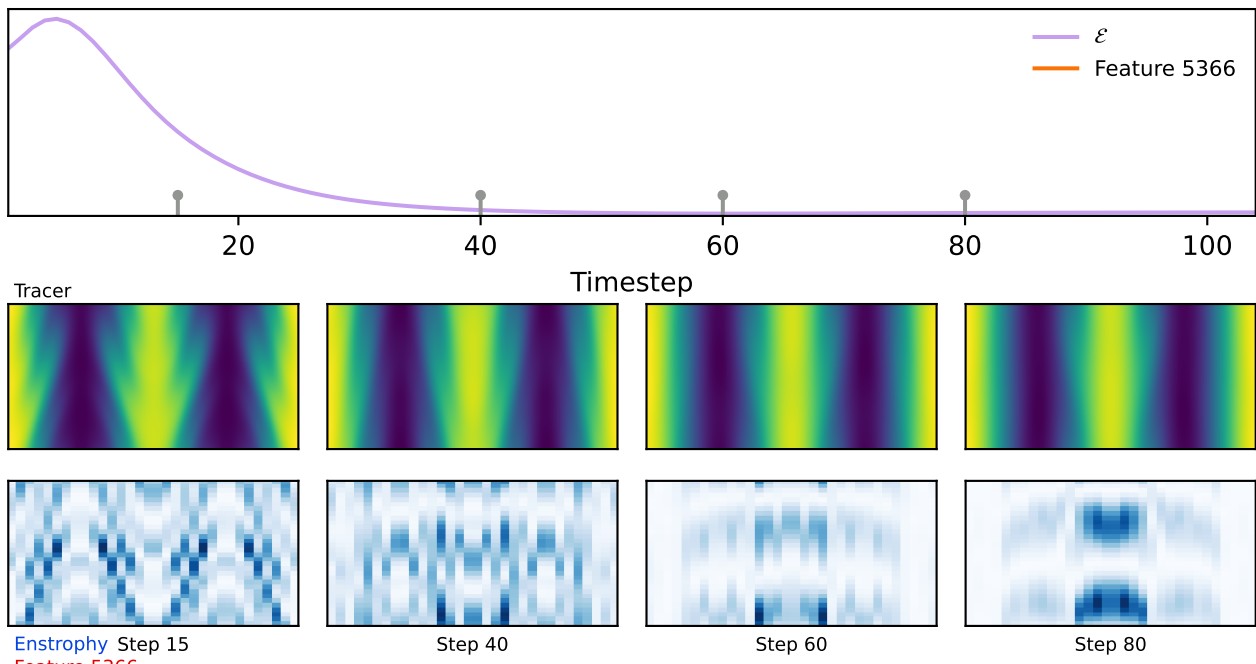

Tracer

Enstrophy Step 15
Feature 5366

Step 40

Step 60

Step 80

## A.8 TOP ENSTROPHY FEATURES: TRAJECTORY 56 (TRAJECTORY 50 FEATURES)

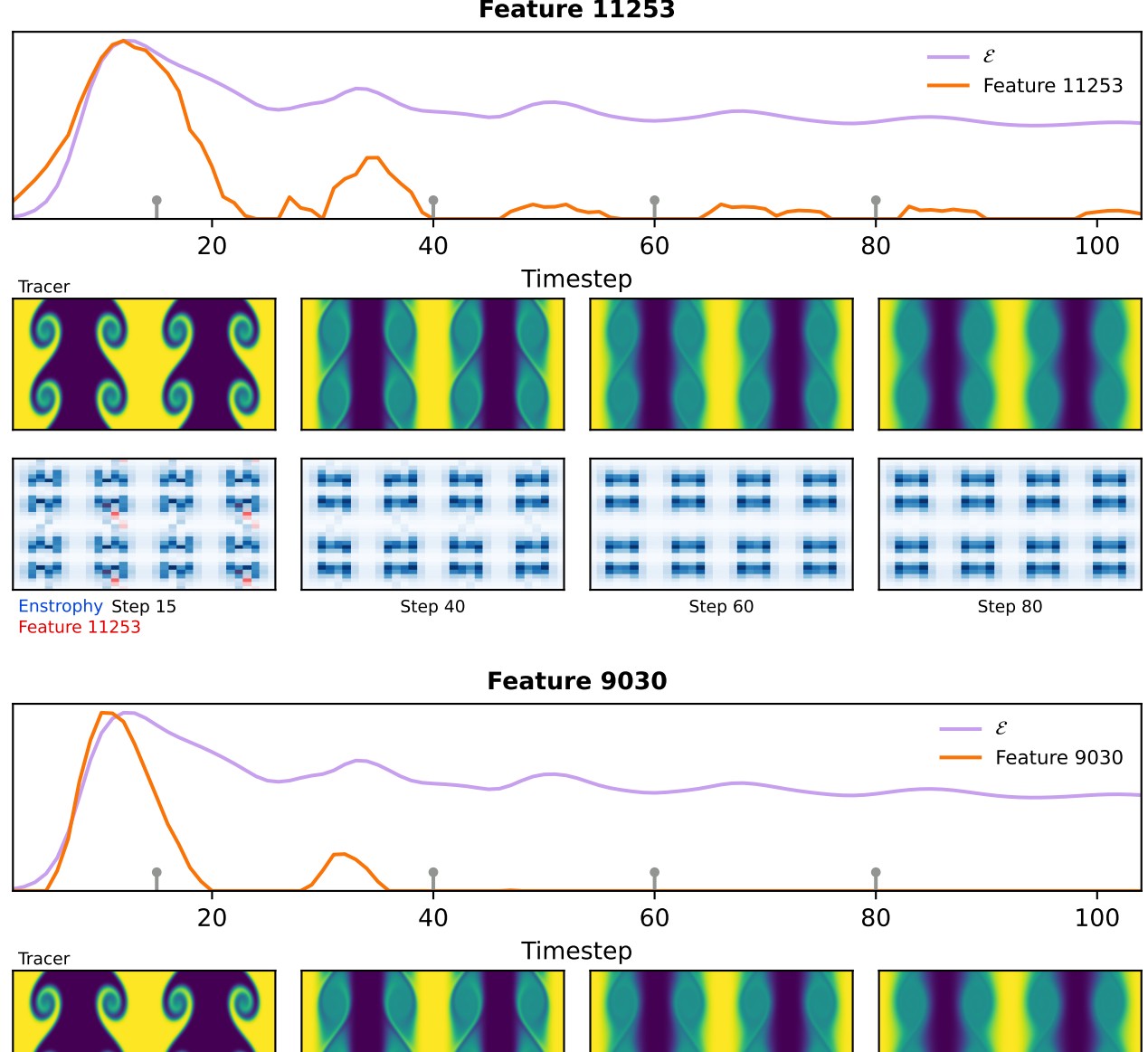

# Feature 1110

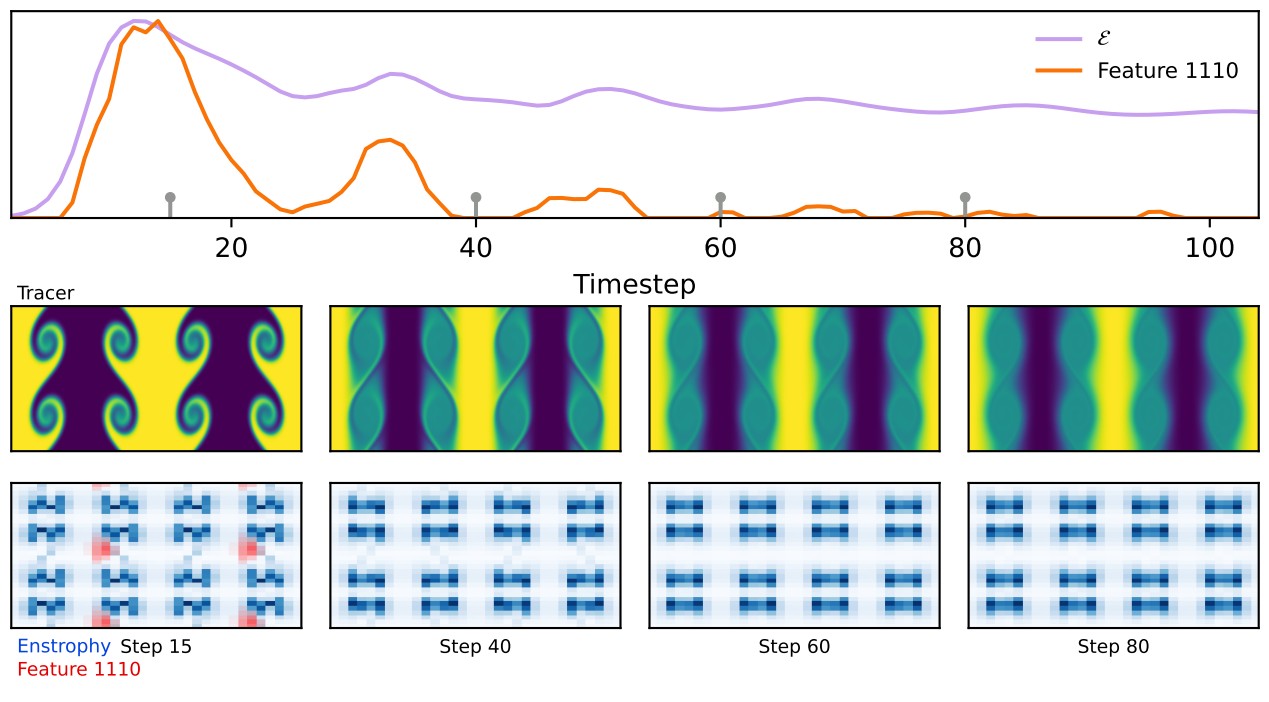

# Feature 1453

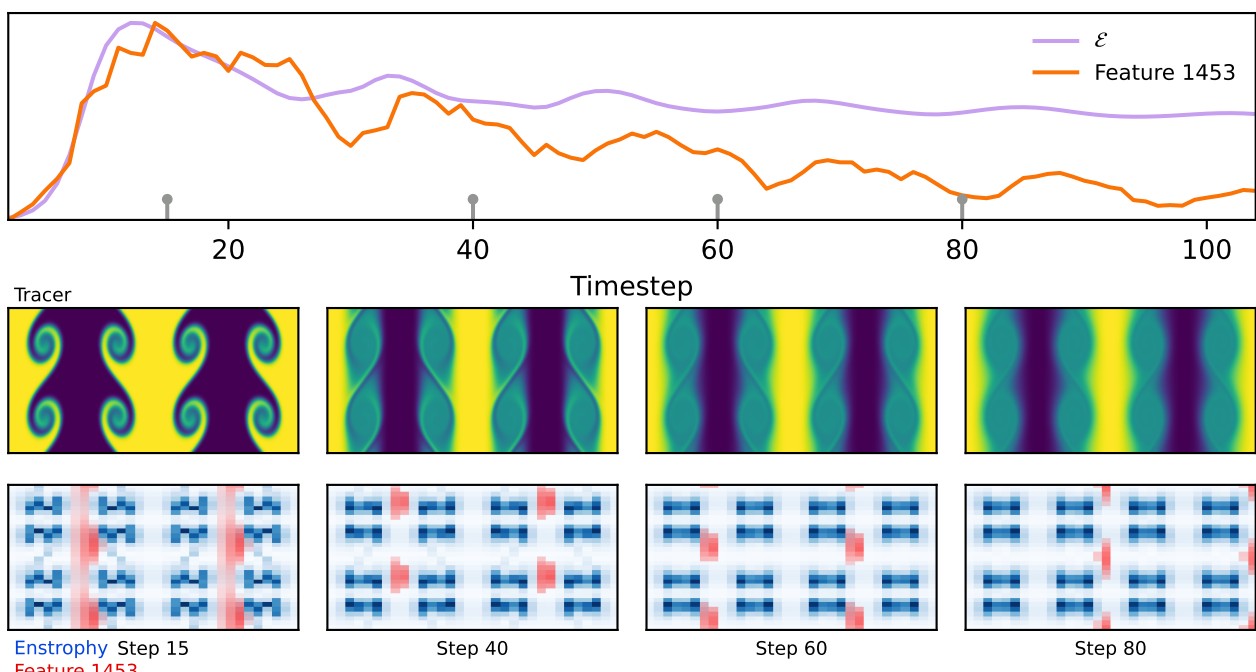

## Feature 12229

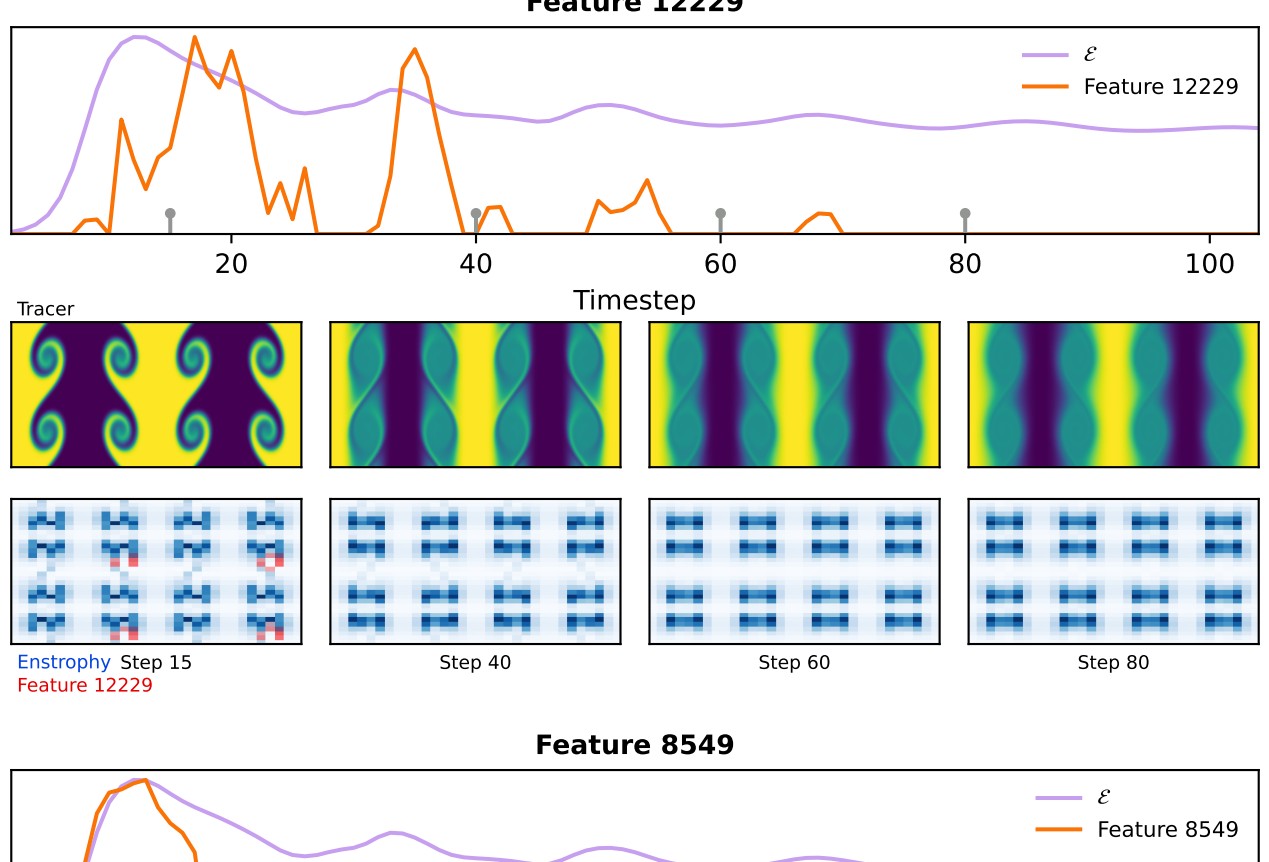

Tracer

Enstrophy Step 15
Feature 12229

Step 40

Step 60

Step 80

## Feature 8549

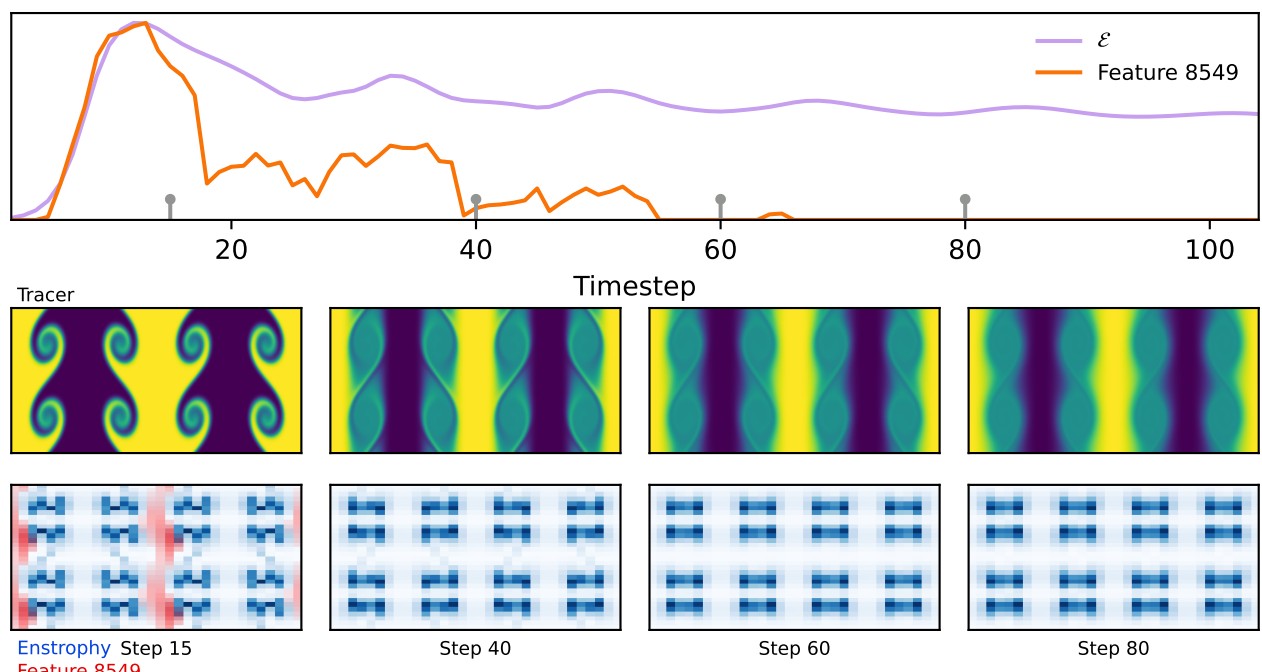

Tracer

Enstrophy Step 15
Feature 8549

Step 40

Step 60

Step 80

## Feature 5423

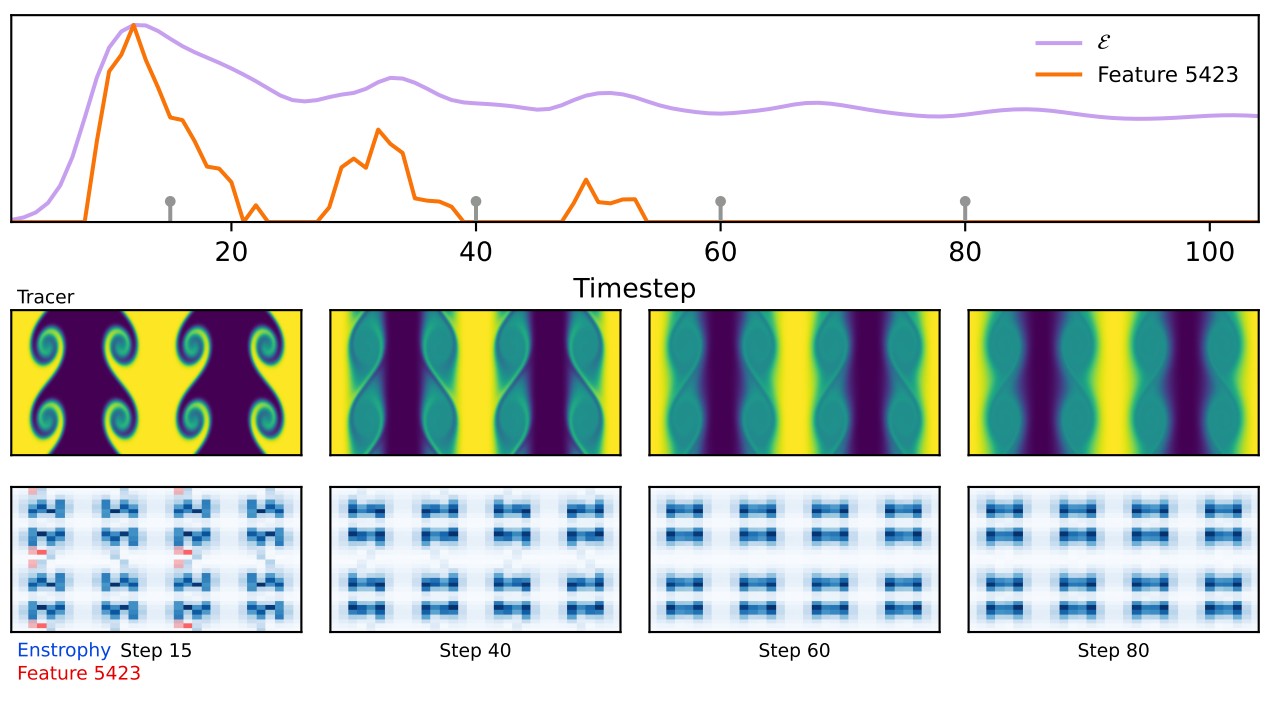

## Feature 5465

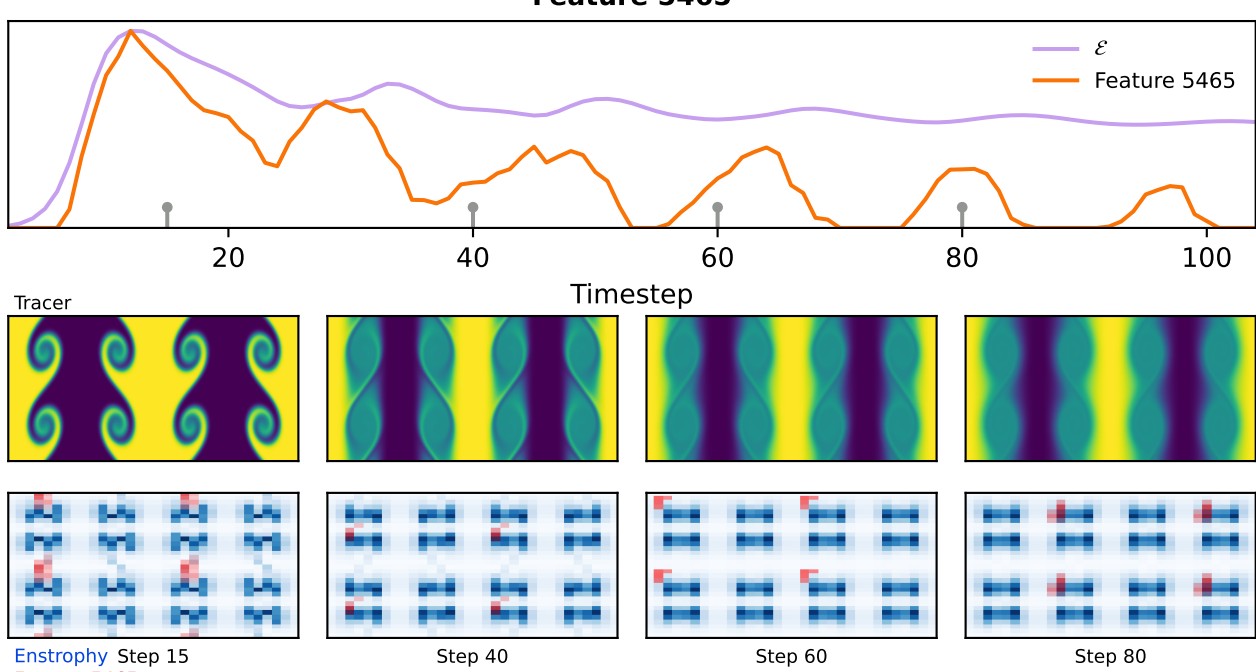

# Feature 15634

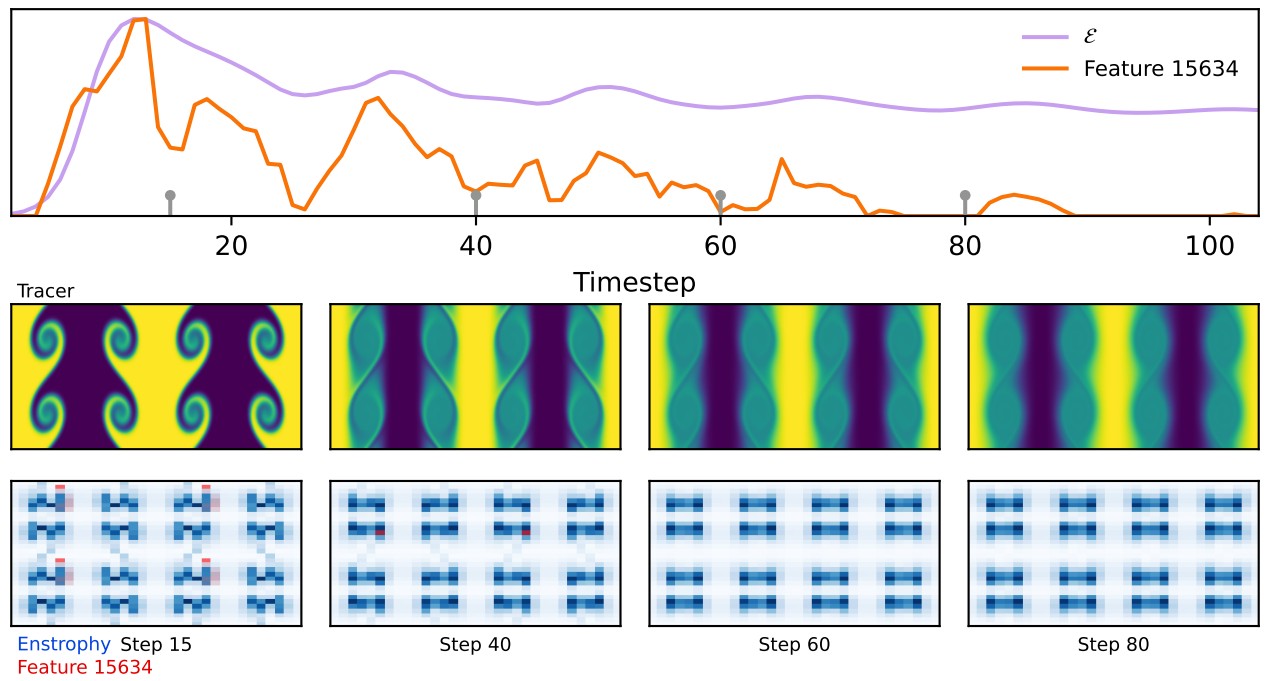

# Feature 5366

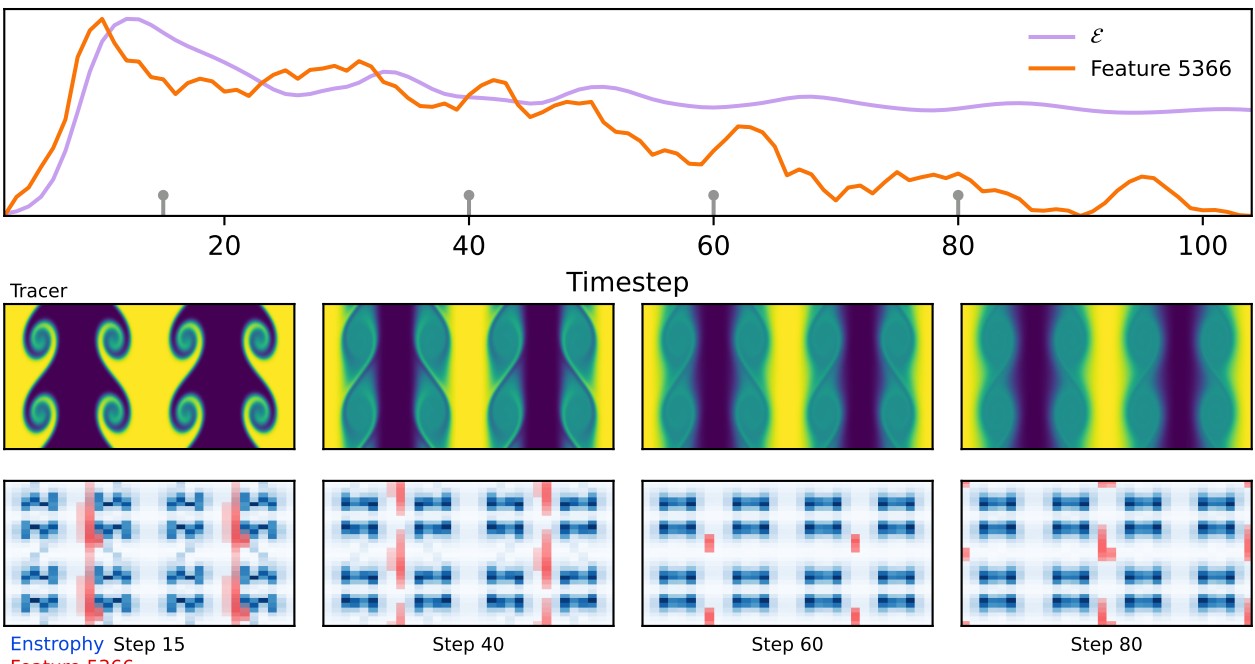

## A.9 TOP ENSTROPHY FEATURES: TRAJECTORY 3 (TRAJECTORY ALL FEATURES)

# Feature 7090

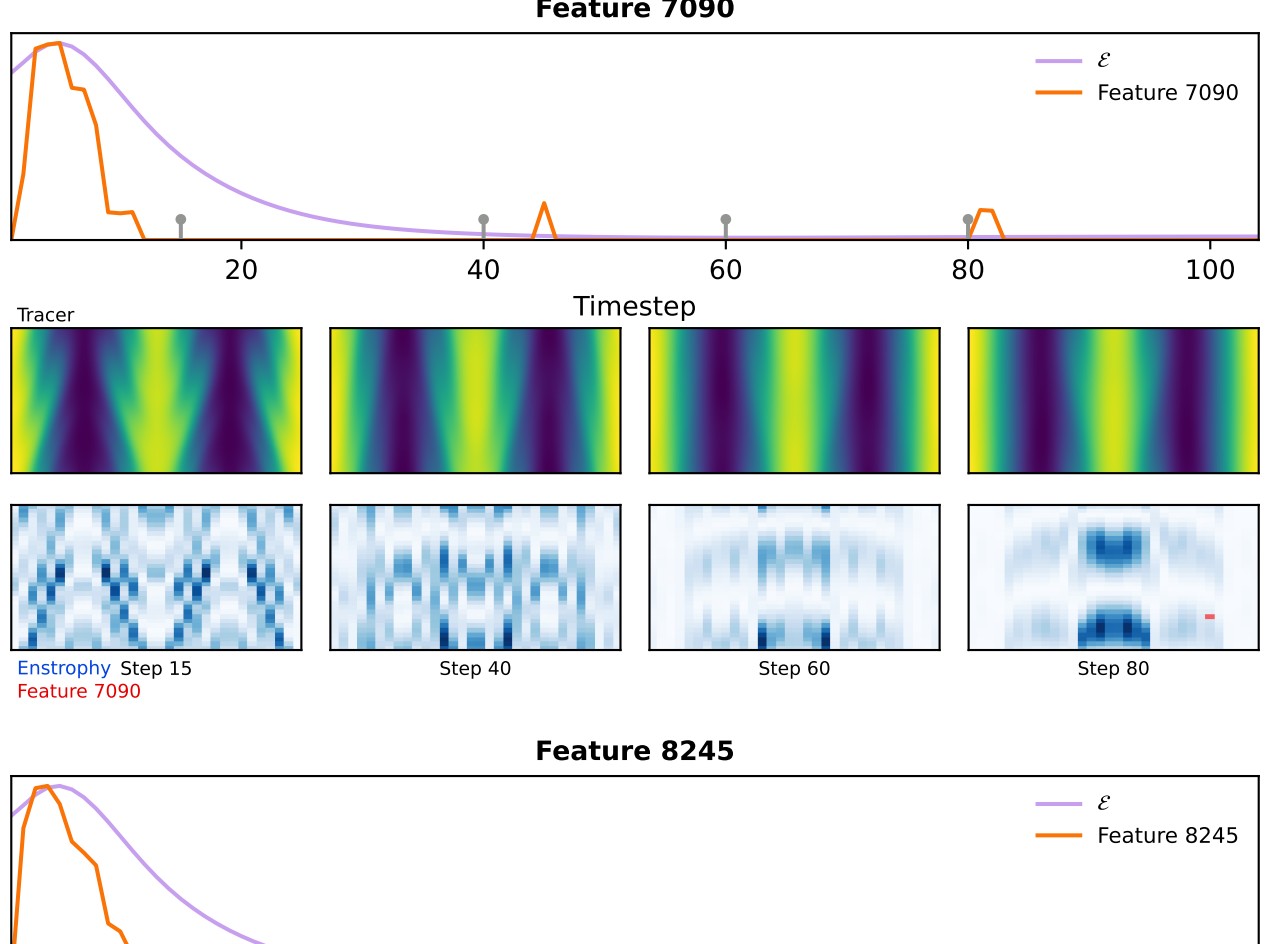

# Feature 8245

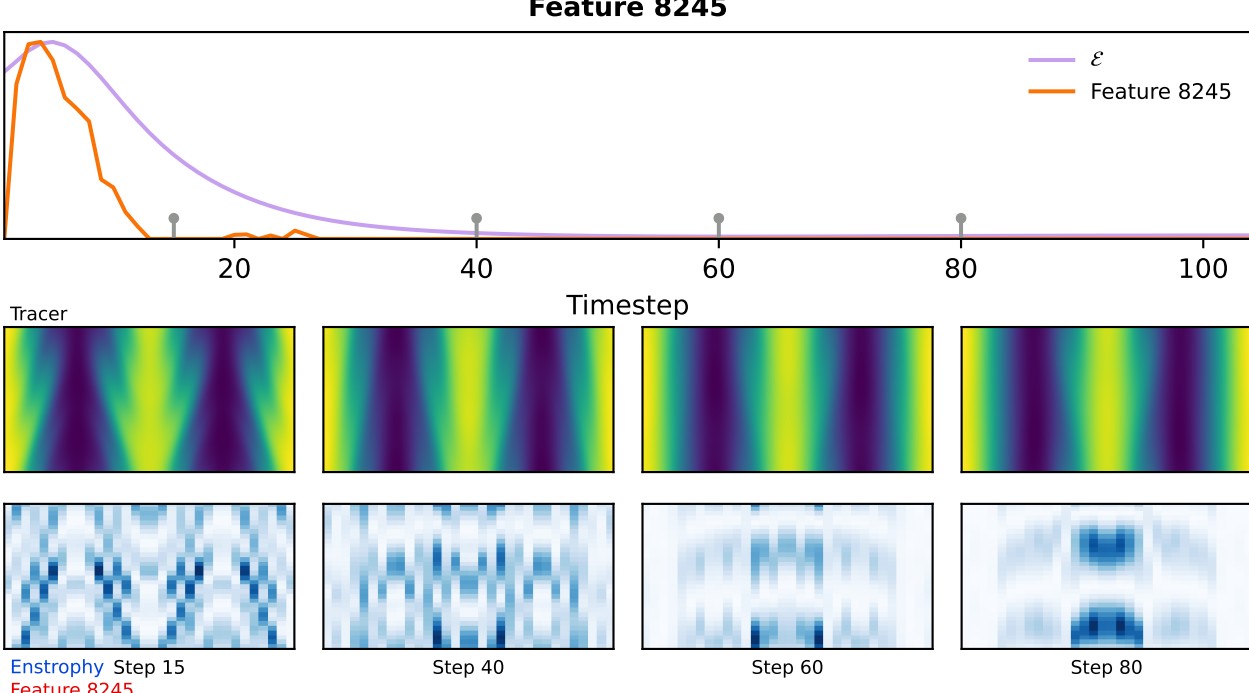

# Feature 4206

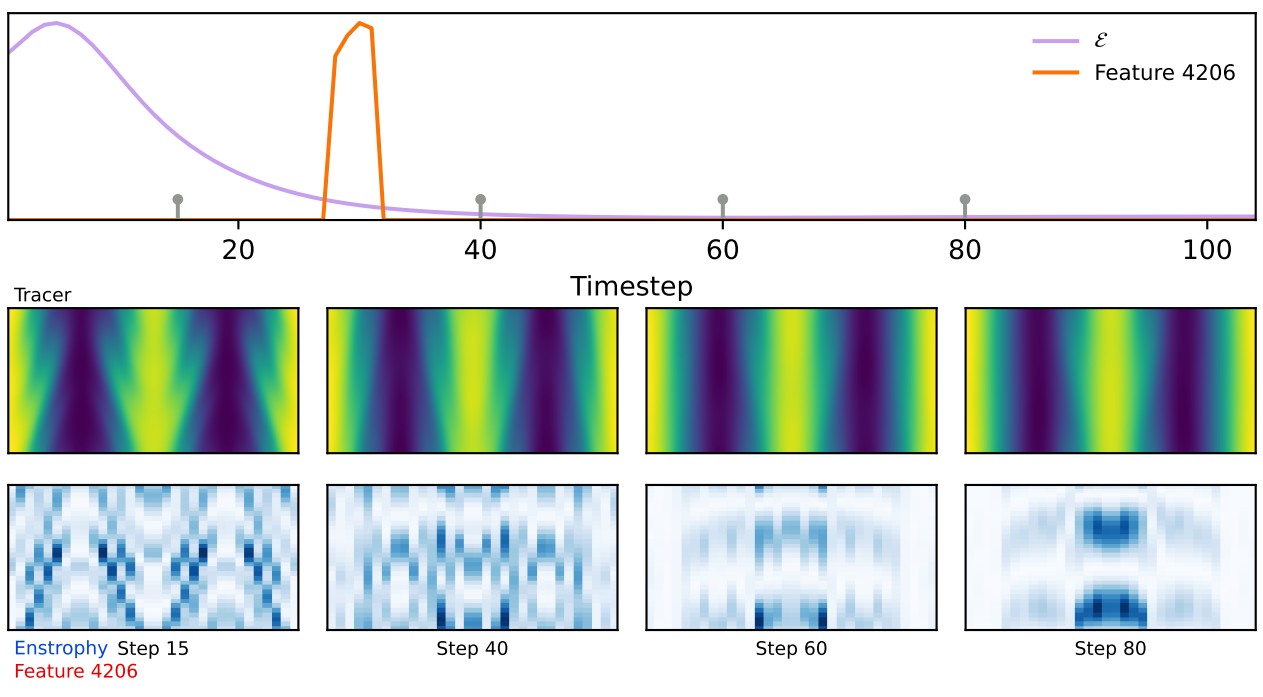

Tracer

Enstrophy Step 15
Feature 4206

Step 40

Step 60

Step 80

# Feature 15633

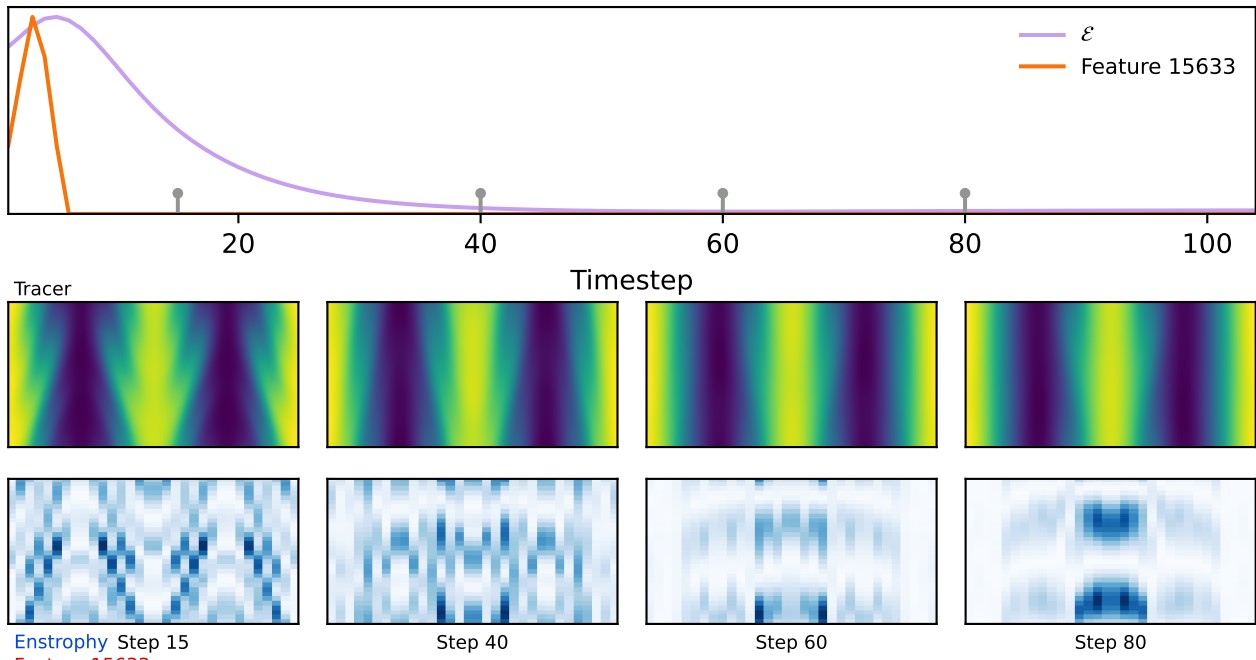

Tracer

Enstrophy Step 15
Feature 15633

Step 40

Step 60

Step 80

## Feature 17576

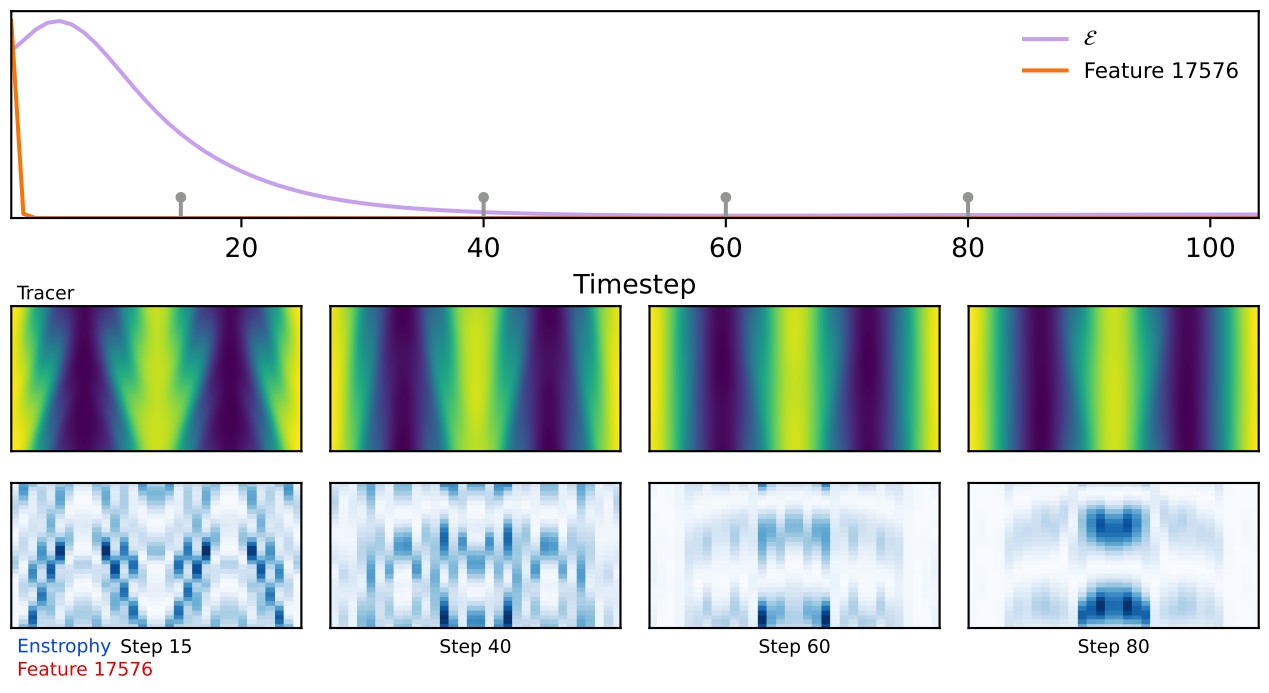

## Feature 15087

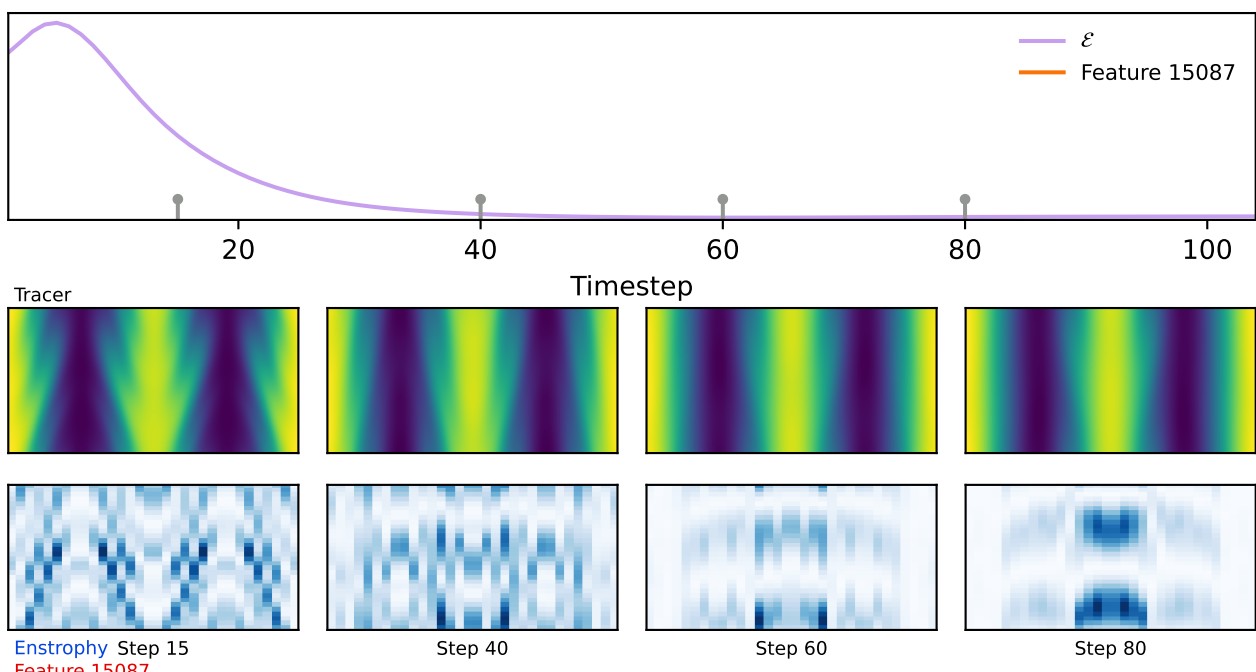

## Feature 6631

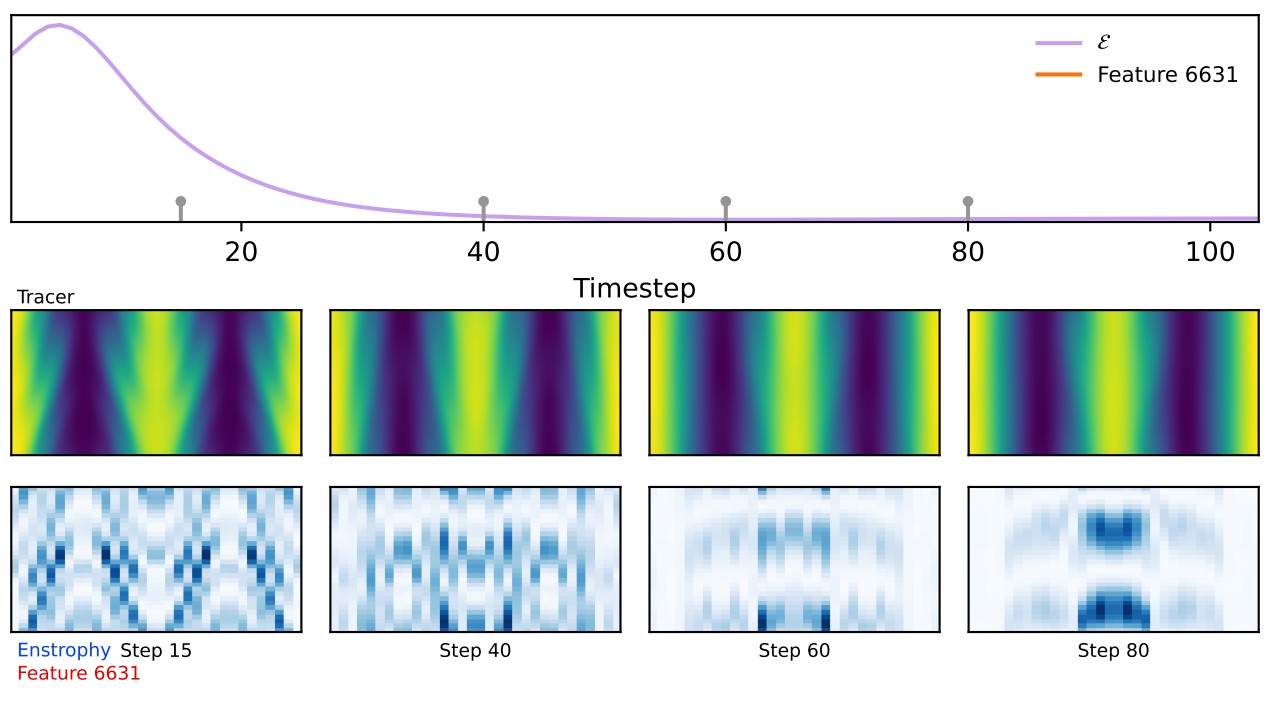

Tracer

Enstrophy Step 15
Feature 6631

Step 40   Step 60   Step 80

## Feature 20652

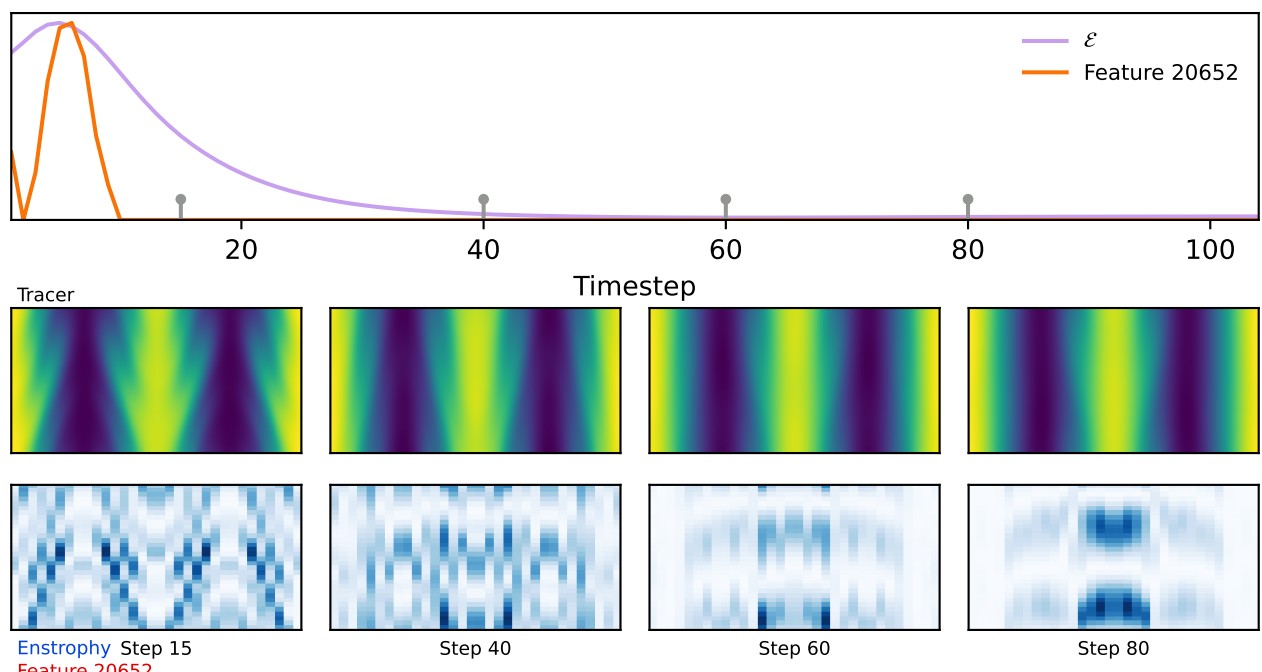

Tracer

Enstrophy Step 15
Feature 20652

Step 40   Step 60   Step 80

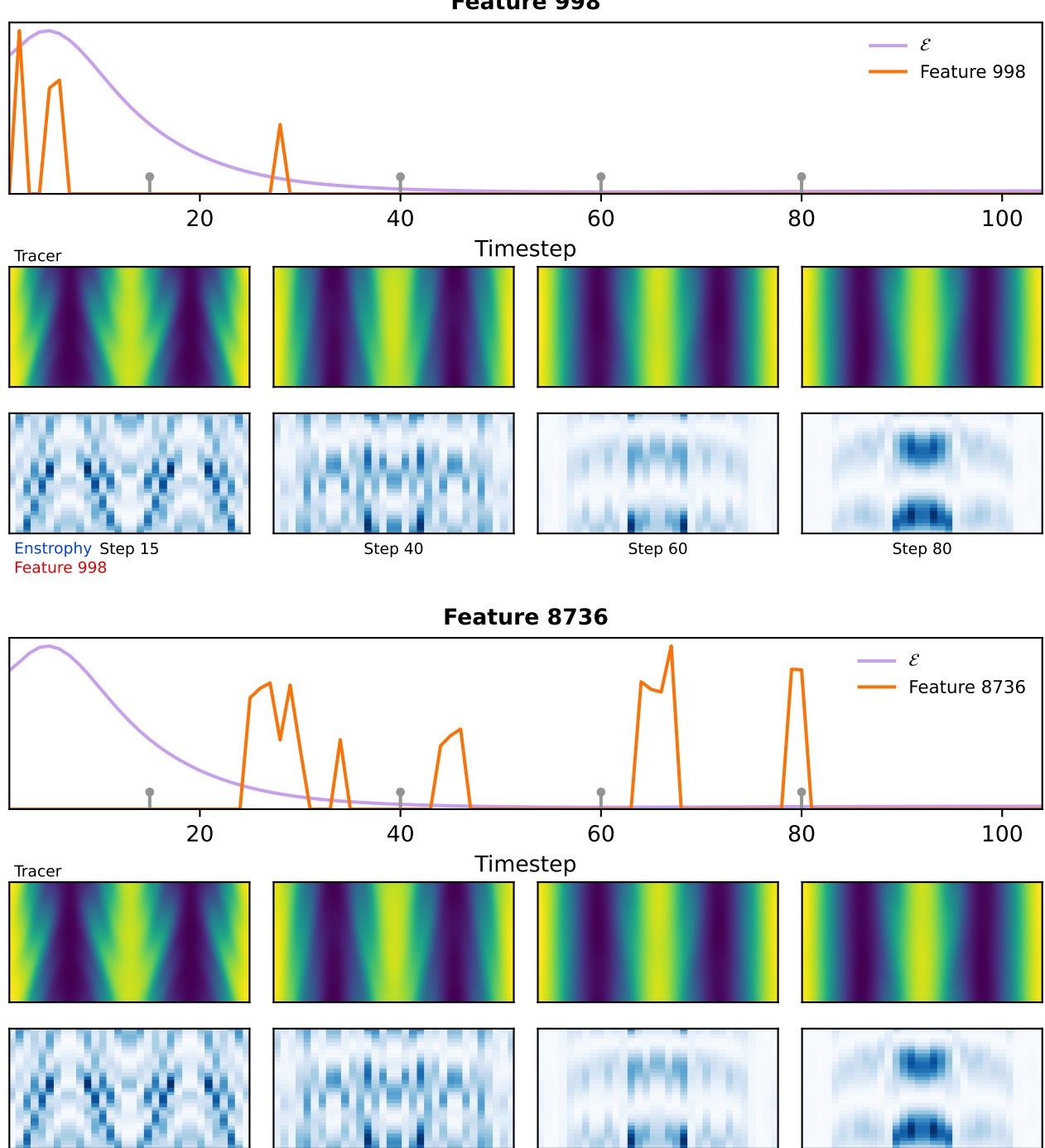

A.10   TOP ENSTROPHY FEATURES: TRAJECTORY 50 (TRAJECTORY ALL FEATURES)

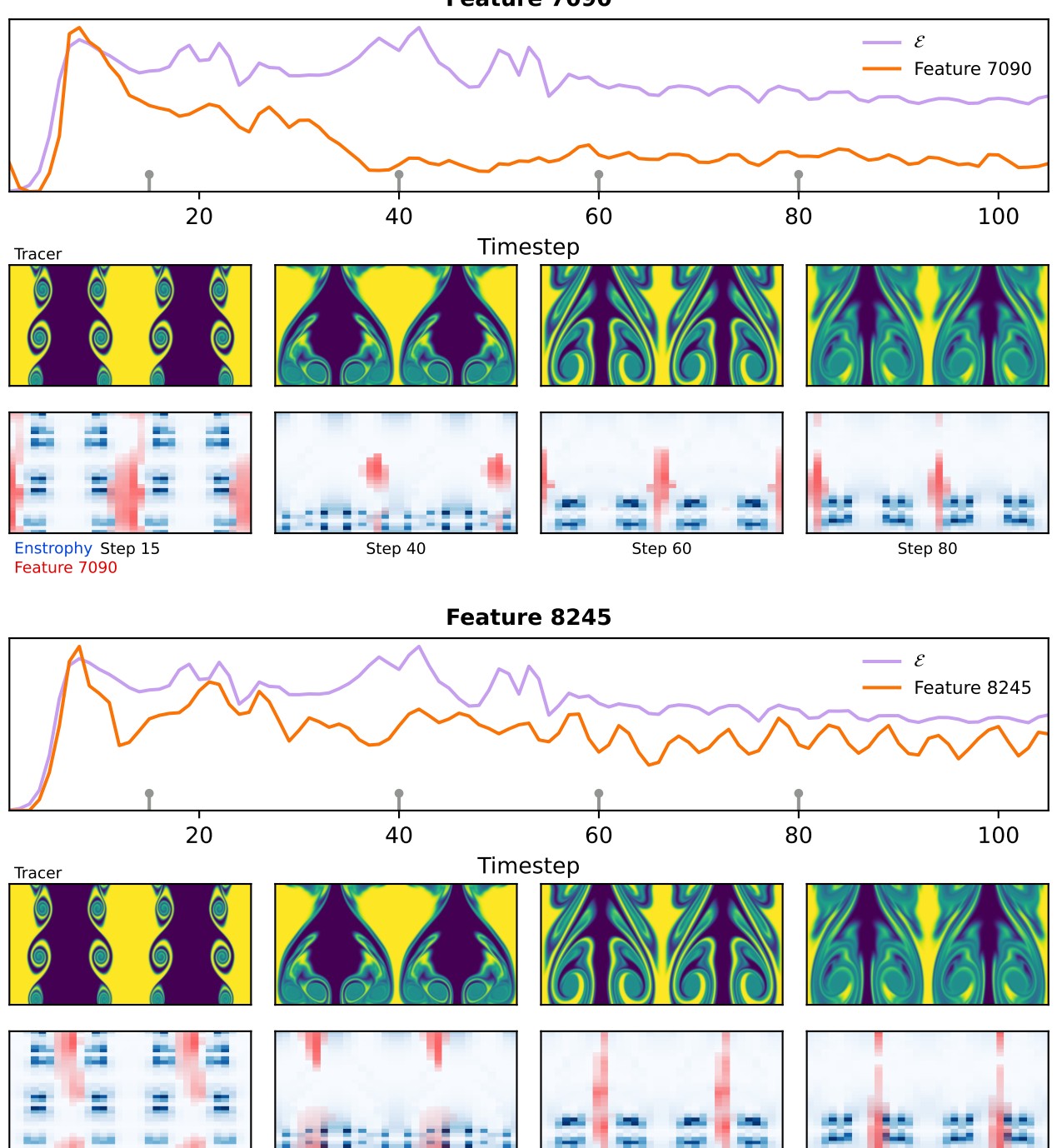

# Feature 4206

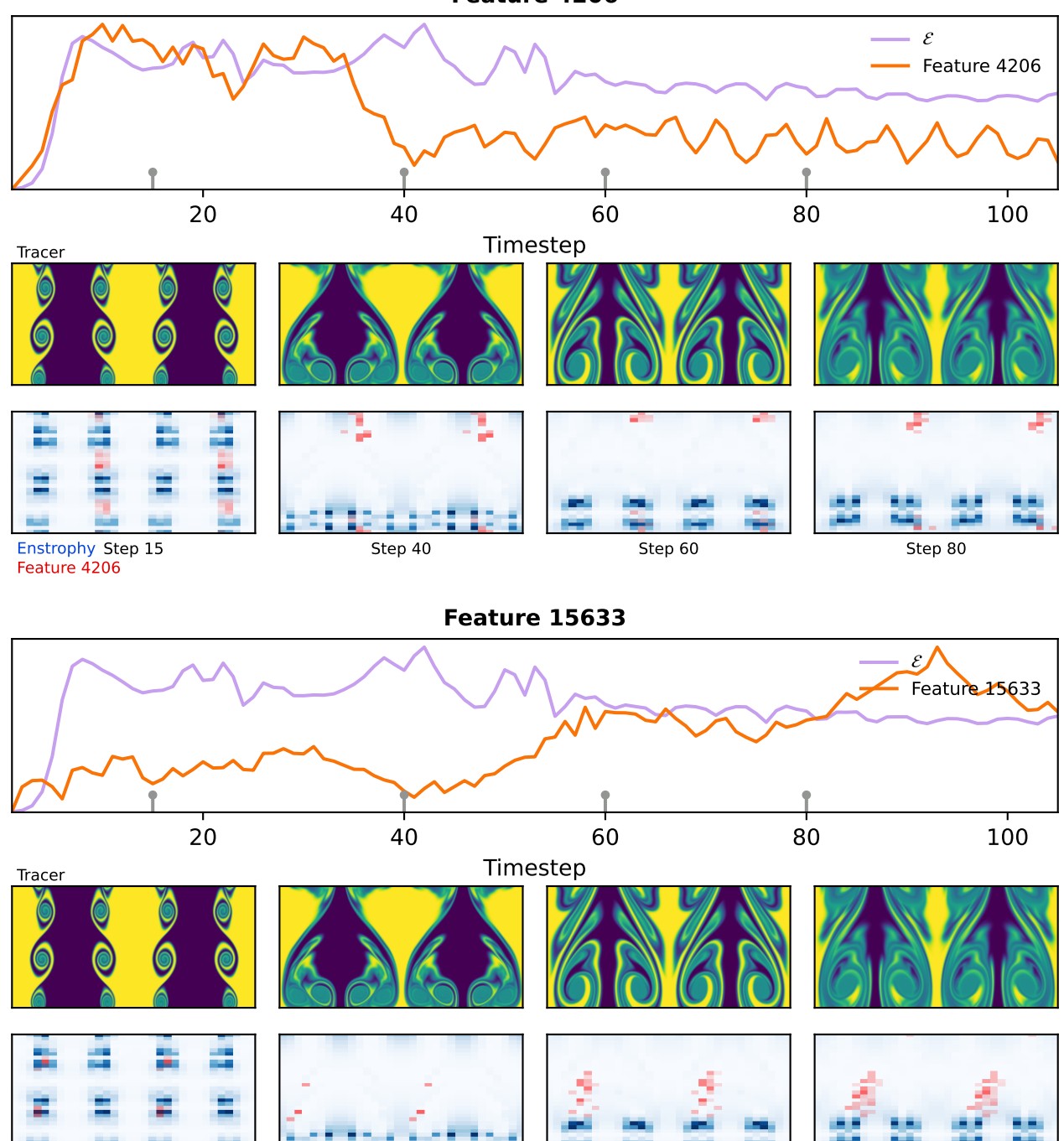

Tracer

Enstrophy Step 15
Feature 4206

Step 40    Step 60    Step 80

# Feature 15633

Tracer

Enstrophy Step 15
Feature 15633

Step 40    Step 60    Step 80

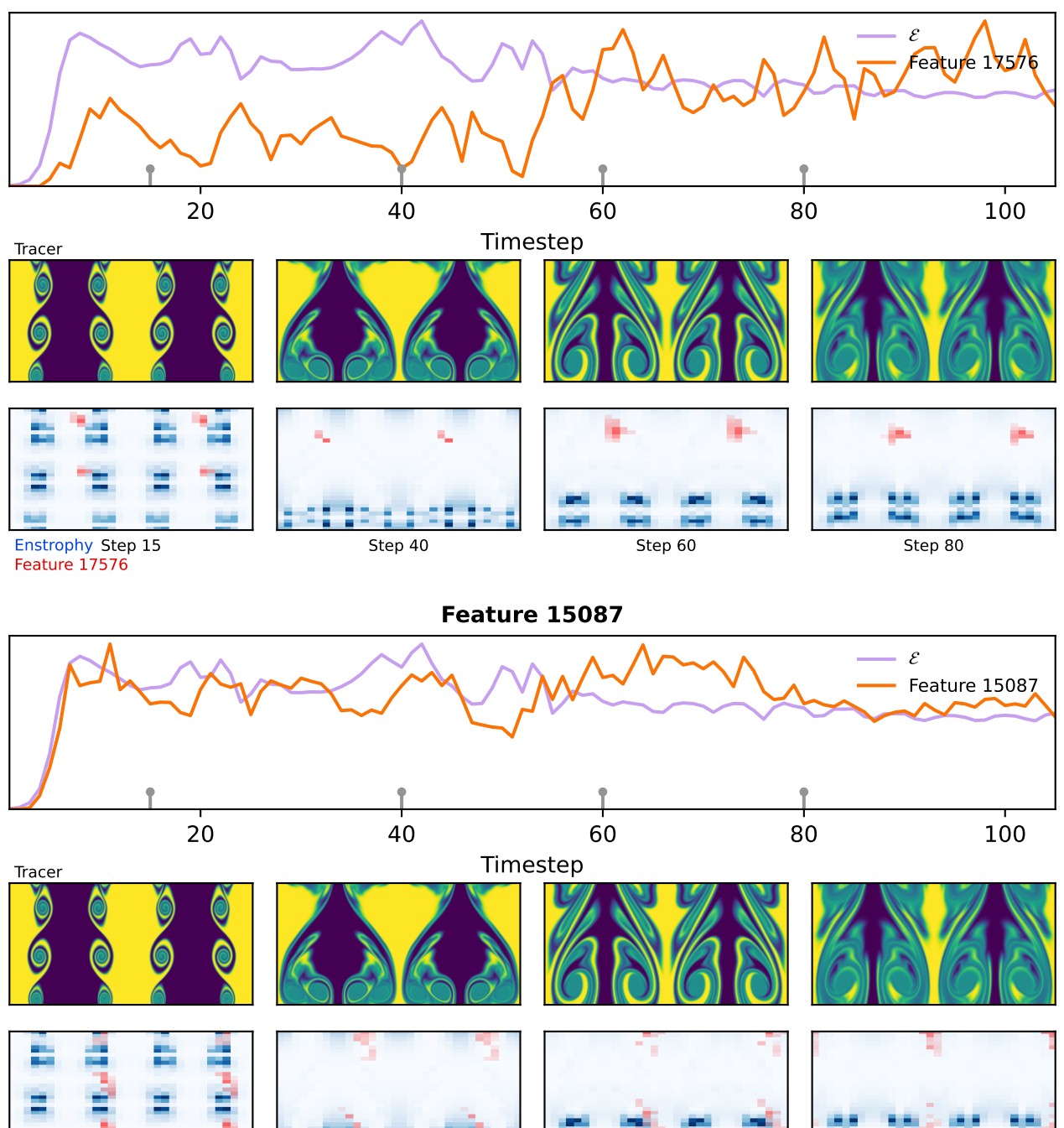

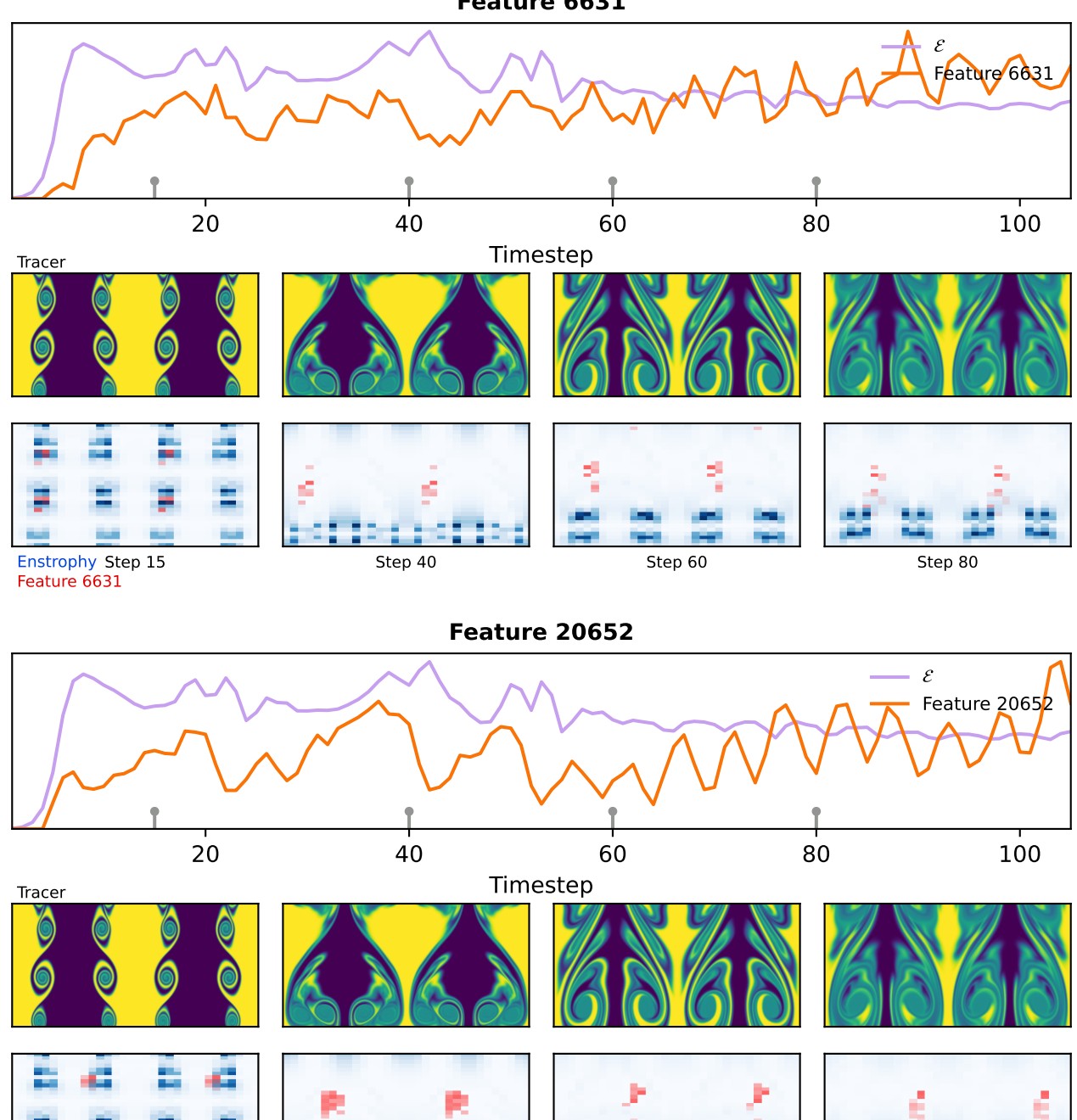

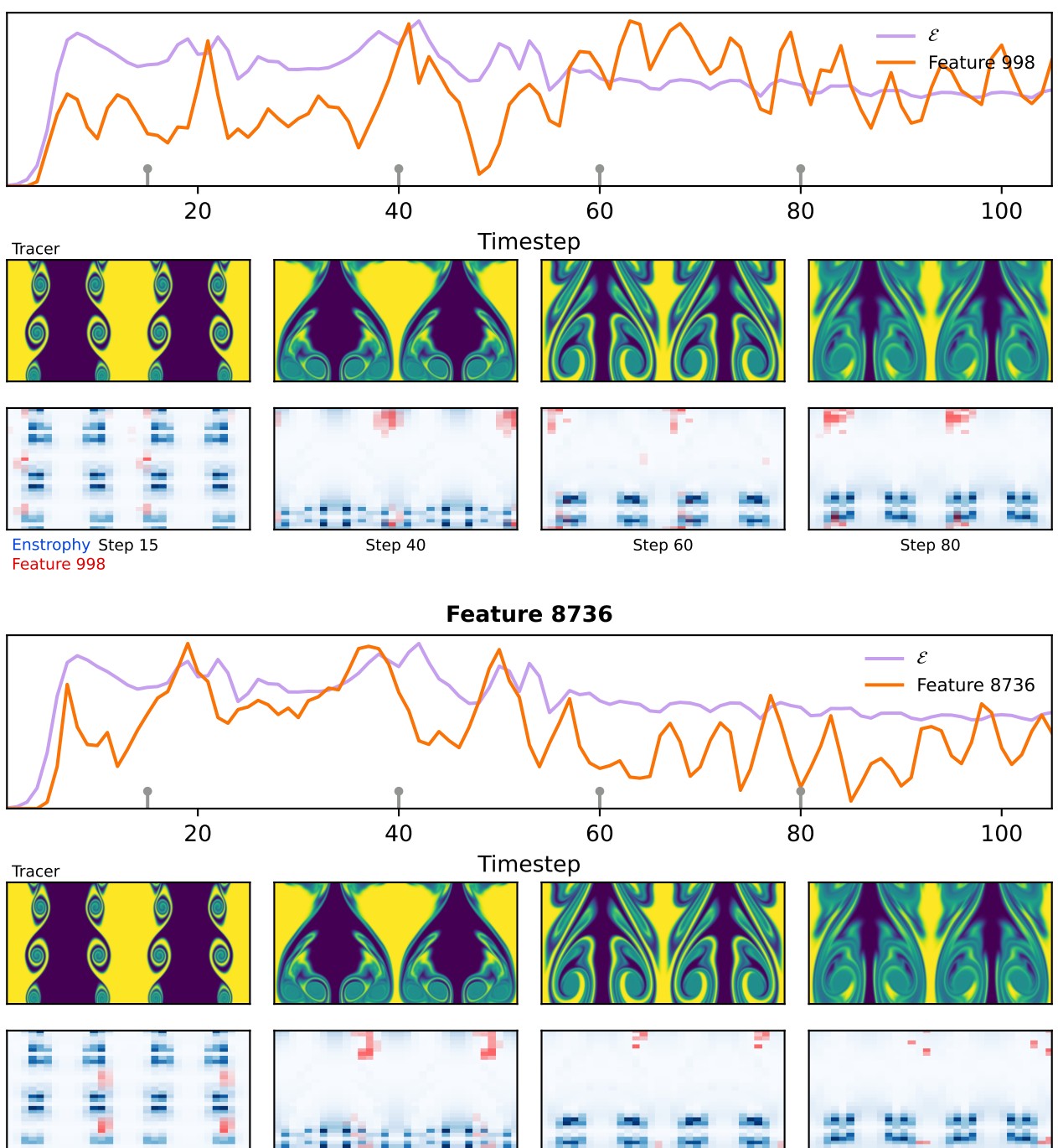

A.11   TOP ENSTROPHY FEATURES: TRAJECTORY 56 (TRAJECTORY ALL FEATURES)

## Feature 7090

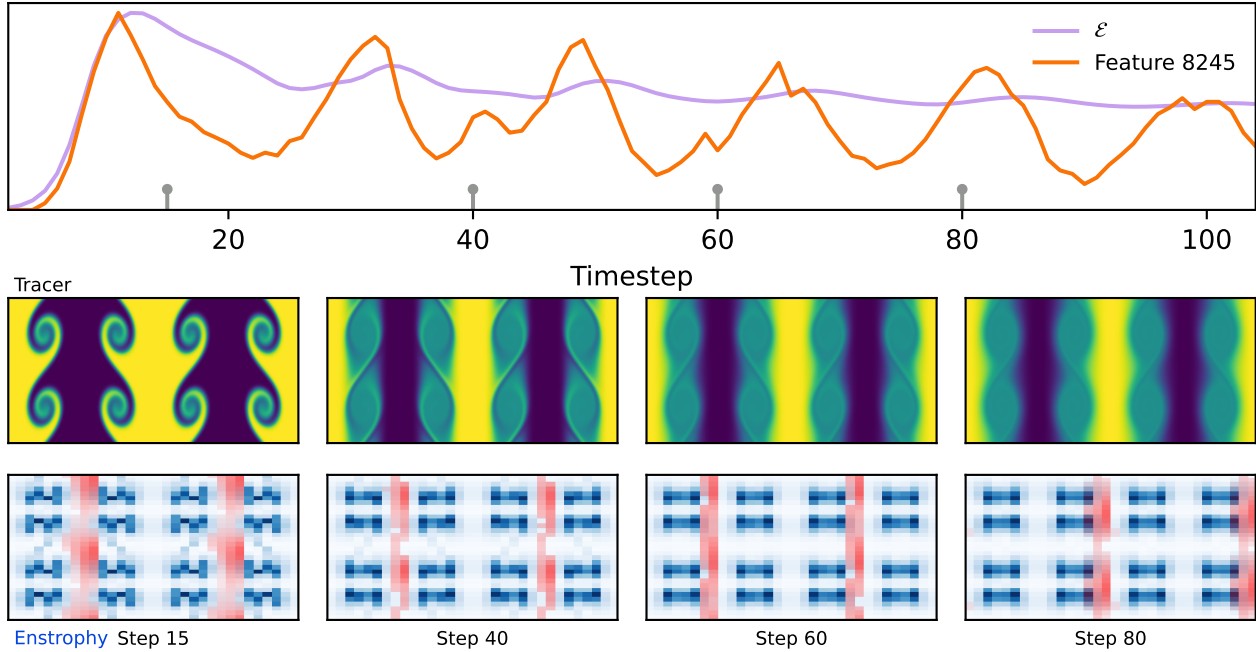

## Feature 8245

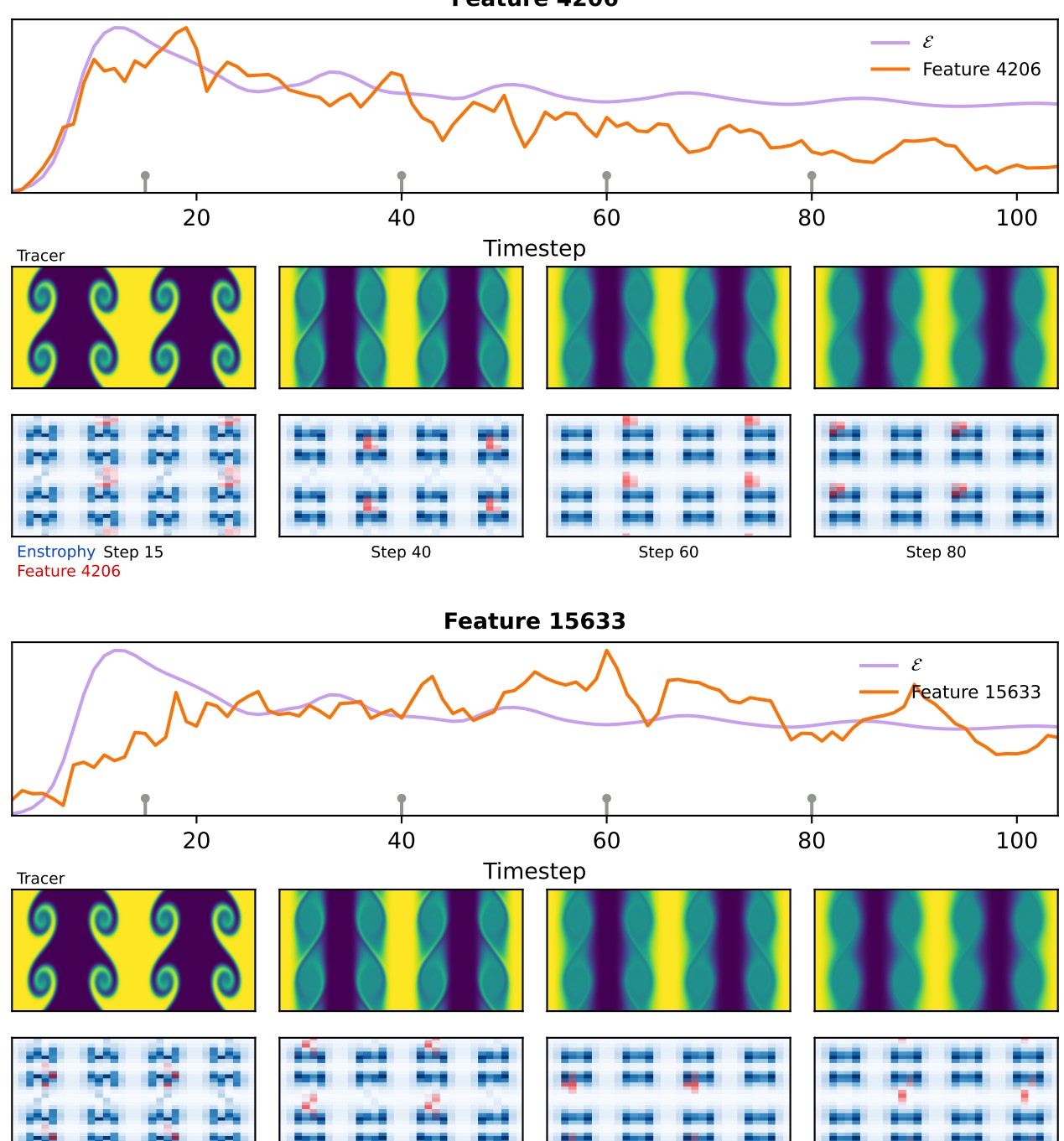

## Feature 17576

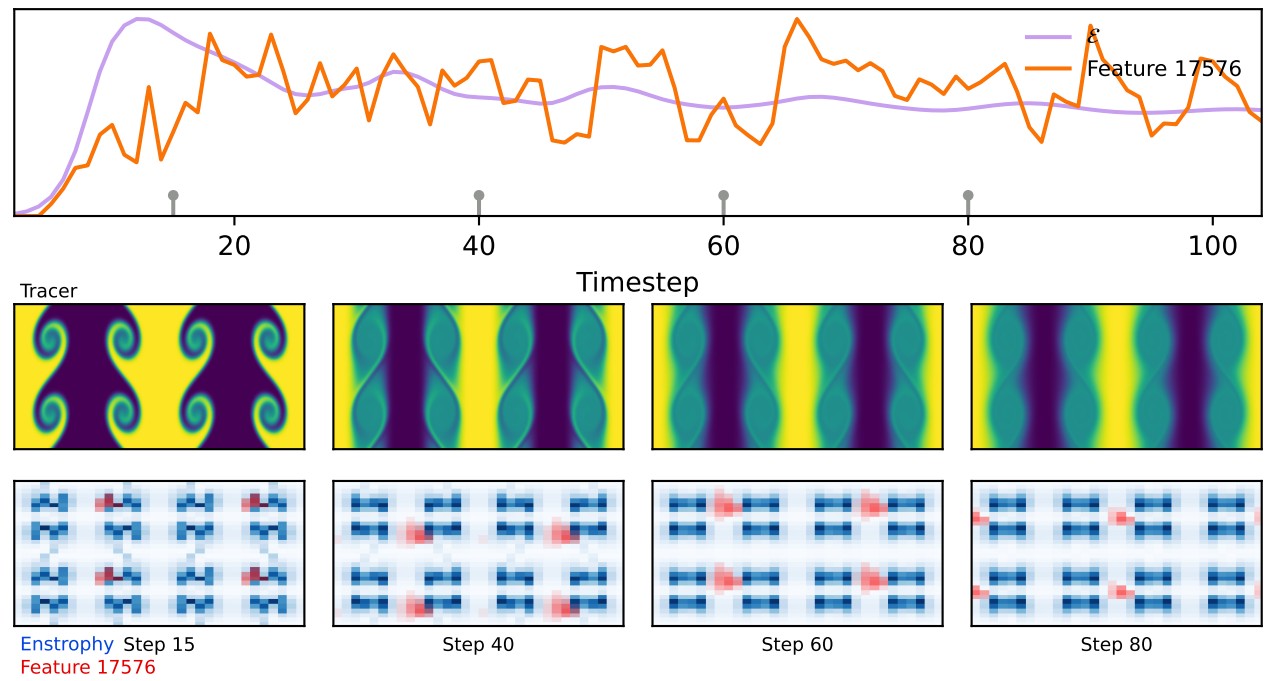

## Feature 15087

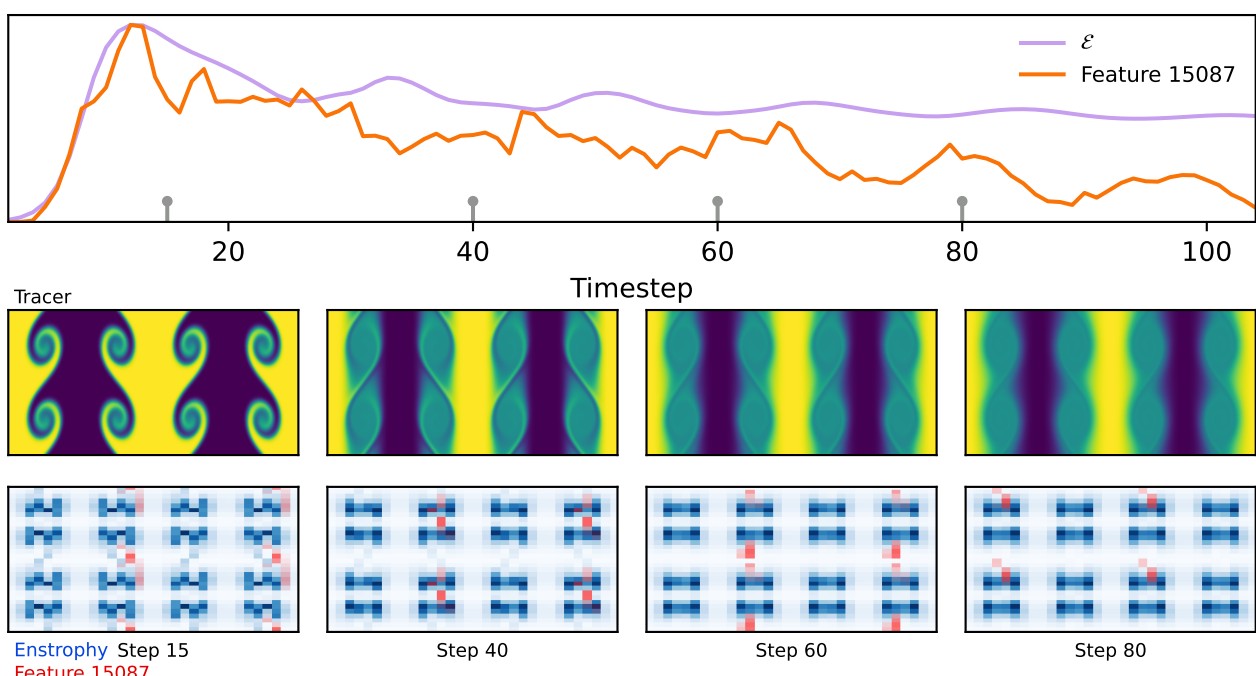

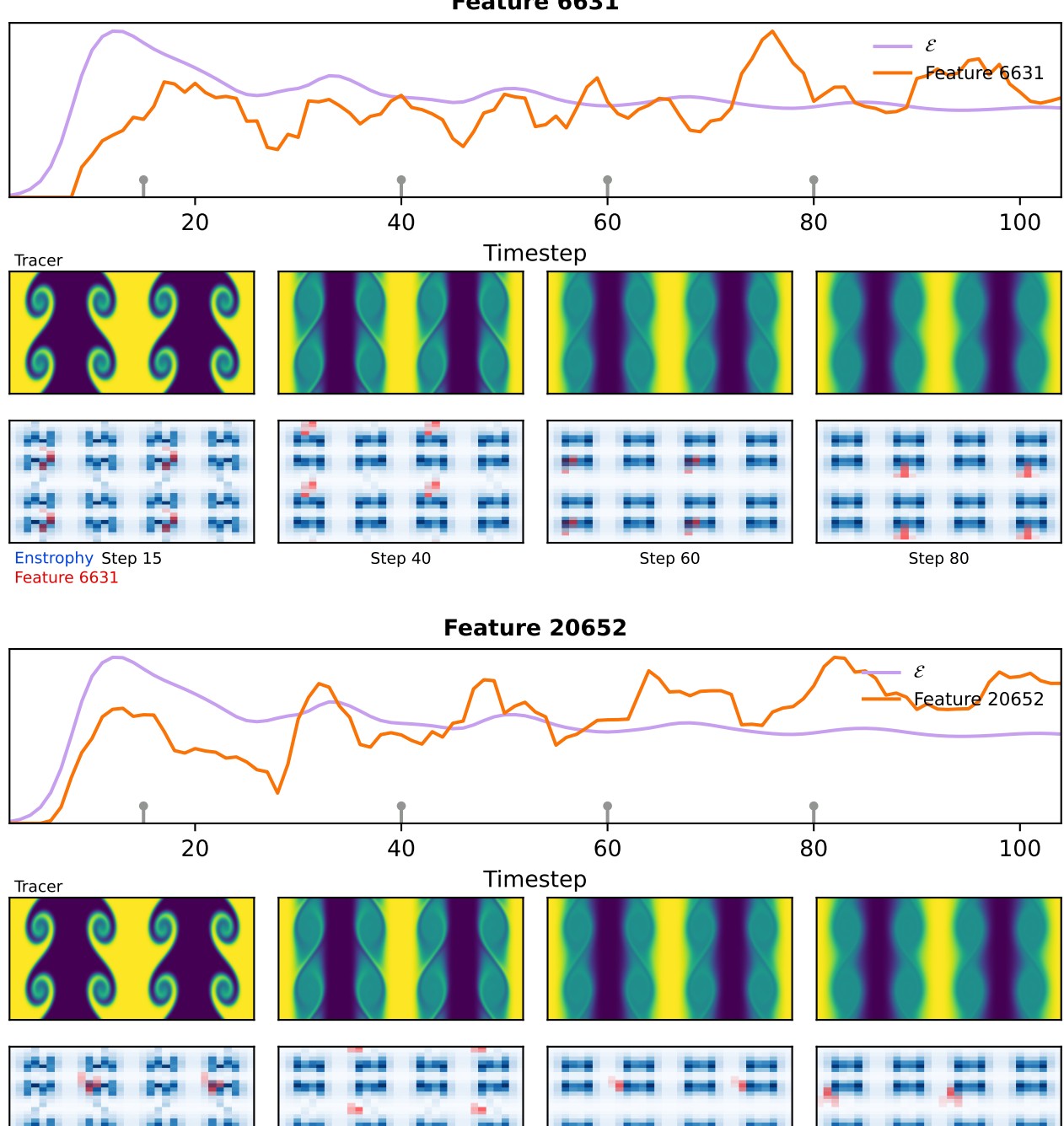

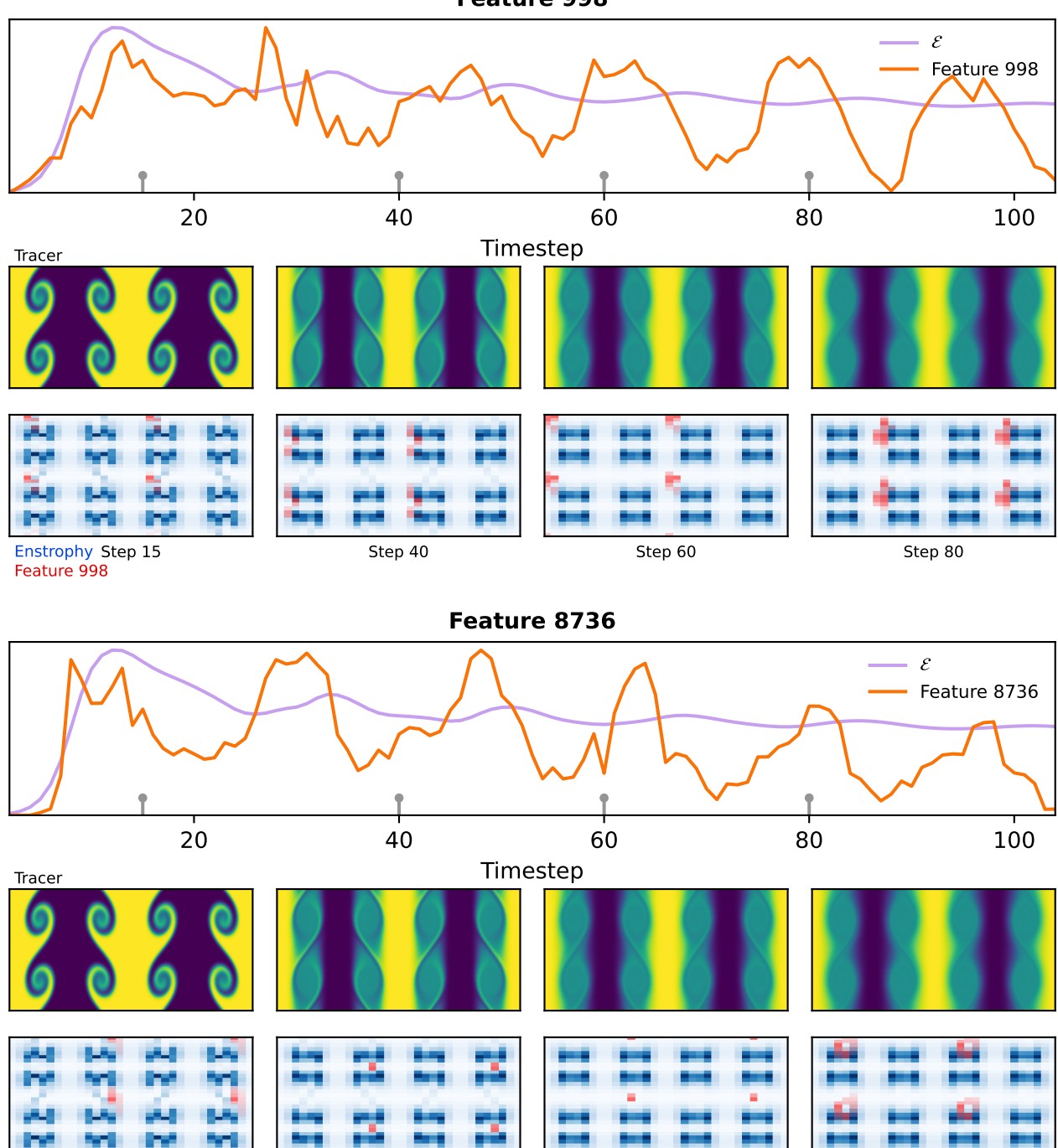