# OpenReview forum: "Sparse Probes, Murky Physics: Interpretability Challenges in a Foundation Model for Continuum Dynamics"
_ICLR.cc/2026/Workshop/FM4Science — ICLR 2026 Workshop FM4Science Poster_

### Official Review · Reviewer_y2cL · 2026-02-19
**Strong and Relevant Submission on Interpretability Challenges in Scientific Foundation Models**

**Rating:** 8
**Confidence:** 4

**Review:**

The paper investigates interpretability challenges in the Walrus foundation model (a 1.3B-parameter transformer for continuum dynamics) using sparse autoencoders (SAEs). Focusing on shear flow as a testbed, the authors train an SAE on activations from a middle layer and use enstrophy (a physical metric related to vorticity) to rank and triage features. They analyze feature activations across multiple simulations varying in Reynolds and Schmidt numbers, finding intermittent piecewise consistency (e.g., features activating similarly in early timesteps across simulations) but no clean mapping to standard physical decompositions. Output-level discrepancies (e.g., over-diffusion or over-localization in energy spectra) are linked to changes in feature usage. The work raises open questions about feature prioritization, separating artifacts from structure, and evaluating "different" representations in scientific FMs.

Relevance to Workshop Themes: This submission strongly aligns with FM4Science's emphasis on principles for scientific foundation models, including interpretability, mechanistic consistency, and integration with physical priors/constraints. It critiques how FMs like Walrus handle causal/spatiotemporal structure in physics data, uses benchmarks (enstrophy, energy spectra) for validation, and discusses failure modes key to reliable deployment in science.
Novel Contributions: Applies SAEs to a physics FM for the first time, introducing enstrophy-based feature triage to handle large feature sets (~22k). Cross-simulation and timestep analysis reveals intermittent coherence, providing empirical evidence on representation stability. Highlights diffusivity (physical and representational) as a new lens for interpretability challenges.
Technical Soundness and Rigor: SAE training is detailed and reproducible (Top-K with aux loss, hyperparameters swept). Evaluations include statistical tests (Spearman ρ, permutation nulls), visualizations (activation heatmaps overlaid on fields), and comparisons to numerical ground truth. Openly addresses limitations like single-layer focus and potential artifacts.
Potential Impact: By exposing gaps between outputs and internals, it advances discussions on trustworthy scientific FMs, potentially guiding hybrid systems or interpretable architectures. Relevant for domains like fluid dynamics, where understanding mechanisms is crucial for discovery.

Limited Scope and Generalizability: Analysis restricted to one layer, one scenario (shear flow), and a few simulations; lacks extension to other Walrus domains (e.g., astrophysics, plasma). Small dataset (112 trajectories for training) may not capture broader behaviors.
Lack of Causal Validation: Purely observational— no interventions like feature ablation, steering, or causal probing to confirm mechanistic roles. Correlations (e.g., ρ=0.85) are insightful but not sufficient for claiming "learning physics."
Feature Triage Challenges: Enstrophy ranking is grounded but permissive (~10% features significant), risking over-interpretation. No comparison to alternative metrics (e.g., energy, vorticity directly) or multi-metric approaches.
Minor Presentation Issues: Citations include future dates (2025), likely placeholders. Appendices (e.g., additional sims) are crucial but not detailed here; could better tie to workshop topics like UQ or multi-modality.

How might this approach scale to other Walrus domains or multi-layer analysis? Any plans for 3D flows or more diverse physics?
Could interventions (e.g., steering high-enstrophy features) reveal causal links to outputs, addressing the "effective vs. informative" question?
Plans to release SAE weights, code, or processed activations? This would enable community extensions.
How do you envision addressing diffusivity e.g., via physics-constrained SAEs or better triage?

This is a compelling submission that tackles core interpretability issues in scientific FMs, with strong ties to workshop aims on principles, evaluations, and workflows. It provides novel insights via physics-grounded methods and candidly highlights open problems, ideal for sparking discussion. Broader scope and causal tests would elevate it.

---

### Official Review · Reviewer_oxw6 · 2026-02-24
**A solid paper using SAE method to explain the weakness of the foundation model for shear flow in continuum dynamics**

**Rating:** 9
**Confidence:** 4

**Review:**

1. Summary

Unlike the stable, predictable N-body simulation, hydrodynamical simulation including continuum dynamics always serves as more unpredictable tasks for astrophysicists. Walrus by Polymathic came out with the ambition to be a generative foundation model for continuum dynamics, to predict next time step by the past snashots as input. This paper used shear flow as an example, which would exist around the boundaries between the fluids generate a lot of eddies and complex dynamical structures, thus making the physical quantity enstrophy to have a peak value. So this gives the authors a great way to measure whether the foundation model has captured the hidden physics: just separating the foundation model's hidden weights to different modes, and to see which mode has the largest correlation with the enstrophy. Inspired by this idea, they carried out the SAE method on the foundation model to separate those superpositioned modes, and found that some certain feature indeed have similar trend among different simulations, but they would get unstable due to the noise in later evolution. They confirmed this to be the reason why Walrus cannot perform well at small scale modes. Overall, it is a great paper with solid workload, to open the black box and well explain foundation model's behavior.

2. Strengths

- A timely work for the foundation model Walrus to point out their problem of small scale distortion. This paper has used the tool of energy spectra analysis from the perspective of a physicist, to locate the problem of over-estimation at small scales. This may not clearly appear in the original study of Walrus since just smoothing out the up and downs in small scales would be a quite easy way for model to optimize the loss function.

- The good choice of specific region in continuum dynamics and good metric borrowed from physics. Instead of looking at all kinds of continuum simulation at all physical regions, they focused on the shear flow and took good advantage of the existing quantity oof enstrophy in hydrodynamical physics. Then they have a quite clean and feasible way to determine a "representative trend" for the simulation, and makes the explanation later on possible.


3. Weaknesses & Critiques

- How to explain those features do not follow the enstrophy's trend, but still share similar parttern among different simulations? In the appendix of this paper, I have seen that there are also some features would just follow enstrophy at the very beginning and have a quick smooth drop then. Can this be called a different mode compared to the 8245, and be explained somehow by other physical quantities? Maybe it worth a try.

---

### Official Review · Reviewer_9KK3 · 2026-02-25
**Review for "Sparse Probes, Murky Physics: Interpretability Challenges in a Foundation Model for Continuum Dynamics"**

**Rating:** 8
**Confidence:** 4

**Review:**

This paper presents a case study of mechanistic interpretability for a physics foundation model (Walrus) using SAE probes, for the shear-flow example. The paper investigates if internal features in a scientific foundation model align with meaningful physical structure. A few strengths are the physics based analysis through enstrophy/spectral diagnostics and the inspection of feature behavior across trajectories/simulations, showing higher coherence earlier in trajectories..

The main weaknesses are well addressed by the authors in the discussion: primarily that it is difficult to attribute the findings to gaps in the foundation model representation compared to in the analysis method. The paper also shows that the current enstrophy-based feature ranking is only moderately selective. There may be potential for causal attribution in future work to better address this limitation. Overall, I would have liked to see a clearer conclusion from this analysis however the discussion was suitable and motivating for future work both for designing and analyzing foundation models for PDEs. These postmortems have proven very useful in other fields and this is a solid step for scientific machine learning.

I found this paper useful and interesting and trust the audience of this workshop would as well, I strongly recommend it for acceptance.

---

### Official Review · Reviewer_wSZ8 · 2026-02-25
**A Timely but Limited Empirical Study for Mechanistic Interpretability in Scientific Foundation Models**

**Rating:** 5
**Confidence:** 3

**Review:**

This paper presents a thoughtful and timely investigation into the mechanistic interpretability of foundation-style models for continuum dynamics. The central question of "how to determine whether internally learned representations meaningfully reflect known physics, rather than merely reproducing outputs" is clearly articulated and well motivated. The work is original in shifting attention from performance benchmarking to the structure and stability of internal representations, particularly in domains where strong theoretical priors already exist. Its significance lies in foregrounding interpretability as a scientific validation problem rather than a purely technical one.

The use of sparse autoencoders (SAEs) combined with a physically grounded metric (enstrophy) to triage a large feature space is conceptually well framed. The reported discrepancies between emulator outputs and numerical simulations are described clearly, and the attempt to connect these discrepancies to specific feature recruitment patterns strengthens the interpretability narrative.

## Pros:

- The question of how to separate stable internal structure from analysis artifacts (e.g., SAE limitations, single-layer probing) is novel, timely, and highly relevant for scientific foundation models.
- The work proposes a concrete methodology for benchmarking internal representations against physically meaningful criteria, offering a structured way to evaluate mechanistic alignment.
- The identification and communication of output-level discrepancies are precise and clearly linked to feature-level behavior, improving transparency and diagnostic value.

## Cons:

- The scope is limited to a deliberately simple testbed (shear flow) and a single modeling scenario, which constrains the generalizability of the conclusions.
- The experimental methodology—probing a single layer with an SAE and using a predefined physical metric—while reasonable, appears technically straightforward and may not substantially advance interpretability tooling.
- The experimental results, as described, provide limited insight into resolving the core question; the evidence of “piecewise consistency” and intermittent structure remains suggestive rather than decisive.

Overall, I think the work addresses an important emerging problem conceptually, but its empirical depth and breadth may limit the strength of its conclusions.

---

### Official Review · Reviewer_Su4i · 2026-02-25
**Interpretability studies provide needed intuitive understanding of complex foundation models**

**Rating:** 8
**Confidence:** 3

**Review:**

The authors provide a qualitative study into how intermediate layers in the Walrus foundation model track important physcial quantities. In doing so they aim to interpret whether the model is able to understand important physical phenomena and quantities, or if the model has memorized solutions.

**Strengths**
- Interpretability and validation in scientific foundation models is crucial work. Unlike NLP systems, where errors are more tolerated and there are many ways to say the same thing, errors in physical systems (e.g. by invalidating hard constraints) lead to erroneous answers that can result in costly errors for a company.
- The authors provide an honest exploration of interpretability by recognizing their work's strengths and weaknesses, while not overstating their results. This is much appreciated.
- The authors show how some activations track enstrophy in incompressible shear flow during early times, but said activations' correlation to enstrophy decays with time.
- The authors show how Walrus is fairly accurate for low wavenumbers (large features) but has significant error at high wavenumbers (small features).
- Ultimately, the authors demonstrate that some features are tracking physical quantities (e.g. enstrophy) for some times, and thus Walrus may be learning some physical principles. However, existing probes make claims like this very challenging to validate.

**Weaknesses**
The paper is generally well written but suffers from a lack of details.
- Some crucial concepts and variables that may be trivial to the authors are not explained and lead to confusion: tracers, which features they are highlighting, what part of the SAE are the features coming from, etc.
- This work would greatly benefit from a figure that describes the architecture and shows examples of the input and output.
- The authors mention specific features very often, but it is unclear to me where these features are coming from. Therefore, I am missing some crucial information and cannot understand this work to the degree I would like to. Are the features coming from the Walrus layer, the middle of the SAE, or the output of the SAE? Having an architecture figure and pointing to those features would be extremely helpful.

I think the content is important and quite helpful for the community so I will give it a high rating, but the authors need to reevaluate their experimental description and results section from the viewpoint of a reader who may be seeing plots like this and reading words like "tracers" for the first time.

---

### Meta-Review · Area_Chair_bHe4 · 2026-02-27

**Recommendation:** Accept (Poster)
**Confidence:** 4

**Metareview:**

The average review score is above 6, which means reviewers recommended an acceptance.

---

### Decision · Program_Chairs · 2026-03-03

Accept (Poster)